# On the Global Convergence of Gradient Descent for multi-layer ResNets in the mean-field regime

## Abstract

Finding the optimal configuration of parameters in ResNet is a nonconvex minimization problem, but first order methods nevertheless find the global optimum in the overparameterized regime. We study this phenomenon with mean-field analysis, by translating the training process of ResNet to a gradient-flow partial differential equation (PDE) and examining the convergence properties of this limiting process. The activation function is assumed to be 2-homogeneous or partially 1-homogeneous; the regularized ReLU satisfies the latter condition. We show that if the ResNet is sufficiently large, with depth and width depending algebraically on the accuracy and confidence levels, first order optimization methods can find global minimizers that fit the training data.

## 1 Introduction

Training of multi-layer neural networks (NN) requires us to find weights in the network such that its outputs perfectly match the prescribed outputs for a given set of training data. The usual approach is to formulate this problem as a nonconvex minimization problem and solve it with a first-order optimization method based on gradient descent (GD). Extensive computational experience shows that in the overparametrized regime (where the total number of parameters in the NN far exceeds the minimum number required to fit the training data), GD methods run for sufficiently many iterations consistently find a global minimum achieving the zero-loss property, that is, a perfect fit to the training data.

What is the mechanism that allows GD to perform so well on this large-scale nonconvex problem?

Part of the explanation is that in the overparametrized case, the parameter space contains many global minima, and some evidence suggests that they are distributed throughout the space, making it easier for the optimization process to find one such solution. Many approaches have been taken to characterize this phenomenon more rigorously, including landscape analysis, the neural tangent kernel approach, and mean-field analysis. All such viewpoints aim to give an idea of the structure and size of the NN required to ensure global convergence.

Our approach in this paper is based on mean-field analysis and gradient-flow analysis, the latter being the continuous and mean-field limit of GD. We will examine residual neural networks (ResNets), and study how deep and wide a ResNet needs to be to match the data with high accuracy and high confidence. To relax the assumptions on the activation function as far as possible, we follow the setup in (Chizat & Bach, 2018), which requires this function to be either 2-homogeneous or partially 1-homogeneous. We show that both depth and width of the NN depend algebraically on $\epsilon$ and $\eta$, which are the accuracy and confidence levels, respectively.

Mean-field analysis translates the training process of the ResNet to a gradient-flow partial differential equation (PDE). The training process evolves weights on connections between neurons. When dealing with wide neural networks, instead of tracing the evolution of each weight individually, one can record the evolution of the full distribution of the weight configuration. This perspective translates the coupled ordinary differential equation system (ODE) that characterizes evolution of individual weights into a PDE (the gradient-flow equation) characterizing the evolution of the distribution. The parameters in the PDE naturally depend on the properties of the activation functions. Gradient-flow

analysis is used to show that the PDE drives the solution to a point where the loss function becomes zero. We obtain our results on zero-loss training of ResNet with GD by translating the zero-loss property of the gradient-flow PDE back to the discrete-step setting.

This strategy of the proof was taken in an earlier paper (Ding et al., 2021) where multi-layer ResNets were also analyzed. The main difference in this current paper is that the assumptions on the activation function and the initial training state for obtaining the global convergence are both much relaxed. This paper adopts the setup from (Chizat & Bach, 2018) of minimal Lipschitz continuity requirements on the activation function. Furthermore, the paper (Ding et al., 2021) required a dense support condition to be satisfied on the final parameter configuration has a support condition. This condition is hard to justify in any realistic setting, and is discared from current paper. Further details on these issues appear in Section 3.

We discuss the setup of the problem and formally derive the continuous and mean-field limits in Section 2. In Section 3, we discuss related work, identify our contribution and present the main theorem in its general terms. After precise definitions and assumptions are specified in Section 4, we present the two main ingredients in the proof strategy. The mean-field limit is obtained by connecting the training process of the ResNet to a gradient-flow PDE in Section 5, and the zero-loss property of the limiting PDE is verified in Section 6. The main theorem is a direct corollary of Theorem 5.1 and Theorem 6.1 (or Theorem 6.2).

## 2 RESNET AND GRADIENT DESCENT

The ResNet can be specified as follows:

$$z_{l+1}(x) = z_l(x) + \frac{1}{ML} \sum_{m=1}^{M} f(z_l(x), \theta_{l,m}), \quad l = 0, 1, \ldots, L-1, \tag{1}$$

where $M$ and $L$ are the width and depth, respectively; $z_0(x) = x \in \mathbb{R}^d$ is the input data; and $z_L(x)$ is the output from the last layer. The configuration of the NN is encoded in parameters $\Theta_{L,M} = \{\theta_{l,m}\}_{l=0,m=1}^{L-1,M}$, where each parameter $\theta_{l,m}$ is a vector in $\mathbb{R}^k$ and $f : \mathbb{R}^d \times \mathbb{R}^k \to \mathbb{R}^d$ is the activation function. The formulation (1) covers "conventional" ResNets, which have the specific form

$$z_{l+1}(x) = z_l(x) + \frac{1}{ML} \sum_{m=1}^{M} U_{l,m} \sigma(W_{l,m}^\top z_l(x) + b_{l,m}), \quad l = 0, 1, \ldots, L-1,$$

where $W_{l,m}, U_{l,m} \in \mathbb{R}^d$, $b_{l,m} \in \mathbb{R}$, and $\sigma$ is the ReLU activation function. In this example, we have $\theta_{l,m} = (W_{l,m}, U_{l,m}, b_{l,m}) \in \mathbb{R}^k$, with $k = 2d + 1$.

Denote by $Z_{\Theta_{L,M}}(l; x)$ the output of the ResNet defined by (1). (This quantity is the same as $z_L(x)$ defined above, but we use this alternative notation to emphasize the dependece on parameters $\Theta_{L,M}$.) The goal of training ResNet is to seek parameters $\Theta_{L,M}$ that minimize the following mismatch or *loss* function:

$$E(\Theta_{L,M}) = \mathbb{E}_{x \sim \mu} \left[ \frac{1}{2} \left( g(Z_{\Theta_{L,M}}(L; x)) - y(x) \right)^2 \right], \tag{2}$$

where $g(x) : \mathbb{R}^d \to \mathbb{R}$ is a given measuring function, $y(x) \in \mathbb{R}$ is the label corresponding to $x$, and $\mu$ is the probability from which the data $x$ is drawn.

Classical gradient descent updates the parameters according to the formula

$$\Theta_{L,M}^{n+1} = \Theta_{L,M}^n - h\nabla_\Theta E(\Theta_{L,M}^n),$$

where $h$ is the step length. In the limit as $h \to 0$, the updating process can be characterized by the following ODE (Chizat & Bach, 2018, Def 2.2) (rescaled by $L, M$):

$$\frac{d\Theta_{L,M}(s)}{ds} = -ML\nabla_\Theta E(\Theta_{L,M}), \quad \text{for } s \geq 0, \tag{3}$$

where $s$ represents pseudo-time, the continuous analog of the discrete stepping process.

## 2.1 THE CONTINUOUS LIMIT AND THE MEAN-FIELD LIMIT

The *continuous limit* of (1) is obtained when the ResNet is infinitely deep, with $L \to \infty$. By reparametrizing the indices $l = [0, \cdots, L-1]$ with the continuous variable $t \in [0, 1]$, we can view $z$ in (1) as a function in $t$ that satisfies a coupled ODE, with $1/L$ being the stepsize in $t$. Accordingly, $\theta_{l,m}$ can be recast as $\theta_m(t = l/L)$, and denoting $\Theta(t) = \{\theta_m(t)\}_{m=1}^M$, we can write the continuous limit of (1) as

$$\frac{\mathrm{d}z(t;x)}{\mathrm{d}t} = \frac{1}{M} \sum_{m=1}^M f(z(t;x), \theta_m(t)), \quad t \in [0,1], \quad \text{with } z(0;x) = x. \tag{4}$$

Extending (2), we define the cost functional $E$ as

$$E(\Theta) = \mathbb{E}_{x \sim \mu} \left[ \frac{1}{2} \left( g(Z_\Theta(1;x)) - y(x) \right)^2 \right], \tag{5}$$

where $Z_\Theta(t;x)$ solves (4) for a given collection $\Theta(t)$ of the $M$ functions $\{\theta_m(t)\}$. Similar to (3), we can use GD to find the configuration of $\Theta(t)$ that minimizes (5) by making $\Theta(t)$ flow in the descending direction of $E(\Theta)$. Denote $s$ the pseudo-time of the training process, and $\Theta(s;t)$ the collection of functions at the training time $s$:

$$\frac{\partial \Theta}{\partial s} = -M \left. \frac{\delta E}{\delta \Theta} \right|_{\Theta(s;\cdot)}, \quad s > 0, \quad t \in [0,1] \tag{6}$$

where $\frac{\delta E}{\delta \Theta}$ is the functional derivative of $E$ with respect to $\Theta$, and thus a list of $M$ functions of $t$ for every fixed $s$.

The *mean-field limit* is obtained by making the ResNet infinitely wide, that is, $M \to \infty$. Considering that the right hand side of (4) has the form of an expectation, it approaches an integral in the limit, with respect to a certain probability density. Denoting this PDF by $\rho(\theta, t) \in \mathcal{C}([0,1]; \mathcal{P}^2)$[1], and assuming that the $\theta_m$ are drawn from it, the ODE for $z$ translates to the following:

$$\frac{\mathrm{d}z(t;x)}{\mathrm{d}t} = \int_{\mathbb{R}^k} f(z(t;x), \theta) \, \mathrm{d}\rho(\theta, t), \quad t \in [0,1] \quad \text{with } z(0;x) = x. \tag{7}$$

Mimicking (5), we define the following cost function in the mean-field setting:

$$E(\rho) = \mathbb{E}_{x \sim \mu} \left[ \frac{1}{2} \left( g(Z_\rho(1;x)) - y(x) \right)^2 \right], \tag{8}$$

where $Z_\rho(t;x)$ is the solution to (7) for a given $\rho$. Then, similar to the gradient flow for $\Theta_{L,M}$ and $\Theta(t)$, the probability distribution $\rho$ that encodes the configuration of $\theta$ flows in the descending direction of $E(\rho)$ in pseudo-time $s$. Since $\rho(\theta, t, s)$ needs to be a probability density for all $s$ and $t$, its evolution in $s$ is characterized by a gradient flow in the Wasserstein metric (Chizat & Bach, 2018; Lu et al., 2020; Ding et al., 2021):

$$\frac{\partial \rho}{\partial s} = \nabla_\theta \cdot \left( \rho \nabla_\theta \left. \frac{\delta E}{\delta \rho} \right|_{\rho(\cdot, \cdot, s)} \right), \quad s > 0, \ t \in [0,1] \quad \text{with} \quad \rho(\theta, t, 0) = \rho_{\mathrm{ini}}(\theta, t), \tag{9}$$

where $\frac{\delta E}{\delta \rho}$ is the functional derivative with respect to $\rho$, and thus a function of $(\theta, t)$ for every fixed $s$. Using the classical calculus-of-variations method, this functional derivative can be computed as:

$$\left. \frac{\delta E}{\delta \rho} \right|_\rho (\theta, t) = \mathbb{E}_{x \sim \mu} \left( p_\rho^\top(t;x) f(Z_\rho(t;x), \theta) \right), \tag{10}$$

where $p_\rho(\cdot; x)$, parameterized by $x$, maps $[0,1] \to \mathbb{R}^d$, and is a vector solution to the following ODE:

$$\frac{\mathrm{d}p_\rho^\top}{\mathrm{d}t} = -p_\rho^\top \int_{\mathbb{R}^k} \partial_z f(Z_\rho, \theta) \rho(\theta, t) \, \mathrm{d}\theta. \tag{11}$$

---

[1]A collection of probability distribution that is continuous in $t$ and has bounded second moment in $\theta$ for all $t$. The definition is to be made rigorous in Def 4.1.

with $p_\rho(t = 1; x) = (g(Z_\rho(1; x)) - y(x)) \nabla g(Z_\rho(1; x))$. In the later sections, to emphasize the $s$ dependence, we use $\frac{\delta E(\Theta(s))}{\delta \Theta}$ and $\frac{\delta E(\rho(s))}{\delta \rho}$ to denote $\frac{\delta E}{\delta \Theta}\big|_{\Theta(s;\cdot)}$ and $\frac{\delta E}{\delta \rho}\big|_{\rho(\cdot,\cdot,s)}$ respectively. As a summary, to update $\rho(\theta, t, s)$ to $\rho(\theta, t, s + \delta s)$ with an infinitesimal $\delta s$, we solve (7) for $Z_\rho(t; x)$, using the given $\rho(\theta, t, s)$, and compute $p_\rho$ using (11). This then allows us to compute $\frac{\delta E(\rho(s))}{\delta \rho}(\theta, t)$ which, in turn, yields $\rho(\theta, t, s + \delta s)$ from (9). In (11), $\partial_z f$ is a $d \times d$ matrix that stands for the Jacobian of $f$ with respect to its $z$ argument.

## 3 RELATED WORK AND CONTRIBUTION

There is a vast literature addressing the overparameterization of DNN. Many perspectives have been taken to justify the success of the application of the first order (gradient descent) optimization methods, in this overparameterized regime. We briefly review related works, and identify our contribution.

The earliest approach to understanding overparametrization was landscape analysis, in which the countours of the nonconvex objective function were studied to find which properties make it possible for a first order method to converge to the optimizer. Different NN structures are then analyzed to see which have these properties (Jin et al., 2017; Ge et al., 2015; Du et al., 2017; Ge et al., 2018; Nguyen & Hein, 2018; Du & Lee, 2018; Soltanolkotabi et al., 2019; Nguyen & Hein, 2017; Kawaguchi, 2016; Yun et al., 2018). This approach naturally limits the types of DNN that can be "explained," since most DNN structures do not satisfy the required properties.

Another approach taken in the literature is related to the Neural Tangent Kernel (NTK) regime, which is the regime in which the nonlinear problem is reduced to a nearly linear model due to the confinement of the iterates to a small region around the initial values. Insensitivity of the so-called Gram matrix is evaluated in the limit of the number of weights (Allen-Zhu et al., 2019; Du et al., 2019a; Zhang et al., 2019; Chatterji et al., 2021; Du et al., 2019b; Jacot et al., 2018; Liu et al., 2020; Frei et al., 2019). The argument is that zero-loss solutions are close to every point in the space, and one can find an optimal point within a small region of the initial guess. The NTK arguments are shown to work well in several real application problems, such as the classification problem (Li & Liang, 2018; Zou et al., 2019). However, as pointed out by (Ba et al., 2020; Wei et al., 2019; Fang et al., 2019), NTK approximately views nonlinear DNN as a linear kernel model, a rather limited description, so the estimates obtained through NTK might not be sharp. Indeed, the empirical observation in (Allen-Zhu & Li, 2019; Arora et al., 2019) have suggested that the kernel models are not as general as NN, and certain (nonlinear) features of NN are not captured.

Finally, there is the mean-field limit perspective that we adopt in this paper. The term "mean-field" indicates that in a system with a large ensemble of particles, the field formed by averaging across all samples exerts a force on each sample. Instead of tracing the trajectory of each sample, one can characterize the evolution of the full distribution function that represents the field. This idea originated in statistical physics, and is made rigorous under the framework of kinetic theory. In training an overparametrized ResNet context, a large number of weights evolve to decrease the cost function. In the mean-field limit, the training process evolves the distribution function of these weights. A significant advantage of the mean-field approach is that once we derive a formula for the gradient flow, standard PDE techniques can be adopted to describe the convergence behavior. This approach was taken in (Araújo et al., 2019; Fang et al., 2019; Nguyen, 2019; Du et al., 2019a; Chatterji et al., 2021; Chizat & Bach, 2018; Mei et al., 2018; Wojtowytsch, 2020; Lu et al., 2020; Sirignano & Spiliopoulos, 2021; 2020). The case of a single hidden layer NN in the regime as $M \to \infty$ is studied by Chizat & Bach (2018); Mei et al. (2018); Wojtowytsch (2020), who justified the mean-field approach and demonstrated convergence of the gradient flow process to a zero objective. In the multi-layer case, Lu et al. (2020) showed the convergence of a PDE that can be viewed as a modified version of the true gradient flow, hinting at convergence of the real mean-field limit. Nguyen & Pham (2021) also gave the global convergence of the mean-field limit of DNN for a certain class of NN structures, but their work excludes such important practical NN structures as ResNet. The work most closely related to ours is (Ding et al., 2021), but this paper makes technical assumptions on $\rho_\infty$ and $f$ that restrict the usefulness of the results, as we discuss below following the statement of Theorem 3.1.

We note that in certain parameter regimes, the mean-field and NTK perspectives can sometimes be unified; see (Chen et al., 2020).

We follow the roadmap of Chizat & Bach (2018); Ding et al. (2021), which shows that the PDE (9) achieves the global minimum for which $E(\rho(\theta, t, s = \infty)) = 0$, and that the gradient flow in the discrete setting (3) can be closely approximated by the PDE, so that $E(\Theta_{L,M}(s)) \approx E(\rho(\cdot, \cdot, s))$. These two results together show that $E(\Theta_{L,M}(s)) \approx 0$ for pseudo-time $s$ sufficiently large. Specifically, the two main tasks of the paper are as follows.

- Task 1: We need to give a rigorous proof of the continuous and mean-field limit. This will be stated in Theorem 5.1, to justify that for every fixed $s < \infty$, when $M, L \to \infty$, $E(\Theta_{L,M}(s)) \approx E(\Theta(s; \cdot)) \approx E(\rho(\cdot, \cdot, s))$. The dependence of these approximations on $L$ and $M$ are made precise.
- Task 2: We need to demonstrate the convergence to global minimum. This is stated in Theorem 6.1 and 6.2, for two different cases. In both theorems, we obtain the global convergence for the gradient flow, assuming certain homogeneity and the Sard-type regularity for $f$. A weak assumption of the initialization of $\rho_{\mathrm{ini}}$ is also imposed.

By combining these two, we obtain the main result of the paper.

**Theorem 3.1** *Let the conditions in Theorem 5.1 and 6.1 (or 6.2) hold. Then for any positive $\epsilon$ and $\eta$, there exist positive constants $C_0$ depending on $\rho_{\mathrm{ini}}(\theta, t), \epsilon$ and $C$ depending on $\rho_{\mathrm{ini}}(\theta, t), s$ such that when*

$$s > C_0(\rho_{\mathrm{ini}}(\theta, t), \epsilon), \quad M > \frac{C(\rho_{\mathrm{ini}}(\theta, t), s)}{\epsilon^2 \eta}, \quad L > \frac{C(\rho_{\mathrm{ini}}(\theta, t), s)}{\epsilon},$$

*we have*

$$\mathbb{P}\left(|E(\Theta_{L,M}(s))| \leq \epsilon\right) \geq 1 - \eta,$$

*where $E$ is defined in (2) and $\Theta_{L,M}$ solves (3).*

This theorem gives quantitative bounds for $M$ and $L$. The number of weights required to reduce the cost function below $\epsilon$ is $O(ML) = O(1/\epsilon^3)$. The theorem also suggests that $L$ and $M$ are independent parameters.

The results resonate with those obtained in (Chizat & Bach, 2018) for the 2-layer NN, and extend those in (Ding et al., 2021) greatly. Specifically, compared with the results in (Chizat & Bach, 2018), where $\rho(\theta, s)$ follows a typical gradient flow in the probability space on $\theta$ in time $s$ (Ambrosio et al., 2008), we have, at each training time $s$, a "list" of probability measures $\rho(\theta, t)$ on $\theta$, for all $t$. The members of this list are coupled, flowing together in $s$ in the descending direction of the cost function $E$. New analytical estimates are developed to deal with this non-traditional gradient flow.

Ding et al. (2021) takes a similar approach to ours, but their assumptions on the support of $\rho(\theta, t, s = \infty)$ are quite strong: The limiting probability measure $\rho(\theta, t, s = \infty)$ is assumed to have the full support over $\theta$. The assumption greatly reduces the technical difficulty of the proof, but it is impractical and hard to justify, thus preventing the results from being of practical use. In the current paper, this support condition is replaced by the well-accepted homogeneity condition adopted by (Chizat & Bach, 2018). As a consequence, the structure of the gradient flow must be examined closely to demonstrate convergence, requiring considerable technical complications. Lu et al. (2020) also investigates gradient flow for training multi-layer neural network, but the gradient flow structure is modified for mathematical convenience. All blocks are integrated together, making $\rho$ a probability measure over the full $(\theta, t)$-space. This design is inconsistent with the structure of the ResNet design that we investigate in this paper.

## 4 Notations, Assumptions, and Definitions

Throughout the paper we denote the collection of probability distributions with bounded second moments as $\mathcal{P}^2(\mathbb{R}^k)$, that is, $\mathcal{P}^2(\mathbb{R}^k) = \{\rho : \int_{\mathbb{R}^k} |\theta|^2 \, \mathrm{d}\rho(\theta) < \infty\}$. We assume certain regularity properties for the activation function $f$, the measuring function $g$, the data $y$, and the input measure $\mu$, as follows.

**Assumption 4.1 (Assumptions on $f$)** *Let $f : \mathbb{R}^d \times \mathbb{R}^k \to \mathbb{R}^d$ be a $\mathcal{C}^2$ function.*

1. *(linear growth) For all $x \in \mathbb{R}^d, \theta \in \mathbb{R}^k$, there is a constant $C_1$ such that*

$$|f| \leq C_1(|\theta|^2 + 1)(|x| + 1). \tag{12}$$

2. *(locally Lipschitz) For all $r > 0$, and $|x| < r, \theta \in \mathbb{R}^k$, we have for $C_2(r)$ monotonically increasing with respect to $r$ that the following bounds hold:*

$$|\partial_x f| \leq C_2(r)(|\theta|^2 + 1), \quad |\partial_\theta f| \leq C_2(r)(|\theta| + 1). \tag{13}$$

3. *(local smoothness) There exists $k_1 \in (0, k]$ with the following property: Denoting $\theta = (\theta_{[1]}, \theta_{[2]})$, $r = \max\{|x|, |\theta_{[1]}|\}$, where $\theta_{[1]} \in \mathbb{R}^{k_1}$, $\theta_{[2]} \in \mathbb{R}^{k-k_1}$, we have for $C_3(r)$ monotonically increasing with respect to $r$ that the following bounds hold:*

$$\left| \partial_x^i \partial_\theta^j f(x, \theta) \right| \leq C_3(r), \quad i + j = 2, \ i, j \geq 0. \tag{14}$$

*When $k_1 < k$, we have in addition that*

$$\max\{|\partial_x f|, |f|\} \leq C_3(r) \left( |\theta_{[2]}| + 1 \right), \ \left| \partial_{\theta_{[1]}} f \right| \leq C_3(|x|) \left( |\theta_{[1]}| + 1 \right). \tag{15}$$

4. *(universal kernel) The function set $\left\{ h \middle| h = \int_{\mathbb{R}^k} f(x, \theta) \, d\rho(\theta), \ \rho \in \mathcal{P}^2(\mathbb{R}^k) \right\}$ is dense in $\mathcal{C}\left( |x| < R; \mathbb{R}^d \right)$ for all $R > 0$.*

**Assumption 4.2 (Assumptions on data)** *Let $g, y : \mathbb{R}^d \to \mathbb{R}$ be $\mathcal{C}^2$ functions, and let $\mu$ be the probability distribution of $x$. We assume the following.*

5. *$\mu(x)$ is compactly supported, meaning that there is $R_\mu > 0$ such that the support of $\mu$ is within a ball of size $R_\mu$ around the origin, that is, $\text{supp}(\mu) \subset \mathcal{B}_{R_\mu}(\vec{0})$.*

6. *$y(x) \in L^\infty_{\text{loc}}(\mathbb{R}^d)$, that is, $\sup_{|x| \leq R} |y(x)| < \infty$.*

7. *$g(x)$ and $\nabla g(x)$ are Lipschitz continuous. Moreover, $|\nabla g(x)|$ has a positive lower bound, that is, $\inf_{x \in \mathbb{R}^d} |\nabla g(x)| > 0$.*

We note that Assumption 4.1 admits many commonly used activation functions (E et al., 2020). One example of a function satisfying this assumption is $f(x, \theta) = f(x, \theta_{[1]}, \theta_{[2]}, \theta_{[3]}) = \theta_{[3]} \sigma(\theta_{[1]} x + \theta_{[2]})$, where $\theta_{[1]} \in \mathbb{R}^{d \times d}$, $\theta_{[2]} \in \mathbb{R}^d$, $\theta_{[3]} \in \mathbb{R}$ and $\sigma$ is a component-wise regularized ReLU activation function, see Remark H.2.

We now build the metric on the function space for our solutions. Note that the solution $\rho(\theta, t, s)$ to (9) is expected to be a continuous function in $(t, s)$, and a distribution of $\theta$ for each $(t, s)$. For this non-standard probability space, we first introduce the following metric.

**Definition 4.1** [2] *$\mathcal{C}([0, 1]; \mathcal{P}^2)$ is a collection of continuous paths of probability distribution $\nu(\theta, t)$ $(\theta \in \mathbb{R}^k, t \in [0, 1])$ where 1. $\nu(\cdot, t) \in \mathcal{P}^2(\mathbb{R}^k)$ for every fixed $t \in [0, 1]$; 2. For any $t_0 \in [0, 1]$, $\lim_{t \to t_0} W_2(\nu(\cdot, t), \nu(\cdot, t_0)) = 0$, where $W_2$ is the classical Wasserstein-2 distance. The space $\mathcal{C}([0, 1]; \mathcal{P}^2)$ is equipped with the following metric $d_1(\nu_1, \nu_2) = \sup_t W_2(\nu_1(\cdot, t), \nu_2(\cdot, t))$.*

*Accordingly, $\mathcal{C}([0, \infty); \mathcal{C}([0, 1]; \mathcal{P}^2))$ is a collection of continuous paths of probability distribution $\nu(\theta, t, s)$ (with $\theta \in \mathbb{R}^k$, $t \in [0, 1]$, $s \in [0, \infty)$), where 1. $\nu(\cdot, \cdot, s) \in \mathcal{C}([0, 1]; \mathcal{P}^2)$ for every fixed $s \in [0, \infty)$. 2. For any $s_0 \in [0, \infty)$, $\lim_{s \to s_0} d_1(\nu(\cdot, \cdot, s), \nu(\cdot, \cdot, s_0)) = 0$ (where $d_1$ is defined above). The metric in $\mathcal{C}([0, \infty); \mathcal{C}([0, 1]; \mathcal{P}^2))$ is defined by $d_2(\nu_1, \nu_2) = \sup_{t,s} W_2(\nu_1(\cdot, t, s), \nu_2(\cdot, t, s))$.*

Since $\mathcal{P}^2$ is complete in $W_2$ distance, $\mathcal{C}([0, 1]; \mathcal{P}^2)$ and $\mathcal{C}([0, \infty); \mathcal{C}([0, 1]; \mathcal{P}^2))$ are complete metric spaces under $d_1$ and $d_2$ respectively also. To give a rigorous justification of the mean-field limit, we use the particle representation of $\rho(\theta, t, s)$. Thus, at least we need to assume that we can find a stochastic process that is drawn from the initial condition $\rho_{\text{ini}}(\theta, t)$. We call such initial conditions *admissible*.

**Definition 4.2** *We call a continuous path of probability distribution $\nu(\theta, t) \in \mathcal{C}([0, 1]; \mathcal{P}^2)$ admissible if it has a particle representation, namely there exists a continuous stochastic process $\theta(t) : [0, 1] \to \mathbb{R}^k$ and $r > 0$ such that for any $t_0 \in [0, 1]$, we have*

$$\theta(t_0) \sim \nu(\theta, t_0), \quad \lim_{t \to t_0} \mathbb{E}\left( |\theta(t) - \theta(t_0)|^2 \right) = 0, \quad |\theta(t_0)| < r. \tag{16}$$

---

[2] $\mathcal{C}([0, T]; \mathcal{A})$ is defined to be the set of functions $f(a, t)$ such that for any $t \in [0, T]$, $f(\cdot, t) \in \mathcal{A}$ and $f(\cdot, t)$ is continuous with respect to $t$ under the metric defined on $\mathcal{A}$. In Definition 4.1, we set $T = 1$ and $\mathcal{A} = \mathcal{P}^2$, where the natural metric on $\mathcal{A}$ is $W_2$.

*Furthermore, $\nu(\theta, t)$ is called* limit-admissible *if its $M$-averaged trajectory is bounded and Lipschitz with high probability. (See the rigorous definition in Definition E.1.)*

We note that without the dependence on $t$, probability distributions are "admissible", in the sense that one can draw a sample from a given distribution. In Appendix A, we show that if the initial condition $\rho_{\mathrm{ini}}(\theta, t)$ is admissible, (9) has a unique solution $\rho(\theta, t, s)$ that is admissible for each $s$.

The global convergence result depends on Sard-type regularity, defined as follows.

**Definition 4.3** *Given a metric space $\Theta$, a differentiable function $h : \Theta \to \mathbb{R}$, and a subset $\widetilde{\Theta} \subset \Theta$, we say $h$ satisfies Sard-type regularity in $\widetilde{\Theta}$ if the set of regular values[3] of $h|_{\widetilde{\Theta}}$ is dense in its range, where $h|_{\widetilde{\Theta}} : \widetilde{\Theta} \to \mathbb{R}$ is a confinement of $h$ in $\widetilde{\Theta}$.*

**Remark 4.1** *This regularity assumption is not a common one; we have adopted it from Chizat & Bach (2018). This property is essentially that most of the points in the range of $h$ lie in an open set and $h$ is locally monotonic. The assumption is rather mild and can be satisfied by most commonly seen regular functions, unless the function oscillates wildly.*

## 5 MEAN-FIELD AND CONTINUOUS LIMIT

In this section, we focus on the justification of mean-field and continuous-limit result. This is to prove that $E(\rho(\cdot, \cdot, s))$, $E(\Theta(s; \cdot))$, and $E(\Theta_{L,M}(s))$ are asymptotically close to each other for every $s$. In the next section, we prove convergence of $E(\rho(\cdot, \cdot, s))$ as $s \to \infty$.

To show the asymptotic equivalence of the three quantities, we need to compare (3), (6), and (9), and take the measurement in $E$ according to (2), (5) and (8).

**Theorem 5.1** *Suppose that Assumptions 4.1 and 4.2 are satisfied. Assume that $\rho_{\mathrm{ini}}(\theta, t)$ is limit-admissible and $\mathrm{supp}_{\theta}(\rho_{\mathrm{ini}}(\theta, t)) \subset \{\theta | |\theta_{[1]}| \le R\}$ with some $R > 0$ for all $t \in [0, 1]$. Let $\{\theta_m(0; t)\}_{m=1}^{M}$ in (6) be i.i.d. drawn from $\rho_{\mathrm{ini}}(\theta, t)$. Let*

- *$\Theta_{L,M}(s) = \{\theta_{l,m}(s)\}$ be the solution to (3) with initial condition $\theta_{l,m}(s = 0) = \theta_m\left(0; \frac{l}{L}\right)$;*

- *$\theta_m(s; t)$ be the solution to (6) with the initial condition $\theta_m(0; t)$;*

- *$\rho(\theta, t, s)$ be the solution to (9) with initial condition $\rho_{\mathrm{ini}}(\theta, t)$.*

*Then for any positive $\epsilon$, $\eta$, and $S$, there exists a constant $C > 0$ that depends on $\rho_{\mathrm{ini}}(\theta, t)$ and $S$ such that when*

$$M > \frac{C(\rho_{\mathrm{ini}}(\theta, t), S)}{\epsilon^2 \eta}, \quad L > \frac{C(\rho_{\mathrm{ini}}(\theta, t), S)}{\epsilon}, \quad s < S,$$

*we have:*

$$\min\{\mathbb{P}\left(|E(\Theta_{L,M}(s)) - E(\Theta(s; \cdot))| \le \epsilon/2\right), \mathbb{P}\left(|E(\Theta(s; \cdot)) - E(\rho(\cdot, \cdot, s))| \le \epsilon/2\right)\} \ge 1 - \tfrac{1}{2}\eta.$$

*It follows that*

$$\mathbb{P}\left(|E(\Theta_{L,M}(s)) - E(\rho(\cdot, \cdot, s))| \le \epsilon\right) \ge 1 - \eta, \quad \forall s < S.$$

*Here $E(\Theta_{L,M}(s))$, $E(\Theta(s; \cdot))$, and $E(\rho(\cdot, \cdot, s))$ are defined in (2), (5), and (8), respectively.*

The proof of this result appears in Appendix E. This theorem suggests that for every fixed $S > 0$, the gradient descent of $\Theta_{L,M}$ is approximately the same as the gradient flow of $\rho(\theta, t)$, in the sense that the two costs are close to each other with high probability, when $L$ and $M$ are sufficiently large. The size of the ResNet depends negative-algebraically on $\epsilon$ (the desired accuracy) and $\eta$ (the confidence of success). The result translates the evolution (gradient descent) of $\Theta_{L,M}$ to the evolution (gradient flow) of $\rho(\theta, t)$, and thus matches the zero-loss property of the parameter configuration of a finite sized ResNet to its limiting PDE, whose analysis can be performed with standard PDE tools.

---

[3]For a function $h : \widetilde{\Theta} \to \mathbb{R}$, a regular value is a real number $\alpha$ in the range of $h$ such that $h^{-1}(\alpha)$ is included in an open set where $h$ is differentiable and where $\mathrm{d}h$ does not vanish.

The proof of Theorem 5.1 divides naturally into two components. We show that for all $s < S$, $E(\rho(\cdot, \cdot, s)) \approx E(\Theta(s; \cdot))$ and $E(\Theta(s; \cdot)) \approx E(\Theta_{L,M}(s))$ with high probability. The former is obtained from mean-field limit theory, justifying that the particle trajectory $\theta_m(t, s)$ follows $\rho(\theta, t, s)$ for all $t$ in pseudo-time $s \in [0, S]$. The latter makes use of continuity in $t$ and traces the differences between $\theta_m(\frac{l}{L})$ and $\theta_{l,m}$. These two components of the proof are summarized in Theorems E.1 and E.2, respectively. According to the formula of the Fréchet derivatives (10) and (11), the estimates in these theorems naturally route through the boundedness of $p_\rho$, $p_\Theta$, $p_{\Theta_{L,M}}$, and similarly $Z_{\rho,\Theta,\Theta_{L,M}}$. It is technically demanding to derive these bounds, but they are not surprising. We dedicate a large portion of the appendix to addressing the well-posedness of these systems. See Appendices A-D, where we show these equations have unique solutions with proper initial conditions, along with the required bounds. Naturally, these estimates depend on regularity of $f$, $g$, $y$, and $\mu$.

To gain some intuition for the equivalence between (6) and (9), we test them on the same smooth function $h(\theta)$. To test (9), we multiply both sides by $h$ and perform integration by parts to obtain $\frac{d}{ds} \int_{\mathbb{R}^k} h \, d\rho(\theta) = -\int_{\mathbb{R}^k} \nabla_\theta h \nabla_\theta \frac{\delta E(\rho(s))}{\delta \rho} \, d\rho$. This is to say $\frac{d}{ds} \mathbb{E}(h) = \mathbb{E}\left(\nabla_\theta h \nabla_\theta \frac{\delta E(\rho(s))}{\delta \rho}\right)$. Testing $h$ on (6) also gives the same formula. Supposing that $\rho = \frac{1}{M} \sum_{m=1}^M \delta_{\theta_m}$, we have $\frac{d}{ds} \mathbb{E}(h) = \frac{1}{M} \sum_{m=1}^M \nabla_\theta h(\theta_m) \frac{d}{ds} \theta_m = -\sum_{m=1}^M \nabla_\theta h(\theta_m) \frac{\delta E}{\delta \theta_m}$. The right hand side is also $\mathbb{E}\left(\nabla_\theta h \nabla_\theta \frac{\delta E(\rho(s))}{\delta \rho}\right)$ if and only if $M \frac{\delta E(\Theta(s;\cdot))}{\delta \theta_m} = \nabla_\theta \frac{\delta E(\rho)}{\delta \rho}(\theta_m, t)$. This will be shown to hold true in Appendix F.

## 6 CONVERGENCE TO GLOBAL MINIMIZER

After translating the study of $\Theta_{L,M}$ to the study of $\rho(\theta, t)$, this section presents results on when and how $E(\rho)$ converges to zero loss by examining the conditions for the global convergence of (9). We first identify the property of global minimum.

**Proposition 6.1** *Suppose that $\rho \in \mathcal{C}([0, 1]; \mathcal{P}^2)$ has $E(\rho) > 0$. Then for any $t_0 \in [0, 1]$, there exists a measure $\nu(\theta)$ of $\mathbb{R}^k$ such that $\int_{\mathbb{R}^k} d\nu(\theta) = 0$ and*

$$\int_{\mathbb{R}^k} \left.\frac{\delta E}{\delta \rho}\right|_\rho (\theta, t_0) \, d\nu(\theta) < 0. \tag{17}$$

See Appendix H.1 for the proof of this result. At the stationary point of the cost function, $\frac{\delta E}{\delta \rho}(\theta, t_0) = 0$, then there is no $\nu$ satisfying (17), so $E(\rho)$ is necessarily trivial. Our task now becomes to check under what conditions on $f$ we can have $\frac{\delta E}{\delta \rho}$ becoming 0 as $s \to \infty$. We give two possibilities, both requiring a separation initialization and certain homogeneities. In the first case, we require $f$ to be 2-homogeneous, and the result is collected in Section 6.1. In the second case, we require $f$ to be partially 1-homogeneous; see Section 6.2.

### 6.1 THE 2-HOMOGENEOUS CASE

The results in this section are obtained under the following assumption on $f$. (Functions can be designed to satisfy this assumption easily; see Remark H.1.)

**Assumption 6.1** *The function $f(x, \theta) : \mathbb{R}^d \times \mathbb{R}^k$ is 2-homogeneous, meaning that $f(x, \lambda\theta) = \lambda^2 f(x, \theta)$ for all $(x, \theta, \lambda) \in \mathbb{R}^d \times \mathbb{R}^k \times \mathbb{R}$.*

**Theorem 6.1** *Let Assumption 6.1 and conditions of Theorem 5.1 hold true with $k_1 = k$. Let $\rho_\infty(\theta, t)$ be the long-time limit of (9), that is, $\rho(\theta, t, s)$ converges to $\rho_\infty(\theta, t)$ in $\mathcal{C}([0, 1]; \mathcal{P}^2)$ as $s \to \infty$. Then $E(\rho_\infty) = 0$ if the following hold:*

- *Separation initialization: There exists $t_0 \in [0, 1]$ such that $\rho_{\text{ini}}(\theta, t_0)$ separates[4] the spheres $r_a \mathbb{S}^{k-1}$ and $r_b \mathbb{S}^{k-1}$, for some $0 < r_a < r_b$.*

---

[4]We say that a set $C$ separates the sets $A$ and $B$ if any continuous path in with endpoints in $A$ and $B$ intersects $C$.

- *Sard-type regularity: $\left.\frac{\delta E}{\delta \rho}\right|_{\rho_\infty} (\theta, t_0)$ satisfies the Sard-type regularity condition in $\mathbb{S}^{k-1}$.*

A proof appears in Appendix H.2. A corollary of this theorem, when combined with Theorem 5.1 with $k_1 = k$, gives Theorem 3.1, the main result of our paper.

## 6.2 THE PARTIALLY 1-HOMOGENEOUS CASE

The following assumption is used in this section. (Functions that satisfy this assumption include regularized ReLU; see Remark H.2.)

**Assumption 6.2** *Let $\theta = (\theta_{[1]}, \theta_{[2]})$ with $\theta_{[1]} \in \mathbb{R}$ and $\theta_{[2]} \in \mathbb{R}^{k-1}$:*

1. *(partially 1-homogeneous in $\theta$) $f$ can be written as $f(x, \theta) = f(x, \theta_{[1]}, \theta_{[2]}) = \theta_{[1]} \widehat{f}(x, \theta_{[2]})$,*

2. *(locally bounded and smooth) For any $r > 0$, $\widehat{f}(x, \theta_{[2]})$ is bounded and Lipschitz with Lipschitz continuous differential for $(x, \theta_{[2]}) \in \mathcal{B}_r(\vec{0}) \times \mathbb{R}^{k-1}$.*

When Assumption 6.2 holds true, Assumption 4.1 part 3 can be satisfied with $k_1 = 1$. Then, we introduce the main result in this section.:

**Theorem 6.2** *Let Assumption 6.2 and conditions of Theorem 5.1 hold true with $k_1 = 1$. Let $\rho_\infty(\theta, t)$ be the limit of (9) as $s \to \infty$. Then $E(\rho_\infty) = 0$ if the following hold:*

- *Separation initialization: There exists $t_0 \in [0, 1]$ such that $\rho_{\mathrm{ini}}(\theta_{[1]}, \theta_{[2]}, t_0)$ separates the spheres $\{-r_0\} \times \mathbb{R}^{k-1}$ and $\{r_0\} \times \mathbb{R}^{k-1}$ for some $r_0 > 0$, where $\theta_{[1]}, \theta_{[2]}$ are defined by Assumption 4.1 item 3 with $k_1 = 1$.*

- *Sard-type regularity: $\left.\frac{\delta E}{\delta \rho}\right|_{\rho_\infty} ((1, \theta_{[2]}), t_0) : \mathbb{R}^{k-1} \to \mathbb{R}$ satisfies the Sard-type regularity condition.*

- *For any $\tilde{\rho} \in \mathcal{C}([0,1], \mathcal{P}^2)$, define $H_{r,\tilde{\rho}}\left(\tilde{\theta}_{[2]}\right) = \left.\frac{\delta E(\rho)}{\delta \rho}\right|_{\tilde{\rho}} \left(1, r\tilde{\theta}_{[2]}, t_0\right)$ where $\theta_{[2]} = r\tilde{\theta}_{[2]}$ with $r = |\theta_2|$ and $\tilde{\theta}_{[2]} \in \mathbb{S}^{k-2}$. Suppose that $H_{r,\tilde{\rho}}$ converges in $\mathcal{C}^1(\mathbb{S}^{k-2})$ as $r \to \infty$ to a function $H_{\infty,\tilde{\rho}}$. Furthermore, assume that $H_{\infty,\rho_\infty}$ satisfies the Sard-type regularity condition in $\mathbb{S}^{k-2}$ and that the intersection of regular values of $H_{\infty,\rho_\infty}$ and $\left.\frac{\delta E(\rho)}{\delta \rho}\right|_{\rho_\infty} (1, \theta_{[2]}, t_0)$ is also dense in the intersection of their range.*

The proof is found in Appendix H.3. This result, combined with Theorem 5.1 in the case of $k_1 = 1$, gives the main result in Theorem 3.1. The assumptions in Theorem 6.2 are rather technical and seemingly tedious. However, we note that only the first two assumptions — 1-homogeneous and the separation assumption — are crucial. The third and fourth assumptions concern the regularity of the Fréchet derivative of $\rho$; they are rather mild and serve to exclude wildly oscillating functions; see Remark 4.1. Most commonly seen functions are regular enough that these two assumptions are satisfied.

## 7 ETHICS STATEMENT

This work does not present any foreseeable societal consequence.

## 8 REPRODUCIBILITY STATEMENT

The notations, assumptions and definitions are clarified in Section 4. And all proofs appear in the Appendix.

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

APPENDIX: INTRODUCTION

The appendix contains proofs and supporting analysis for the theorems in the main text. We start by proving well-posedness of the ResNet ODE and the gradient flow. We then justify the continuous and mean-field limit. Finally, we prove the global convergence of the gradient-flow PDE. The sections are organized as follows.

Appendix A: Well-posedness. We summarize the well-posedness of the continuous limit ODE (4), the mean-field limit ODE (7), and the associated gradient flows (3), (6), and (9). The results are collected in Theorem A.1.

Appendix B: This section collects the detailed proof for the well-posedness of the ResNet ODE in its continuous limit (4), and the mean-field limit (7), and finalizes the proof of Theorem A.2.

Appendix C: This section prepares some a-priori estimates for showing the well-posedness of the gradient flow equations.

Appendix D: This section contains a detailed proof for the well-posedness of gradient flow equations (3) and (9) and finalizes the proofs of Theorem A.3 and A.4.

Appendix E-G: Proof of Theorem 5.1, the continuous and mean-field limit. Section E lays out the structure of the proof, Section F shows the continuous limit, and Section G shows the mean-field limit.

Appendix H: Proof of Theorem 6.1 and 6.2: Global convergence of the gradient flow.

The analytical core of the paper lies in Appendices H and E, which describe properties of the gradient flow PDEs and explain why the gradient descent method for ResNet can be explained by these equations. The technical results of Appendix B-D can be skipped by readers who are interested to proofs of the main results.

Throughout, we denote by $C(\cdot)$ a generic constant whose value depends only on its arguments. The precise value of this constant may change each time it is invoked. Throughout Appendices A-G, we assume that Assumption 4.1 holds for some $0 < k_1 \leq k$.

## A  WELL-POSEDNESS RESULT

In this section, we show the well-posedness of ODEs (4), (7) and gradient flows (3), (6), and (9).

**Theorem A.1** *The following claims hold.*

– *Well-posedness of ODE:*

  • *If $\{\theta_m(t)\}_{m=1}^M$ is continuous, then (4) has a unique $\mathcal{C}^1$ solution.*
  • *If $\rho \in \mathcal{C}([0,1]; \mathcal{P}^2)$, then (7) has a unique $\mathcal{C}^1$ solution.*

– *Well-posedness of gradient flow:*

  • *(3) has a unique solution.*
  • *If $\{\theta_m(0;t)\}_{m=1}^M$ is continuous, then (6) has a unique solution $\{\theta_m(s;t)\}_{m=1}^M$ that is continuous in $(s,t)$.*
  • *If $\rho_{\mathrm{ini}}(\theta,t)$ is admissible and $\mathrm{supp}_\theta(\rho_{\mathrm{ini}}(\theta,t)) \subset \{\theta | |\theta_{[1]}| \leq R\}$ with some $R > 0$ for all $t \in [0,1]$, then (9) has a unique solution $\rho(\theta,t,s)$ in $\mathcal{C}([0,\infty); \mathcal{C}([0,1]; \mathcal{P}^2))$ with initial condition $\rho_{\mathrm{ini}}(\theta,t)$. Furthermore, for each $s$, $\rho(\theta,t,s)$ is admissible.*

Note that the well-posedness of (4) and (6) are direct corollaries of that of (7), (3), and (9) (the corresponding continuous versions), according to Remark A.1, A.2. Thus, we merely prove well-posedness on the continuous level, in Theorems A.2, A.3, and A.4. Specifically,

  • Theorem A.2 (Appendix A.1) shows the well-posedness of the dynamical system for $z$;
  • Theorems A.3 and A.4 (Appendix A.2) justify the well-posedness of the gradient flow of the parameter configuration.

## A.1 Well-posedness of the OID (7)

As $L$ and $M$ approach $\infty$, $z$ satisfies the ordinary-integral equation (7). We justify that this differential equation is well-posed, in the sense that the solution is unique and stable.

**Theorem A.2** *Suppose that Assumption 4.1 holds with and that $x$ is in the support of $\mu$, then (7) has a unique $\mathcal{C}^1$ solution. Specifically, for $\rho_1, \rho_2 \in \mathcal{C}([0,1]; \mathcal{P}^2)$, we have*

$$|Z_{\rho_1}(t; x)| \leq C(\mathcal{L}_1), \tag{18}$$

*and for all $t \in [0,1]$:*

$$|Z_{\rho_1}(t; x) - Z_{\rho_2}(t; x)| \leq C(\mathcal{L}_1, \mathcal{L}_2) d_1(\rho_1, \rho_2), \tag{19}$$

*where $\mathcal{L}_i$ are the second moments of $\rho_i$, that is,*

$$\mathcal{L}_i = \int_0^1 \int_{\mathbb{R}^k} |\theta|^2 d\rho_i(\theta, t) \, \mathrm{d}t, \quad i = 1, 2. \tag{20}$$

We leave the proof to Appendix B. Besides the well-posedness result, the theorem also suggests that a small perturbation to $\rho$ is reflected linearly in $Z_\rho$, the solution to (7). This means that a small perturbation in the parameterization of the ResNet leads to only a small perturbation to the ResNet output.

**Remark A.1** *Although we do not directly show the well-posedness of (4), it follow immediately from Theorem A.2. One way to make this connection is to reformulate the discrete probability distribution as*

$$\rho^{\mathrm{dis}}(\theta, t) = \frac{1}{M} \sum_{m=1}^{M} \delta_{\theta_m(t)}(\theta),$$

*where $\Theta(t) = \{\theta_m(t)\}_{m=1}^{M}$ is the list of trajectories. Since $\theta_m(t)$ is continuous in $t$, we have $\rho^{\mathrm{dis}}(\theta, t) \in \mathcal{C}([0,1]; \mathcal{P}^2)$. Since*

$$\frac{1}{M} \sum_{m=1}^{M} f(z(t; x), \theta_m(t)) = \int_{\mathbb{R}^k} f(z(t; x), \theta) \, \mathrm{d}\rho^{\mathrm{dis}}(\theta, t),$$

*using Theorem A.2, (4) has a unique $\mathcal{C}^1$ solution when $\Theta(t)$ is continuous.*

## A.2 Well-posedness of the gradient flow

The gradient flow of the parameterization is also well-posed, both in the discrete setting and the continuous mean-field limit. In the discrete setting, we have the following result.

**Theorem A.3** *(3) has a unique solution.*

Further, (9) characterizes the dynamics of the continuous mean-field limit of the parameter configuration, and is also well-posed.

**Theorem A.4** *If $\rho_{\mathrm{ini}}(\theta, t)$ is admissible and $\mathrm{supp}_\theta(\rho_{\mathrm{ini}}(\theta, t)) \subset \{\theta | \|\theta_{[1]}| \leq R\}$ with some $R > 0$ for all $t \in [0,1]$. Then (9) has a unique solution $\rho(\theta, t, s)$ in $\mathcal{C}([0, \infty); \mathcal{C}([0,1]; \mathcal{P}^2))$ that is admissible for each $s$. Further, we have*

$$\frac{\mathrm{d}E(\rho(\cdot, \cdot, s))}{\mathrm{d}s} \leq 0. \tag{21}$$

The proofs of these two theorems can be found in Appendix D.

**Remark A.2** *Using the same argument as in Remark A.1, calling*

$$\rho^{\mathrm{dis}}(\theta, t, s) = \frac{1}{M} \sum_{m=1}^{M} \delta_{\theta_m(s;t)}(\theta), \tag{22}$$

*the well-posedness of (6) follows immediately from Theorem A.4. Furthermore, according to the definition (5) and (8), we have*

$$E\left(\rho^{\mathrm{dis}}(\cdot, \cdot, s)\right) = E\left(\Theta(s; \cdot)\right).$$

*As a consequence, if $\Theta(s; t)$ satisfies (6), then $\rho^{\mathrm{dis}}$ satisfies (9), and vice versa. The well-posedness result in Theorem A.4 for (9) then can be extended to justify well-posedness of (6).*

# B    PROOF OF THEOREM A.2

This section contains the proof of Theorem A.2. We rewrite (7) as follows:

$$\frac{\mathrm{d}Z_\rho(t;x)}{\mathrm{d}t} = F(Z_\rho, t), \quad \forall t \in [0,1] \quad \text{with} \quad z(0;x) = x, \tag{23}$$

where for a given $\rho \in \mathcal{C}([0,1]; \mathcal{P}^2)$ we use the notation:

$$F(z,t) = \int_{\mathbb{R}^k} f(z,\theta) \, \mathrm{d}\rho(\theta, t).$$

The proof of Theorem A.2 relies on the classical Lipschitz condition for the well-posedness of an ODE.

**Proof** [Proof of Theorem A.2] Since $\rho \in \mathcal{C}([0,1]; \mathcal{P}^2)$, we have a constant $C$ such that

$$\sup_{0 \le t \le 1} \int_{\mathbb{R}^k} |\theta|^2 \, \mathrm{d}\rho(\theta, t) < C < \infty.$$

For any $t \in [0,1]$, using (12) from Assumption 4.1 equation, we have

$$|F(z,t)| \le \int_{\mathbb{R}^k} |f(z_1, \theta)| \, \mathrm{d}\rho(\theta, t) \le C_1(|z| + 1) \int_{\mathbb{R}^k} (|\theta|^2 + 1) \, \mathrm{d}\rho(\theta, t). \tag{24}$$

To show the boundedness result (18), we multiply (23) by $Z_{\rho_1}(t;x)$ and use (24) to obtain

$$\frac{\mathrm{d}|Z_{\rho_1}(t;x)|^2}{\mathrm{d}t} \le 2C_1 \left(|Z_{\rho_1}|^2 + |Z_{\rho_1}|\right) \int_{\mathbb{R}^k} (|\theta|^2 + 1) \, \mathrm{d}\rho_1(\theta, t)$$

$$\le 4C_1 \int_{\mathbb{R}^k} (|\theta|^2 + 1) \, \mathrm{d}\rho_1(\theta, t) \left(|Z_{\rho_1}(t;x)|^2 + 1\right).$$

Using Grönwall's inequality, we have

$$|Z_{\rho_1}(t;x)| \le \exp\left(2C_1 \left(\int_0^1 \int_{\mathbb{R}^k} |\theta|^2 \, \mathrm{d}\rho_1 \, \mathrm{d}t + 1\right)\right) (|x| + 1)$$

$$\le \exp\left(2C_1 \left(\mathcal{L}_1 + 1\right)\right) (|x| + 1),$$

where $\mathcal{L}_1 = \int_0^1 \int_{\mathbb{R}^k} |\theta|^2 \, \mathrm{d}\rho_1 \, \mathrm{d}t$. Since $x \in \operatorname{supp} \mu$ (so that $|x| < R$), we have (18). This gives us an a-priori estimate of (23).

Next, using (13) from Assumption 4.1, we have

$$|f(z_1, \theta) - f(z_2, \theta)| \le C_2(|z_1| + |z_2|) \left(|\theta|^2 + 1\right) |z_1 - z_2|. \tag{25}$$

Then, using boundedness of the second moment of $\theta$, we have

$$|F(z_1, t) - F(z_2, t)| \le \left| \int_{\mathbb{R}^k} (f(z_1, \theta) - f(z_2, \theta)) \, \mathrm{d}\rho(\theta, t) \right|$$

$$\le C_2(|z_1| + |z_2|) \int_{\mathbb{R}^k} (|\theta|^2 + 1) \, \mathrm{d}\rho(\theta, t) |z_1 - z_2|$$

$$< C_2(|z_1| + |z_2|)(C + 1)|z_1 - z_2|,$$

which implies that $F(z, t)$ is locally Lipschitz in $z$ for all $t \in [0, 1]$. Combining this with the a-priori estimate in (18), classical ODE theory implies that (23) has a unique $\mathcal{C}^1$ solution.

To prove the stability result (19), we define $Z_{\rho_i}$ as in (23) parameterized by $\rho_i \in \mathcal{C}([0,1]; \mathcal{P}^2)$, and denote

$$\Delta(t;x) = Z_{\rho_1}(t;x) - Z_{\rho_2}(t;x).$$

Then by subtracting the two equations, we obtain

$$
\frac{\mathrm{d}|\Delta(t;x)|^2}{\mathrm{d}t}
$$

$$
= 2\left\langle \Delta(t;x), \int_{\mathbb{R}^k} f(Z_{\rho_1}(t;x),\theta)\,\mathrm{d}\rho_1(\theta,t) - \int_{\mathbb{R}^k} f(Z_{\rho_2}(t;x),\theta)\,\mathrm{d}\rho_2(\theta,t) \right\rangle
$$

$$
= 2\left\langle \Delta(t;x), \underbrace{\int_{\mathbb{R}^k} f(Z_{\rho_1}(t;x),\theta)\,\mathrm{d}\rho_1(\theta,t) - \int_{\mathbb{R}^k} f(Z_{\rho_2}(t;x),\theta)\,\mathrm{d}\rho_1(\theta,t)}_{(\mathrm{I})} \right\rangle \tag{26}
$$

$$
+ 2\left\langle \Delta(t;x), \underbrace{\int_{\mathbb{R}^k} f(Z_{\rho_2}(t;x),\theta)\,\mathrm{d}\rho_1(\theta,t) - \int_{\mathbb{R}^k} f(Z_{\rho_2}(t;x),\theta)\,\mathrm{d}\rho_2(\theta,t)}_{(\mathrm{II})} \right\rangle .
$$

We now bound (I) and (II). For (I), we have using (13) and (25) that

$$
|(\mathrm{I})| \le 2C_2\left(|Z_{\rho_1}(t;x)| + |Z_{\rho_2}(t;x)|\right)|\Delta(t;x)| \int_{\mathbb{R}^k}(|\theta|^2 + 1)\,\mathrm{d}\rho_1(\theta,t)
$$

$$
\le C(\mathcal{L}_1,\mathcal{L}_2)|\Delta(t;x)| \int_{\mathbb{R}^k}(|\theta|^2 + 1)\,\mathrm{d}\rho_1(\theta,t), \tag{27}
$$

where the second inequality comes from (18). For (II), we denote the particle representation $\theta_1(t) \sim \rho_1(\theta,t)$ and $\theta_2(t) \sim \rho_2(\theta,t)$ such that $\left(\mathbb{E}\left(|\theta_1 - \theta_2|^2\right)\right)^{1/2} = W_2(\rho_1(\cdot,t),\rho_2(\cdot,t))$. Then

$$
\begin{aligned}
|(\mathrm{II})| &\le \mathbb{E}\left(|f(Z_{\rho_2}(t;x),\theta_1) - f(Z_{\rho_2}(t;x),\theta_2)|\right) \\
&\le C_2(Z_{\rho_2}(t;x))\mathbb{E}\left((|\theta_1| + |\theta_2| + 1)|\theta_1 - \theta_2|\right) \\
&\le C(\mathcal{L}_2)\mathbb{E}\left((|\theta_1| + |\theta_2| + 1)|\theta_1 - \theta_2|\right) \\
&\le C(\mathcal{L}_2)\left(\mathbb{E}\left(|\theta_1|^2 + |\theta_2|^2 + 1\right)\right)^{1/2}\left(\mathbb{E}\left(|\theta_1 - \theta_2|^2\right)\right)^{1/2} \\
&\le C(\mathcal{L}_2)\left(\mathbb{E}\left(|\theta_1|^2 + |\theta_2|^2 + 1\right)\right)^{1/2}W_2(\rho_1(\cdot,t),\rho_2(\cdot,t)) \\
&\le C(\mathcal{L}_2)\left(\int_{\mathbb{R}^k}|\theta|^2 d\rho_1(\theta,t) + \int_{\mathbb{R}^k}|\theta|^2 d\rho_2(\theta,t) + 1\right)^{1/2} d_1(\rho_1,\rho_2),
\end{aligned} \tag{28}
$$

where we use mean-value theorem and (13) in the first inequality to obtain

$$
\begin{aligned}
|f(Z_{\rho_2}(t;x),\theta_1) - f(Z_{\rho_2}(t;x),\theta_2)| &\le |\partial_\theta f(Z_{\rho_2}(t;x),\theta_1 + \lambda\theta_2)||\theta_1 - \theta_2| \\
&\le C_2(Z_{\rho_2}(t;x))(|\theta_1| + |\theta_2| + 1)|\theta_1 - \theta_2|,
\end{aligned}
$$

for some $\lambda \in [0,1]$. We use (18) in the second inequality of (28).

Plugging (27)-(28) into (26) and using Hölder's inequality, we obtain that

$$
\begin{aligned}
\frac{\mathrm{d}|\Delta(t;x)|^2}{\mathrm{d}t} \le\, & C(\mathcal{L}_1,\mathcal{L}_2)|\Delta(t;x)|^2 \int_{\mathbb{R}^k}|\theta|^2\,\mathrm{d}\rho_1(\theta,t) \\
& + C(\mathcal{L}_2)|\Delta(t;x)|\left(\int_{\mathbb{R}^k}|\theta|^2 d\rho_1(\theta,t) + \int_{\mathbb{R}^k}|\theta|^2 d\rho_2(\theta,t) + 1\right)^{1/2} d_1(\rho_1,\rho_2) \\
\le\, & C(\mathcal{L}_1,\mathcal{L}_2)\left(\int_{\mathbb{R}^k}|\theta|^2 d\rho_1(\theta,t) + \int_{\mathbb{R}^k}|\theta|^2 d\rho_2(\theta,t) + 1\right)\left(|\Delta(t;x)|^2 + d_1^2(\rho_1,\rho_2)\right),
\end{aligned}
$$

where we used Young's inequality in the last line. Since $|\Delta(0;x)| = 0$, we complete the proof of (19) using Grönwall's inequality. ∎

## C A-PRIORI ESTIMATION OF THE COST FUNCTION

Some a-priori estimates are necessary in the proof for the main theorems. We first consider the case when $f$ satisfies only Assumption 4.1 with $0 < k_1 \le k$. (Better a-priori estimates can be obtained when $f$ also satisfies the homogeneity properties of Sections 6.1 or 6.2.)

## C.1    A-PRIORI ESTIMATE FOR GENERAL $f$

According to (10), the Fréchet derivative can be computed, similarly to (Lu et al., 2020), as follows:

$$\frac{\delta E(\rho)}{\delta \rho}(\theta, t) = \mathbb{E}_{x \sim \mu} \left( p_\rho^\top(t; x) f(Z_\rho(t; x), \theta) \right), \tag{29}$$

where $p_\rho(t; x)$ is the solution to the following ODE:

$$\begin{cases} \dfrac{\partial p_\rho^\top}{\partial t} = -p_\rho^\top \displaystyle\int_{\mathbb{R}^k} \partial_z f(Z_\rho(t; x), \theta) \, \mathrm{d}\rho(\theta, t), \\[4mm] p_\rho(1; x) = (g(Z_\rho(1; x)) - y(x)) \nabla g(Z_\rho(1; x)). \end{cases} \tag{30}$$

We now show that $p_\rho$ is Lipschitz continuous with respect to $\rho$.

**Lemma C.1** *Suppose that $x$ is in the support of $\mu$. Suppose that $\rho_1, \rho_2 \in \mathcal{C}([0, 1]; \mathcal{P}^2)$ and $p_{\rho_1}$, $p_{\rho_2}$ are the corresponding solutions of* (30). *Denote $\mathcal{L}_1$ and $\mathcal{L}_2$ as in* (20) *and*

$$\mathcal{R} = \min_r \left\{ r \big| \mathrm{supp}(\rho_1) \cup \mathrm{supp}(\rho_2) \subset \left\{ \theta \big| |\theta_{[1]}| < r \right\} \right\}.$$

*Then the following two bounds are satisfied:*

$$|p_{\rho_1}(t; x)| \le C(\mathcal{L}_1), \tag{31}$$

*and*

$$|p_{\rho_1}(t; x) - p_{\rho_2}(t; x)| \le C(\mathcal{R}, \mathcal{L}) \, d_1(\rho_1, \rho_2), \tag{32}$$

*where $\mathcal{L} = \max\{\mathcal{L}_1, \mathcal{L}_2\}$.*

**Proof** From (13) in Assumption 4.1,

$$\begin{aligned} \left| \int_{\mathbb{R}^k} \partial_z f(Z_{\rho_1}(t; x), \theta) \, \mathrm{d}\rho_1(\theta, t) \right| &\le C(Z_{\rho_1}(t; x)) \int_{\mathbb{R}^k} (|\theta|^2 + 1) \, \mathrm{d}\rho_1(\theta, t) \\ &\le C(\mathcal{L}_1) \int_{\mathbb{R}^k} (|\theta|^2 + 1) \, \mathrm{d}\rho_1(\theta, t), \end{aligned} \tag{33}$$

where we use (18) in the second inequality. It follows from the initial condition of (30) that

$$p_{\rho_1}(1; x) \le C(|Z_{\rho_1}(1, x)| + 1) \le C(\mathcal{L}_1),$$

where we use Assumption 4.1 in the first inequality and (18) in the second inequality. Noting that (30) is a linear equation, (31) follows naturally when we combine (33) with the inequality above.

To prove (32), we define

$$\Delta(t; x) = p_{\rho_1}(t; x) - p_{\rho_2}(t; x).$$

For $t = 1$, with $x \in \mathrm{supp}\, \mu$, we have

$$\begin{aligned} |\Delta(1; x)| &= |p_{\rho_1}(1; x) - p_{\rho_2}(1; x)| \\ &= |(g(Z_{\rho_1}(1; x)) - y(x)) \nabla g(Z_{\rho_1}(1; x)) - (g(Z_{\rho_2}(1; x)) - y(x)) \nabla g(Z_{\rho_2}(1; x))| \\ &\le C(\mathcal{L}) |Z_{\rho_1}(1; x) - Z_{\rho_2}(1; x)| \\ &\le C(\mathcal{L}) d_1(\rho_1, \rho_2), \end{aligned} \tag{34}$$

where we use Assumption 4.1, (18), and $|x| < R_\mu$ in the first inequality and (19) in the second inequality. The following ODE is satisfied by $\Delta$:

$$\frac{\partial \Delta^\top(t; x)}{\partial t} = -\Delta^\top(t; x) \int_{\mathbb{R}^k} \partial_z f(Z_{\rho_1}(t; x), \theta) \, \mathrm{d}\rho_1(\theta, t) + p_{\rho_2}^\top(t; x) D_{\rho_1, \rho_2}(t; x), \tag{35}$$

where

$$D_{\rho_1, \rho_2}(t; x) = \int_{\mathbb{R}^k} \partial_z f(Z_{\rho_2}(t; x), \theta) \, \mathrm{d}\rho_2(\theta, t) - \int_{\mathbb{R}^k} \partial_z f(Z_{\rho_1}(t; x), \theta) \, \mathrm{d}\rho_1(\theta, t). \tag{36}$$

To show the boundedness of $D_{\rho_1,\rho_2}(t;x)$, we follow the same strategy as that for (26). By splitting into two terms, we obtain

$$|D_{\rho_1,\rho_2}(t;x)| \leq \underbrace{\left| \int_{\mathbb{R}^k} \partial_z f(Z_{\rho_2}(t;x),\theta)\,\mathrm{d}\rho_2(\theta,t) - \int_{\mathbb{R}^k} \partial_z f(Z_{\rho_2}(t;x),\theta)\,\mathrm{d}\rho_1(\theta,t) \right|}_{\text{(I)}}$$

$$+ \underbrace{\left| \int_{\mathbb{R}^k} \partial_z f(Z_{\rho_2}(t;x),\theta)\,\mathrm{d}\rho_1(\theta,t) - \int_{\mathbb{R}^k} \partial_z f(Z_{\rho_1}(t;x),\theta)\,\mathrm{d}\rho_1(\theta,t) \right|}_{\text{(II)}}. \tag{37}$$

The bound of (II) relies on part 3 of Assumption 4.1: Because the supports of $\rho_1$, $\rho_2$ are contained in $\{\theta\big||\theta_{[1]}| < \mathcal{R}\}$ and $Z$ is bounded by (18), we have

$$|(\mathrm{II})| \leq C(\mathcal{R},\mathcal{L})\,|Z_{\rho_1}(t;x) - Z_{\rho_2}(t;x)| \leq C(\mathcal{R},\mathcal{L})d_1(\rho_1,\rho_2)\,, \tag{38}$$

where we use (19) in the second inequality.

To bound (I), we use the particle representation $\theta_1 \sim \rho_1(\theta,t)$ and $\theta_2 \sim \rho_2(\theta,t)$ such that $\left(\mathbb{E}\left(|\theta_1 - \theta_2|^2\right)\right)^{1/2} = W_2(\rho_1(\cdot,t),\rho_2(\cdot,t))$. We then have

$$\begin{aligned}
(\mathrm{I}) &\leq \mathbb{E}\left(|\partial_z f(Z_{\rho_2}(t;x),\theta_1) - \partial_z f(Z_{\rho_2}(t;x),\theta_2)|\right) \\
&\leq C(\mathcal{R},\mathcal{L})\mathbb{E}\left(|\theta_1 - \theta_2|\right) \\
&\leq C(\mathcal{R},\mathcal{L})\left(\mathbb{E}\left(|\theta_1 - \theta_2|^2\right)\right)^{1/2} \\
&\leq C(\mathcal{R},\mathcal{L})d_1(\rho_1,\rho_2),
\end{aligned} \tag{39}$$

where we use the mean-value theorem, part 3 of Assumption 4.1 with $|\theta_{1,[1]}| < \mathcal{R}$ and $|\theta_{2,[1]}| < \mathcal{R}$, (18) in the second inequality. Substituting (38) and (39) into (37), we obtain

$$|D_{\rho_1,\rho_2}(t;x)| \leq C(\mathcal{R},\mathcal{L})d_1(\rho_1,\rho_2)\,.$$

By substituting this bound into (35) and using (33), we have

$$\frac{\mathrm{d}|\Delta(t;x)|^2}{\mathrm{d}t} \leq C(\mathcal{R},\mathcal{L})\left(|\Delta(t;x)|^2 + d_1^2(\rho_1,\rho_2)\right)\,. \tag{40}$$

The result (32) follows from the initial condition (34) and Grönwall's inequality. ∎

The second lemma concerns the continuity of $\nabla_\theta \frac{\delta E(\rho)}{\delta\rho}$.

**Lemma C.2** *Suppose that $\rho,\rho_1,\rho_2 \in \mathcal{C}([0,1];\mathcal{P}^2)$. Define*

$$\mathcal{L} = \max_{1\leq i\leq 3} \int_0^1 \int_{\mathbb{R}^k} |\theta|^2\,\mathrm{d}\rho_i(\theta,t)\,\mathrm{d}t, \quad \mathcal{L}^{\mathrm{sup}} = \max_{1\leq i\leq 3} \sup_{t\in[0,1]} \int_{\mathbb{R}^k} |\theta|^2\,\mathrm{d}\rho_i(\theta,t)\,,$$

*and*

$$\mathcal{R} = \min_r \left\{ r\big|\mathrm{supp}(\rho) \cup \mathrm{supp}(\rho_1) \cup \mathrm{supp}(\rho_2) \subset \{\theta\big||\theta_{[1]}| < r\} \right\}\,.$$

*Then for any $(\theta,t),(\theta_1,t_1),(\theta_2,t_2) \in \mathbb{R}^k \times [0,1]$ and $s > 0$, the following properties hold.*

- *Boundedness:*
$$\left|\nabla_\theta \frac{\delta E(\rho)}{\delta\rho}(\theta,t)\right| \leq C(\mathcal{L})(|\theta|+1), \quad \left|\partial_{\theta_{[1]}} \frac{\delta E(\rho)}{\delta\rho}(\theta,t)\right| \leq C(\mathcal{L})(|\theta_{[1]}|+1) \tag{41}$$

- *Lipschitz continuity in $\theta$ and $t$: There exists $Q_1 : \mathbb{R}^2 \to \mathbb{R}^+$ that depends increasingly on both arguments such that*
$$\left|\nabla_\theta \frac{\delta E(\rho)}{\delta\rho}(\theta_1,t_1) - \nabla_\theta \frac{\delta E(\rho)}{\delta\rho}(\theta_2,t_2)\right|$$
$$\leq Q_1\left(\mathcal{L}, \max_{i=1,2}(|\theta_{i,[1]}|)\right)|\theta_1 - \theta_2| + Q_1\left(\mathcal{L}^{\mathrm{sup}}, \max_{i=1,2}(|\theta_{i,[1]}|)\right)(|\theta_2|+1)|t_1 - t_2|\,, \tag{42}$$

- *Lipschitz continuity in $\rho$. There exists $Q : \mathbb{R}^2 \to \mathbb{R}^+$ that increasingly depends on both arguments such that*

$$\left| \nabla_\theta \frac{\delta E(\rho_1)}{\delta \rho}(\theta, t) - \nabla_\theta \frac{\delta E(\rho_2)}{\delta \rho}(\theta, t) \right| \leq Q(\mathcal{L}, |\theta_{[1]}|, \mathcal{R})(1 + |\theta|)d_1(\rho_1, \rho_2), \qquad (43)$$

*where $d_1$ is defined in Definition 4.1.*

**Proof** To prove the first bound of (41), we restate (29) as follows

$$\nabla_\theta \frac{\delta E(\rho)}{\delta \rho}(\theta, t) = \mathbb{E}_{x \sim \mu} \left( p_\rho^\top(t; x) \partial_\theta f(Z_\rho(t; x), \theta) \right) \,,$$

from which it follows that

$$\left| \nabla_\theta \frac{\delta E(\rho)}{\delta \rho}(\theta, t) \right| \leq \mathbb{E}_{x \sim \mu} \left( |\partial_\theta f(Z_\rho(t; x), \theta)| \, |p_\rho(t; x)| \right) \leq C(\mathcal{L})(|\theta| + 1) \,,$$

where we use (13), (18), and (31) in the second inequality. To prove the second bound in (41), we use the bound of $\left| \partial_{\theta_{[1]}} f(x, \theta) \right|$ according to Assumption 4.1 part 3 to obtain

$$\left| \partial_{\theta_{[1]}} \frac{\delta E(\rho)}{\delta \rho}(\theta, t) \right| \leq \mathbb{E}_{x \sim \mu} \left( \left| \partial_{\theta_{[1]}} f(Z_\rho(t; x), \theta) \right| \, |p_\rho(t; x)| \right) \leq C(\mathcal{L})(|\theta_{[1]}| + 1) \,,$$

To prove (42), we assume $t_1 > t_2$ without loss of generality, and use the triangle inequality to obtain

$$
\begin{aligned}
&\left| \nabla_\theta \frac{\delta E(\rho)}{\delta \rho}(\theta_1, t_1) - \nabla_\theta \frac{\delta E(\rho)}{\delta \rho}(\theta_2, t_2) \right| \\
&\leq \underbrace{\left| \mathbb{E}_{x \sim \mu} \left( p_\rho^\top(t_1; x) \partial_\theta f(Z_\rho(t_1; x), \theta_1) - p_\rho^\top(t_1; x) \partial_\theta f(Z_\rho(t_1; x), \theta_2) \right) \right|}_{(\mathrm{I})} \\
&\quad + \underbrace{\left| \mathbb{E}_{x \sim \mu} \left( p_\rho^\top(t_1; x) \partial_\theta f(Z_\rho(t_1; x), \theta_2) - p_\rho^\top(t_1; x) \partial_\theta f(Z_\rho(t_2; x), \theta_2) \right) \right|}_{(\mathrm{II})} \\
&\quad + \underbrace{\left| \mathbb{E}_{x \sim \mu} \left( p_\rho^\top(t_1; x) \partial_\theta f(Z_\rho(t_2; x), \theta_2) - p^\top(t_2; x) \partial_\theta f(Z_\rho(t_2; x), \theta_2) \right) \right|}_{(\mathrm{III})} \,.
\end{aligned}
\qquad (44)
$$

To bound (I), we use the mean-value theorem, Assumption 4.1 (14), and (18) to obtain

$$
\begin{aligned}
|\partial_\theta f(Z_\rho(t_1; x), \theta_1) - \partial_\theta f(Z_\rho(t_1; x), \theta_2)| &\leq |\partial_\theta^2 f(Z_\rho(t_1; x), (1 - \lambda)\theta_1 + \lambda\theta_2)||\theta_1 - \theta_2| \\
&\leq C\left( \mathcal{L}, \max_{i=1,2}(|\theta_{i,[1]}|) \right) |\theta_1 - \theta_2| \,.
\end{aligned}
$$

For (II), we note first that

$$
\begin{aligned}
|Z_\rho(t_1; x) - Z_\rho(t_2; x)| &\leq \left| \int_{t_2}^{t_1} \int_{\mathbb{R}^k} f(Z_\rho(t; x), \theta) \, \mathrm{d}\rho(\theta, t) \, \mathrm{d}t \right| \\
&\leq C(\mathcal{L}) \left| \int_{t_2}^{t_1} \int_{\mathbb{R}^k} \left( |\theta|^2 + 1 \right) \, \mathrm{d}\rho(\theta, t) \, \mathrm{d}t \right| \\
&\leq C(\mathcal{L})(\mathcal{L}^{\mathrm{sup}} + 1)|t_1 - t_2| \\
&\leq C(\mathcal{L}^{\mathrm{sup}})|t_1 - t_2| \,,
\end{aligned}
$$

where we use Assumption 4.1 (12) together with (18) and $\mathcal{L} \leq \mathcal{L}^{\mathrm{sup}}$. This bound implies that

$$
\begin{aligned}
(\mathrm{II}) &\leq \mathbb{E}_{x \sim \mu} \left( |p_\rho^\top(t_1; x)| \, |\partial_\theta f(Z_\rho(t_1; x), \theta_2) - \partial_\theta f(Z_\rho(t_2; x), \theta_2)| \right) \\
&\leq C(\mathcal{L}) \mathbb{E}_{x \sim \mu} \left( |\partial_\theta f(Z_\rho(t_1; x), \theta_2) - \partial_\theta f(Z_\rho(t_2; x), \theta_2)| \right) \\
&\leq C(\mathcal{L}^{\mathrm{sup}}, |\theta_{2,[1]}|) \mathbb{E}_{x \sim \mu} \left( |Z_\rho(t_1; x) - Z_\rho(t_2; x)| \right) \\
&\leq C(\mathcal{L}^{\mathrm{sup}}, |\theta_{2,[1]}|)|t_1 - t_2| \,,
\end{aligned}
$$

where we use (31) in the first inequality, Assumption 4.1 (14) for $f$, and (18) in the second inequality.

To bound term (III), we again use boundedness of $Z_\rho$, $p_\rho$ and the Lipschitz condition of $f$ to obtain

$$
\begin{aligned}
|p_\rho(t_1; x) - p_\rho(t_2; x)| &\leq \left| \int_{t_2}^{t_1} \int_{\mathbb{R}^k} p_\rho^\top(t; x) \partial_z f(Z_\rho(t; x), \theta) \, \mathrm{d}\rho(\theta, t) \, \mathrm{d}t \right| \\
&\leq C(\mathcal{L}) \left| \int_{t_2}^{t_1} \int_{\mathbb{R}^k} (|\theta|^2 + 1) \, \mathrm{d}\rho(\theta, t) \, \mathrm{d}t \right| \\
&\leq C(\mathcal{L})(\mathcal{L}^{\mathrm{sup}} + 1)|t_1 - t_2| \\
&\leq C(\mathcal{L}^{\mathrm{sup}})|t_1 - t_2| \,,
\end{aligned}
$$

so that

$$
\text{(III)} \leq \mathbb{E}_{x \sim \mu} \left( |\partial_\theta f(Z_\rho(t_2; x), \theta_2)| |p_\rho(t_1; x) - p_\rho(t_2; x)| \right) \leq C(\mathcal{L}^{\mathrm{sup}})(|\theta_2| + 1)|t_1 - t_2| \,,
$$

where we use Assumption 4.1 (13) in the second inequality. By substituting these bounds into (44), we complete the proof of (42).

Finally, to prove (43), we recall the definition of the Fréchet derivative, to obtain

$$
\begin{aligned}
&\left| \nabla_\theta \frac{\delta E(\rho_1)}{\delta \rho}(\theta, t) - \nabla_\theta \frac{\delta E(\rho_2)}{\delta \rho}(\theta, t) \right| \\
\leq &\mathbb{E}_{x \sim \mu} \left( |\partial_\theta f(Z_{\rho_1}(t; x), \theta) p_{\rho_1}(t; x) - \partial_\theta f(Z_{\rho_2}(t; x), \theta) p_{\rho_2}(t; x)| \right) \\
\leq &\mathbb{E}_{x \sim \mu} \left( |\partial_\theta f(Z_{\rho_1}(t; x), \theta) - \partial_\theta f(Z_{\rho_2}(t; x), \theta)| \, |p_{\rho_1}(t; x)| \right) \\
&+ \mathbb{E}_{x \sim \mu} \left( |\partial_\theta f(Z_{\rho_2}(t; x), \theta)| \, |p_{\rho_1}(t; x) - p_{\rho_2}(t; x)| \right) \\
\leq &C(\mathcal{L}, |\theta_{[1]}|) \mathbb{E}_{x \sim \mu} \left( |Z_{\rho_1}(t; x) - Z_{\rho_2}(t; x)| \right) + C(\mathcal{L})(|\theta| + 1) \mathbb{E}_{x \sim \mu} \left( |p_{\rho_1}(t; x) - p_{\rho_2}(t; x)| \right) \\
\leq &C(\mathcal{L}, |\theta_{[1]}|, \mathcal{R})(1 + |\theta|) d_1(\rho_1, \rho_2) \,,
\end{aligned}
$$

where we use Assumption 4.1 for $f$ with (18) and (31) in the third inequality. In the last inequality, we use (19), (32). ∎

## C.2 A-PRIORI ESTIMATE UNDER THE HOMOGENEOUS ASSUMPTION

In Theorem 6.1 and 6.2, 2-homogeneity or partial 1-homogeneity are assumed. When these properties hold, we can sharpen the estimates obtained in the previous section. We summarize our results here.

**Lemma C.3** *Suppose that Assumption 4.1 holds, then there exists a constant $C_3(r)$ depending increasingly on $r$ such that for any $r > 0$ with $|x| < r$ and $\theta \in \mathbb{R}^k$, we have the following.*

- *If $f$ is 2-homogeneous (Assumption 6.1), then*

$$
\left| \partial_x^2 f \right| \leq C_3(r) \left( |\theta|^2 + 1 \right), \quad |\partial_x \partial_\theta f| \leq C_3(r) \left( |\theta| + 1 \right), \quad \left| \partial_\theta^2 f \right| \leq C_3(r). \tag{45}
$$

- *If $f$ is partially 1-homogeneous (Assumption 6.2), then*

$$
\left| \partial_x^2 f \right| \leq C_3(r)(|\theta| + 1), \quad |\partial_x \partial_\theta f| \leq C_3(r)(|\theta| + 1), \quad \left| \partial_\theta^2 f \right| \leq C_3(r)(|\theta| + 1), \tag{46}
$$

*where $|\cdot|$ denotes the Frobenius norm.*

**Proof** When $f$ satisfies Assumption 4.1 and Assumption 6.2, (46) can be obtained from direct calculation. When $f$ is 2-homogeneous in $\theta$, we have

$$
f(x, \theta) = |\theta|^2 f(x, \theta/|\theta|) = f_1(\theta) f_2(x, \theta) \,,
$$

where $f_1(\theta) = |\theta|^2$, $f_2(x, \theta) = f(x, \theta/|\theta|)$. Naturally, with product rule, the derivatives of $f$ becomes the products of derivatives of $f_1$ and $f_2$. We then obtain (45), noting that when $|\theta| > 1$, we have

$$
\left| \partial_x^2 f_1 \right| = 0, \quad |\partial_x \partial_\theta f_1| \leq 0, \quad |\partial_x f_1| = 0, \quad |\partial_\theta f_1| \leq 2|\theta|, \quad \left| \partial_\theta^2 f_1 \right| \leq 2k,
$$

and

$$
\left| \partial_x^2 f_2 \right| = C(r), \quad |\partial_x \partial_\theta f_2| \leq \frac{C(r)}{|\theta|}, \quad |\partial_x f_2| = C(r), \quad |\partial_\theta f_2| \leq \frac{C(r)}{|\theta|}, \quad \left| \partial_\theta^2 f_2 \right| \leq \frac{C(r)}{|\theta|^2} \,.
$$

■

The estimates above allow us to improve the stability results in Lemmas C.1 and C.2.

**Lemma C.4** *Suppose that Assumption 4.1 holds and let $\rho_1, \rho_2 \in \mathcal{C}([0,1]; \mathcal{P}^2)$. Define $\mathcal{L}_1$ and $\mathcal{L}_2$ as in (20) and*

$$\mathcal{L}_1^{\sup} = \sup_{t \in [0,1]} \int_{\mathbb{R}^k} |\theta|^2 d\rho_1(\theta, t) \,.$$

*Then we have the following results.*

- *If $f$ either satisfies Assumption 6.1 or Assumption 6.2, we have the following properties.*

  1. *Stability of $p_\rho$:*
  $$|p_{\rho_1}(t; x) - p_{\rho_2}(t; x)| \leq C(\mathcal{L}_1, \mathcal{L}_2) d_1(\rho_1, \rho_2) \,. \tag{47}$$

  2. *Lipschitz continuity in $\rho$:*
  $$\left| \nabla_\theta \frac{\delta E(\rho_1)}{\delta \rho}(\theta, t) - \nabla_\theta \frac{\delta E(\rho_2)}{\delta \rho}(\theta, t) \right| \leq C(\mathcal{L}_1, \mathcal{L}_2) d_1(\rho_1, \rho_2) (|\theta| + 1) \,, \tag{48}$$
  *where $d_1$ is defined in (4.1).*

- *Lipschitz continuity in $\theta$ and $t$: If $f$ satisfies Assumption 6.1, then for any $(\theta_1, t_1), (\theta_2, t_2) \in \mathbb{R}^k \times [0,1]$, we have*
$$\left| \nabla_\theta \frac{\delta E(\rho_1)}{\delta \rho}(\theta_1, t_1) - \nabla_\theta \frac{\delta E(\rho_1)}{\delta \rho}(\theta_2, t_2) \right| \leq C(\mathcal{L}_1)|\theta_1 - \theta_2| + C(\mathcal{L}_1^{\sup})(|\theta_2| + 1)|t_1 - t_2| \,. \tag{49}$$

- *Lipschitz continuity in $\theta$ and $t$: If $f$ satisfies Assumption 6.2, then for any $(\theta_1, t_1), (\theta_2, t_2) \in \mathbb{R}^k \times [0,1]$, we have*
$$\left| \nabla_\theta \frac{\delta E(\rho_1)}{\delta \rho}(\theta_1, t_1) - \nabla_\theta \frac{\delta E(\rho_1)}{\delta \rho}(\theta_2, t_2) \right| \leq C(\mathcal{L}_1^{\sup})(|\theta_1| + |\theta_2| + 1)(|\theta_1 - \theta_2| + |t_1 - t_2|) \,, \tag{50}$$

**Remark C.1** *We note that in comparing (32) with (47) and (42)-(43) with (48)-(50), the main differences are the dependence of the bounds on $\theta$. The new estimates have explicit (and mild) dependence on $\theta$.*

**Proof** First, to prove (47), we let $\Delta(t; x) = p_{\rho_1}(t; x) - p_{\rho_2}(t; x)$, and recall (35) and (36). Using the boundedness of $Z_\rho$ in (18) and (19), and calling (45) (or (46)), we have from (36) that

$$|D_{\rho_1, \rho_2}(t; x)| \leq C(\mathcal{L}_1, \mathcal{L}_2) \left( \int_{\mathbb{R}^k} |\theta|^2 d\rho_1(\theta, t) + \int_{\mathbb{R}^k} |\theta|^2 d\rho_2(\theta, t) + 1 \right) d_1(\rho_1, \rho_2) \,.$$

By substituting into (35), and using (31), (33), and Hölder's inequality, we have

$$\frac{\mathrm{d}|\Delta(t; x)|^2}{\mathrm{d}t} \leq C(\mathcal{L}_1, \mathcal{L}_2) \left( \int_{\mathbb{R}^k} |\theta|^2 d\rho_1(\theta, t) + \int_{\mathbb{R}^k} |\theta|^2 d\rho_2(\theta, t) + 1 \right) |\Delta(t; x)|^2$$
$$+ C(\mathcal{L}_1, \mathcal{L}_2) \left( \int_{\mathbb{R}^k} |\theta|^2 d\rho_1(\theta, t) + \int_{\mathbb{R}^k} |\theta|^2 d\rho_2(\theta, t) + 1 \right) d_1^2(\rho_1, \rho_2) \,.$$

The result (47) follows from the initial condition (34) together with Grönwall's inequality.

Next, to prove (48), we have

$$\left| \nabla_\theta \frac{\delta E(\rho_1)}{\delta \rho}(\theta, t) - \nabla_\theta \frac{\delta E(\rho_2)}{\delta \rho}(\theta, t) \right|$$
$$\leq \mathbb{E}_{x \sim \mu} \left( |\partial_\theta f(Z_{\rho_1}(t; x), \theta) p_{\rho_1}(t; x) - \partial_\theta f(Z_{\rho_2}(t; x), \theta) p_{\rho_2}(t; x)| \right)$$
$$\leq \mathbb{E}_{x \sim \mu} \left( |\partial_\theta f(Z_{\rho_1}(t; x), \theta) - \partial_\theta f(Z_{\rho_2}(t; x), \theta)| |p_{\rho_1}(t; x)| \right)$$
$$\quad + \mathbb{E}_{x \sim \mu} \left( |\partial_\theta f(Z_{\rho_2}(t; x), \theta)| |p_{\rho_1}(t; x) - p_{\rho_2}(t; x)| \right)$$
$$\leq C(\mathcal{L}_1, \mathcal{L}_2)(|\theta| + 1) \left( \mathbb{E}_{x \sim \mu} \left( |Z_{\rho_1}(t; x) - Z_{\rho_2}(t; x)| \right) + \mathbb{E}_{x \sim \mu} \left( |p_{\rho_1}(t; x) - p_{\rho_2}(t; x)| \right) \right)$$
$$\leq C(\mathcal{L}_1, \mathcal{L}_2) d_1(\rho_1, \rho_2) (|\theta| + 1) \,,$$

where we also use (13), (18), (31), (45) (or (46)) in the second inequality and (19) and (47) in the final inequality.

Finally, to prove (49) and (50), we have as in (44) that

$$
\begin{aligned}
&\left| \nabla_\theta \frac{\delta E(\rho_1)}{\delta \rho}(\theta_1, t_1) - \nabla_\theta \frac{\delta E(\rho_1)}{\delta \rho}(\theta_2, t_2) \right| \\
&\leq \underbrace{\left| \mathbb{E}_{x \sim \mu} \left( p_{\rho_1}^\top(t_1; x) \partial_\theta f(Z_{\rho_1}(t_1; x), \theta_1) - p_{\rho_1}^\top(t_1; x) \partial_\theta f(Z_{\rho_1}(t_1; x), \theta_2) \right) \right|}_{\text{(I)}} \\
&+ \underbrace{\left| \mathbb{E}_{x \sim \mu} \left( p_{\rho_1}^\top(t_1; x) \partial_\theta f(Z_{\rho_1}(t_1; x), \theta_2) - p_{\rho_1}^\top(t_1; x) \partial_\theta f(Z_{\rho_1}(t_2; x), \theta_2) \right) \right|}_{\text{(II)}} \\
&+ \underbrace{\left| \mathbb{E}_{x \sim \mu} \left( p_{\rho_1}^\top(t_1; x) \partial_\theta f(Z_{\rho_1}(t_2; x), \theta_2) - p_{\rho_1}^\top(t_2; x) \partial_\theta f(Z_{\rho_1}(t_2; x), \theta_2) \right) \right|}_{\text{(III)}} .
\end{aligned} \tag{51}
$$

The boundedness of the three terms above relies on Assumption 4.1 and (45) (or (46)).

To bound (I), if $f$ is 2-homogeneous, we have

$$
|(\text{I})| \leq C(\mathcal{L}_1)|\theta_1 - \theta_2|,
$$

where we use (18), (31), and $\partial_\theta^2 f \leq C(|z|)$. If $f$ satisfies Assumption 6.2, we have

$$
|(\text{I})| \leq C(\mathcal{L}_1)(|\theta_1| + |\theta_2| + 1)|\theta_1 - \theta_2|,
$$

where we use (18), (31), and $\partial_\theta^2 f(z, \theta) \leq C(|z|)(|\theta| + 1)$.

The bounds of (II) and (III) are same for both homogeneity assumptions and similar to the proof of Lemma C.2. For (II), we have

$$
|\partial_\theta f(Z_{\rho_1}(t_1; x), \theta_2) - \partial_\theta f(Z_{\rho_1}(t_2; x), \theta_2)| \leq C(\mathcal{L}_1)(|\theta_2| + 1)|Z_{\rho_1}(t_1; x) - Z_{\rho_1}(t_2; x)|
$$

and

$$
|Z_{\rho_1}(t_1; x) - Z_{\rho_1}(t_2; x)| \leq C(\mathcal{L}_1)(\mathcal{L}_1^{\text{sup}} + 1)|t_1 - t_2|.
$$

For (III), we also use

$$
|\partial_\theta f(Z_{\rho_1}(t_2; x), \theta_2)| \leq C(\mathcal{L}_1)(|\theta_2| + 1)
$$

and

$$
|p_{\rho_1}(t_1; x) - p_{\rho_1}(t_2; x)| \leq C(\mathcal{L}_1)(\mathcal{L}_1^{\text{sup}} + 1)|t_1 - t_2|.
$$

These estimates, together with $\mathcal{L}_1 \leq \mathcal{L}_1^{\text{sup}}$ and (51), prove the claims (49) and (50). ∎

## D    WELL POSEDNESS OF GRADIENT FLOW

Theorem A.4 is about well-posedness of the gradient flow equation (9) in the mean-field limit, while Theorem A.3 shows the corresponding result in the discrete setting. The proof for the two are similar. We first show the mean-field limit well-posedness, Theorem A.4.

### D.1    PROOF OF THEOREM A.4

We use the fixed-point argument. To do so, we first define a subset of $\mathcal{C}([0, S]; \mathcal{C}([0, 1]; \mathcal{P}^2))$ with compact support measures, as follows:

$$
\begin{aligned}
\Omega_S = &\left\{ \phi(\theta, t, s) \in \mathcal{C}([0, S]; \mathcal{C}([0, 1]; \mathcal{P}^2)) \big| \exists r > 0, \ \text{supp}(\phi) \subset \{\theta \mid |\theta_{[1]}| < r\}, \ \forall (t, s) \in [0, 1] \times [0, S] \right\} \\
&\cap \left\{ \phi(\theta, t, s) \in \mathcal{C}([0, S]; \mathcal{C}([0, 1]; \mathcal{P}^2)) \big| \phi(\theta, t, 0) = \rho_{\text{ini}}(\theta, t) \right\}
\end{aligned}
$$

For any $\phi(\theta, t, s) \in \Omega_S$ with $\phi(\theta, t, 0) = \rho_{\text{ini}}(\theta, t)$, we define a map

$$
\varphi = \mathcal{T}_S(\phi) : \Omega_S \to \Omega_S \tag{52}
$$

where $\varphi$ solves

$$\begin{cases} \dfrac{\partial \varphi(\theta, t, s)}{\partial s} = \nabla_\theta \cdot \left( \varphi(\theta, t, s) \nabla_\theta \dfrac{\delta E(\phi(s))}{\delta \rho}(\theta, t) \right) , \\ \varphi(\theta, t, 0) = \rho_{\mathrm{ini}}(\theta, t) . \end{cases} \tag{53}$$

The strategy is to show this map is a contraction map, so that there is a fixed point in the set $\Omega_S$, which is then the solution to equation (9).

The proof of Theorem A.4 is divided into three steps:

Step 1: We show $\mathcal{T}_S$ is well-defined map from $\Omega_S$ to $\Omega_S$.

Step 2: We give a bound of $d_2(\mathcal{T}_S(\phi_1), \mathcal{T}_S(\phi_2))$ in terms of $d_2(\phi_1, \phi_2)$. One can then tune $S$ to ensure $\mathcal{T}_S$ is a contraction map, meaning there is a unique fixed point $\phi^*$ so that $\phi^* = \mathcal{T}_S(\phi^*)$, and thus $\phi^*$ solves (53) for $s < S$.

Step 3: We extend the local solution to a global solution.

**Step 1.** According to the definition of (53), for a fixed $\phi(\theta, t, s)$, let

$$\begin{cases} \frac{\mathrm{d}\theta_\phi(s;t)}{\mathrm{d}s} = -\nabla_\theta \frac{\delta E(\phi(s))}{\delta \rho} (\theta_\phi(s;t), t) , \\ \theta_\phi(0;t) \sim \rho_{\mathrm{ini}}(\theta, t) . \end{cases} \tag{54}$$

Then $\theta_\phi \sim \varphi = \mathcal{T}_S(\phi)$. To show the existence of $\varphi$ amounts to showing the wellposedness of (54).

According to Lemma C.2 (41) and (42), the force $\nabla_\theta \frac{\delta E}{\delta \phi(s)}(\cdot, t)$ has a linear growth and is locally Lipschitz. Classical ODE theory then suggests there is a unique solution for $s \in [0, S]$, which depends continuously on the initial value $\theta(0; t)$.

Denoting

$$\begin{aligned} \mathcal{L}_{S,\phi} &= \sup_{0 \le s \le S} \int_0^1 \int_{\mathbb{R}^k} |\theta|^2 \, \mathrm{d}\phi(\theta, t, s) \, \mathrm{d}t, \\ \mathcal{L}_{\mathrm{ini}}^{\sup} &= \sup_{0 \le t \le 1} \int_{\mathbb{R}^k} |\theta|^2 \, \mathrm{d}\rho_{\mathrm{ini}}(\theta, t), \\ \mathcal{L}_{S,\phi}^{\sup} &= \sup_{0 \le t \le 1, 0 \le s \le S} \int_{\mathbb{R}^k} |\theta|^2 \, \mathrm{d}\phi(\theta, t, s) \\ \mathcal{R}_{\mathrm{ini}} &= \inf_{r>0} \left\{ \mathrm{supp}(\rho_{\mathrm{ini}}(\theta, t)) \subset \{\theta | |\theta_{[1]}| < r\}, \ \forall t \in [0, 1] \right\}, \\ \mathcal{R}_{S,\phi} &= \inf_{r>0} \left\{ \mathrm{supp}(\phi(\theta, t, s)) \subset \{\theta | |\theta_{[1]}| < r\}, \ \forall (t, s) \in [0, 1] \times [0, S] \right\}, \end{aligned} \tag{55}$$

we have the following proposition.

**Proposition D.1** *Suppose that $\theta_\phi(s; t)$ solves (54) and $\phi \in \Omega_S$. Then for any $(t_1, s_1), (t_2, s_2) \in [0, 1] \times [0, S]$, we have*

$$\mathbb{E}\left( |\theta_\phi(s_1; t_1)|^2 \right) \le \exp(SC(\mathcal{L}_{S,\phi}))\left( \mathcal{L}_{\mathrm{ini}}^{\sup} + 1 \right), \ |\theta_{\phi,[1]}(s_1; t_1)|$$
$$\le \exp(SC(\mathcal{L}_{S,\phi}))\left( \mathcal{R}_{\mathrm{ini}} + 1 \right) \tag{56}$$

*and*

$$\mathbb{E}\left( |\theta_\phi(s_1; t_1) - \theta_\phi(s_2; t_2)|^2 \right)$$
$$\le C\left( \mathcal{L}_{S,\phi}^{\sup}, \mathcal{R}_{\mathrm{ini}}, S \right) \left( \mathbb{E}\left( |\theta_\phi(0; t_1) - \theta_\phi(0; t_2)|^2 \right) + |t_1 - t_2|^2 + |s_1 - s_2|^2 \right) . \tag{57}$$

**Proof** To prove the first bound in (56), we multiply (54) on both sides by $\theta_\phi$ and use boundedness of the forcing term (41) to obtain

$$\frac{\mathrm{d}\, |\theta_\phi(s; t_1)|^2}{\mathrm{d}s} \le C(\mathcal{L}_{S,\phi})(|\theta_\phi(s; t_1)|^2 + 1) .$$

Using Grönwall's inquality together with $\mathbb{E}\left(|\theta_\phi(0; t_1)|^2\right) \leq \mathcal{L}_{\text{ini}}^{\text{sup}}$, we obtain the first bound in (56). To prove the second bound in (56), we use the bound of $\left|\partial_{\theta_{[1]}} \frac{\delta E(\rho)}{\delta \rho}(\theta, t)\right|$ according to Lemma C.2 (41):

$$\frac{\mathrm{d}\left|\theta_{\phi,[1]}(s; t_1)\right|^2}{\mathrm{d}s} \leq C(\mathcal{L}_{S,\phi})(\left|\theta_{\phi,[1]}(s; t_1)\right|^2 + 1).$$

Using Grönwall's inquality together with $|\theta_{\phi,[1]}(0; t_1)| \leq \mathcal{R}_{\text{ini}}$, we obtain the second bound in (56). To show (57), we first write

$$\mathbb{E}\left(|\theta_\phi(s_1; t_1) - \theta_\phi(s_2; t_2)|^2\right)$$
$$\leq 2\underbrace{\mathbb{E}\left(|\theta_\phi(s_1; t_1) - \theta_\phi(s_1; t_2)|^2\right)}_{(I)} + 2\underbrace{\mathbb{E}\left(|\theta_\phi(s_1; t_2) - \theta_\phi(s_2; t_2)|^2\right)}_{(II)}, \tag{58}$$

then bound the two terms (I) and (II) separately. For (I), we use (54), the second bound of (56), and Lemma C.2 (42)

$$\frac{\mathrm{d}\left|\theta_\phi(s; t_1) - \theta_\phi(s; t_2)\right|^2}{\mathrm{d}s}$$
$$\leq C(\mathcal{L}_{S,\phi}, \mathcal{R}_{\text{ini}})\left|\theta_\phi(s; t_1) - \theta_\phi(s; t_2)\right|^2 + C(\mathcal{L}_{S,\phi}^{\text{sup}}, \mathcal{R}_{\text{ini}})(|\theta_\phi(s_1; t_1)|^2 + |\theta_\phi(s_1; t_2)|^2 + 1)|t_1 - t_2|^2,$$

Using the first bound in (56) and Grönwall's inequality, this implies that

$$\mathbb{E}\left(|\theta_\phi(s_1; t_1) - \theta_\phi(s_1; t_2)|^2\right) \leq C\left(\mathcal{L}_{S,\phi}^{\text{sup}}, \mathcal{R}_{\text{ini}}, S\right)\left(\mathbb{E}\left(|\theta_\phi(0; t_1) - \theta_\phi(0; t_2)|^2\right) + |t_1 - t_2|^2\right). \tag{59}$$

For (II), we obtain an estimate by integrating (54) from $s_1$ to $s_2$ and using the boundedness of $\nabla_\theta \frac{\delta E(\rho)}{\delta \rho}$ in (41). From the Grönwall inequality, we have

$$|\theta_\phi(s_1; t_2) - \theta_\phi(s_2; t_2)| \leq C\left(\mathcal{L}_{S,\phi}\right)\left(\int_{s_1}^{s_2} |\theta_\phi(s; t_2)|\,\mathrm{d}s + |s_1 - s_2|\right).$$

Using first bound in (56) and Hölder's inequality, this implies

$$\mathbb{E}\left|\theta_\phi(s_1; t_2) - \theta_\phi(s_2; t_2)\right|^2 \leq C\left(\mathcal{L}_{S,\phi}, \mathcal{L}_{\text{ini}}^{\text{sup}}, S\right)|s_1 - s_2|^2 \tag{60}$$

and thus combining with (59) and substituting in (58), we complete the proof. ∎

An immediate corollary of Proposition D.1 is that the map $\mathcal{T}_S(\cdot)$ is well defined.

**Corollary D.1** *For every fixed $S > 0$, the map $\mathcal{T}_S$ is well defined. That is, for any $\phi \in \Omega_S$, one can find $\varphi = \mathcal{T}_S(\phi) \in \Omega_S$ as the unique solution of (53). In particular, for any $(t, s) \in [0, 1] \times [0, S]$, we have*

$$\int_{\mathbb{R}^k} |\theta|^2\,\mathrm{d}\varphi(\theta, t, s) \leq \exp(SQ_1(\mathcal{L}_{S,\phi}))\left(\mathcal{L}_{\text{ini}}^{\text{sup}} + 1\right),$$
$$\text{supp}_\theta(\varphi(\theta, t, s)) \subset \left\{\theta \big| |\theta_{[1]}| \leq \exp(SQ_1(\mathcal{L}_{S,\phi}))\left(\mathcal{R}_{\text{ini}} + 1\right)\right\}, \tag{61}$$

*where $Q_1 : \mathbb{R}_+ \to \mathbb{R}_+$ is depends only on $\mathcal{L}_{S,\phi}$ and is an increasing function of its argument.*

**Proof** For fixed $(t, s) \in [0, 1] \times [0, S]$, define $\varphi(\theta, t, s)$ as the distribution of $\theta_\phi(s; t)$. Using classical stochastic theory (Ambrosio et al., 2008, Prop 8.1.8), $\varphi(\theta, t, s)$ is a solution to (53). The estimate of the support is a consequence of (56). Finally, using (16) and (57), we obtain that

$$\lim_{(t,s) \to (t_0, s_0)} W_2(\varphi(\cdot, t, s), \varphi(\cdot, t_0, s_0)) \leq \lim_{(t,s)}\left(\mathbb{E}\left(|\theta_\phi(s; t) - \theta_\phi(s_0; t_0)|^2\right)\right)^{1/2} = 0,$$

which proves the continuity in $s$ and $t$, so that $\varphi \in \mathcal{C}([0, S]; \mathcal{C}([0, 1]; \mathcal{P}^2))$. By combining all the factors above, we conclude that $\varphi \in \Omega_S$. ∎

**Step 2.** We show now that $\mathcal{T}_S$ is a contraction map for $S$ sufficiently small.

**Proposition D.2** *For any $\phi_1, \phi_2 \in \Omega_S$, we have*

$$d_2(\mathcal{T}_S(\phi_1), \mathcal{T}_S(\phi_2)) \leq SQ_2(\mathcal{L}_S, \mathcal{R}_S, S)d_2(\phi_1, \phi_2), \tag{62}$$

*where $Q_2 : \mathbb{R}^3 \to \mathbb{R}_+$ is an increasing function and $\mathcal{L}_S = \max\{\mathcal{L}_{S,\phi_1}, \mathcal{L}_{S,\phi_2}\}$, $\mathcal{R}_S = \max\{\mathcal{R}_{S,\phi_1}, \mathcal{R}_{S,\phi_2}\}$, with $\mathcal{L}_{S,\phi}, \mathcal{R}_{S,\phi}$ defined in (55).*

**Proof** Denote by $\theta_{\phi_i}(s; t)$ the solutions to (54) with $\phi = \phi_i$ using the same initial data, that is,

$$\theta_{\phi_1}(0; t) = \theta_{\phi_2}(0; t).$$

As in the previous subsection, we translate the study of $\varphi_i$ to the study of $\theta_{\phi_i}$. Defining

$$\Delta_t(s) = |\theta_{\phi_1}(s; t) - \theta_{\phi_2}(s; t)|,$$

we have according to the definition of Wasserstein distance that

$$d_2(\mathcal{T}_S(\phi_1), \mathcal{T}_S(\phi_2)) \leq \sup_{(t,s)\in[0,1]\times[0,S]} \mathbb{E}\left(\Delta_t^2(s)\right).$$

Using (54), we obtain

$$\frac{\mathrm{d}(\Delta_t(s))^2}{\mathrm{d}s} \leq 4(\Delta_t(s))^2 + 4\left|\nabla_\theta \frac{\delta E(\phi_1(s))}{\delta\rho}(\theta_{\phi_1}, t) - \nabla_\theta \frac{\delta E(\phi_2(s))}{\delta\rho}(\theta_{\phi_2}, t)\right|^2$$

$$\leq 4(\Delta_t(s))^2 + 8\left|\nabla_\theta \frac{\delta E(\phi_1(s))}{\delta\rho}(\theta_{\phi_1}, t) - \nabla_\theta \frac{\delta E(\phi_1(s))}{\delta\rho}(\theta_{\phi_2}, t)\right|^2 \tag{63}$$

$$+ 8\left|\nabla_\theta \frac{\delta E(\phi_1(s))}{\delta\rho}(\theta_{\phi_2}, t) - \nabla_\theta \frac{\delta E(\phi_2(s))}{\delta\rho}(\theta_{\phi_2}, t)\right|^2.$$

The second term on the right hand side involves the continuity addressed in Lemma C.2 (42)

$$\left|\nabla_\theta \frac{\delta E(\phi_1(s))}{\delta\rho}(\theta_{\phi_1}, t) - \nabla_\theta \frac{\delta E(\phi_1(s))}{\delta\rho}(\theta_{\phi_2}, t)\right|^2 \leq C(\mathcal{L}_S, \mathcal{R}_S, S)(\Delta_t(s))^2,$$

where we use the second bound in (56) to substitute the constant in (42). Then the last term of (63) involves the continuity discussed in (43). In particular, we have

$$\left|\nabla_\theta \frac{\delta E(\phi_1(s))}{\delta\rho}(\theta_{\phi_2}, t) - \nabla_\theta \frac{\delta E(\phi_2(s))}{\delta\rho}(\theta_{\phi_2}, t)\right|$$

$$\leq C(\mathcal{R}_S, |\theta_{\phi_2,[1]}|)(1 + |\theta_{\phi_2}|)d_1(\phi_1, \phi_2) \leq C(\mathcal{L}_S, \mathcal{R}_S, S)(1 + |\theta_{\phi_2}|)d_2(\phi_1, \phi_2),$$

where $d_2$ is defined in Definition 4.1 and in the second inequality we use the second bound in (56) to substitute the constant in (43).

By substituting in (63), we obtain

$$\frac{\mathrm{d}(\Delta_t(s))^2}{\mathrm{d}s} \leq C(\mathcal{L}_S, \mathcal{R}_S, S)\left[(\Delta_t(s))^2 + (1 + |\theta_{\phi_2}|^2)d_2^2(\phi_1, \phi_2)\right],$$

which implies

$$\frac{\mathrm{d}\left(\mathbb{E}(\Delta_t(s))^2\right)}{\mathrm{d}s} \leq C(\mathcal{L}_S, \mathcal{R}_S, S)\left[\left(\mathbb{E}(\Delta_t(s))^2\right) + d_2^2(\phi_1, \phi_2)\right],$$

where we use the first inequality in (56). From the Grönwall inequality, there exists $Q_2 : \mathbb{R}^3 \to \mathbb{R}_+$ is an increasing function such that

$$\mathbb{E}\left((\Delta_t(s))^2\right) \leq SQ_2(\mathcal{L}_S, \mathcal{R}_S, S)d_2^2(\phi_1, \phi_2),$$

completing the proof. ∎

To apply the contraction mapping theorem, we need to verify two conditions in order to show that there exists a fixed point $\phi^* = \mathcal{T}_S(\phi^*)$:

- There is a closed subset in $\Omega_S$ such that $\mathcal{T}_S$ maps this set to itself.
- $\mathcal{T}_S$ is a contraction map in this subset.

For the closed subset, we define

$$
\begin{aligned}
B_{\rho_0} = & \left\{\phi \in \Omega_S \big| \mathrm{supp}(\phi(t,s)) \subset \{\theta||\theta_{[1]}| \leq 4(\mathcal{R}_{\mathrm{ini}}+1)\}, \quad \forall (t,s) \in [0,1] \times [0,S]\right\} \\
& \cap \left\{\phi \in \Omega_S \Big| \int_{\mathbb{R}^k} |\theta|^2 \, \mathrm{d}\phi(\theta,t,s) \leq 4\left(\mathcal{L}_{\mathrm{ini}}^{\mathrm{sup}}+1\right), \quad \forall (t,s) \in [0,1] \times [0,S]\right\} .
\end{aligned}
\tag{64}
$$

We now claim that for small enough $S$, $\mathcal{T}_S$ is a contraction map in $B_{\rho_0}$.

**Proposition D.3** *Suppose that $S$ is small enough that*

$$
\exp(SQ_1(4(\mathcal{L}_{\mathrm{ini}}^{\mathrm{sup}}+1)))\,(\mathcal{L}_{\mathrm{ini}}^{\mathrm{sup}}+1) \leq 4(\mathcal{L}_{\mathrm{ini}}^{\mathrm{sup}}+1\,,
$$

$$
\exp(SQ_1(4(\mathcal{L}_{\mathrm{ini}}^{\mathrm{sup}}+1)))\,(\mathcal{R}_{\mathrm{ini}}+1) \leq 4(\mathcal{R}_{\mathrm{ini}}+1)\,,
$$

$$
SQ_2(4(\mathcal{L}_{\mathrm{ini}}^{\mathrm{sup}}+1), 4(\mathcal{R}_{\mathrm{ini}}+1), S) < \frac{1}{2}\,,
$$

*where $Q_1$ and $Q_2$ are defined in Corollary D.1 and Proposition D.2, respectively. Then we have the following.*

- *If $\phi \in B_{\rho_0}$, then $\mathcal{T}_S(\phi) \in B_{\rho_0}$, that is, for any $(t,s) \in [0,1] \times [0,S]$, we have*
$$
\mathrm{supp}(\mathcal{T}_S(\phi)(t,s)) \subset \{\theta||\theta_{[1]}| \leq 4(\mathcal{R}_{\mathrm{ini}}+1)\}\,,
\tag{65}
$$
    *and*
$$
\int_{\mathbb{R}^k} |\theta|^2 \, \mathrm{d}\mathcal{T}_S(\phi)(\theta,t,s) \leq 4\left(\mathcal{L}_{\mathrm{ini}}^{\mathrm{sup}}+1\right)\,.
\tag{66}
$$

- *$\mathcal{T}_S$ is a contraction map in this subset, meaning that for any $\phi_1, \phi_2 \in B_{\rho_0}$, we have*
$$
d_2(\mathcal{T}_S(\phi_1), \mathcal{T}_S(\phi_2)) < \frac{1}{2} d_2(\phi_1, \phi_2)\,.
\tag{67}
$$

**Proof** First, using Corollary D.1 (61) and noticing $\mathcal{L}_{S,\phi} \leq 4\left(\mathcal{L}_{\mathrm{ini}}^{\mathrm{sup}}+1\right)$, we prove (65), (66). Then, using (62) with (65) and (66), we have

$$
d_2(\mathcal{T}_S(\phi_1), \mathcal{T}_S(\phi_2)) \leq SQ_2(4(\mathcal{L}_{\mathrm{ini}}^{\mathrm{sup}}+1), 4(\mathcal{R}_{\mathrm{ini}}+1), S) < \frac{1}{2} d_2(\phi_1, \phi_2),
$$

which proves (67). ∎

Using the contraction mapping theorem, we can obtain directly that $\mathcal{T}_S(\phi)$ has a fixed point in $B_{\rho_0}$ when $S$ is small enough.

**Corollary D.2** *If $S$ satisfies conditions in Proposition D.3, then there exists a unique $\phi^*(\theta,t,s) \in B_{\rho_0} \subset \Omega_S$ such that $\phi^*(\theta,t,s)$ is a solution to (9) with initial condition $\rho_{\mathrm{ini}}(\theta,t)$.*

This is a direct consequence of the application of contraction mapping theorem.

Finally, we prove that the cost function decreases along the flow.

**Lemma D.1** *Suppose that $\phi^*(\theta,t,s) \in \mathcal{C}([0,S]; \mathcal{C}([0,1]; \mathcal{P}^2))$ solves (9) with initial condition $\rho_{\mathrm{ini}}(\theta,t)$. Then for $0 < s < S$, we have*

$$
\frac{\mathrm{d}E(\phi^*(\theta,t,s))}{\mathrm{d}s} = -\int_0^1 \int_{\mathbb{R}^k} \left|\nabla_\theta \frac{\delta E(\phi^*(s))}{\delta \rho}(\theta,t)\right|^2 \, \mathrm{d}\phi^*(\theta,t,s)\,\mathrm{d}t \leq 0\,.
\tag{68}
$$

**Proof** Denote by $\theta^*(s;t)$ the associated path, meaning that $\theta^*(s;t)$ solves (54) with $\phi = \phi^*$, then $\theta^* \sim \phi^*$, meaning the distribution of $\theta^*$ is $\phi^*$. According to (9), we obtain using a change of variable that

$$
\frac{\mathrm{d}E(\phi^*(\theta,t,s))}{\mathrm{d}s} = -\int_0^1 \int_{\mathbb{R}^k} \left|\nabla_\theta \frac{\delta E(\phi^*(s))}{\delta \rho}(\theta,t)\right|^2 \, \mathrm{d}\phi^*(\theta,t,s)\,\mathrm{d}t \leq 0\,,
\tag{69}
$$

which proves the result. We note that the derivation in (69) is formal. A rigorous proof can be found in (Ding et al., 2021, Appendix I). ∎

**Step 3.** In this final step of the proof, we extend the local solution from Corollary D.2 to a global solution. Lemma D.1 shows that the formula of $\frac{\mathrm{d}E}{\mathrm{d}s}$, so we can then use this formula to improve the bound for the support of the solution (61). This improvement will be shown in the following corollary. This improved estimate helps in extending the local solution to the global solution.

**Corollary D.3** *For fixed $S$ satisfying the condition in Proposition D.3, denote by $\phi^*(\theta, t, s) \in \mathcal{C}([0, S]; \mathcal{C}([0, 1]; \mathcal{P}^2))$ the solution to* (9) *with initial condition $\rho_{\mathrm{ini}}(\theta, t)$. Then for any $(t, s) \in [0, 1] \times [0, S]$, we have*

$$\int_{\mathbb{R}^k} |\theta|^2 \, \mathrm{d}\phi^*(\theta, t, s) \leq C(S, \mathcal{R}_{\mathrm{ini}}, \mathcal{L}_{\mathrm{ini}}^{\mathrm{sup}}),$$

$$\mathrm{supp}(\phi^*(t, s)) \subset \left\{ \theta \big| |\theta_{[1]}| \leq C(S, \mathcal{R}_{\mathrm{ini}}, \mathcal{L}_{\mathrm{ini}}^{\mathrm{sup}}) \right\}, \tag{70}$$

*where the quantity $C$ depends only on $S$, $\mathcal{R}_{\mathrm{ini}}$, and $\mathcal{L}_{\mathrm{ini}}^{\mathrm{sup}}$).*

**Proof** According to (61), it suffices to prove

$$\mathcal{L}_{S, \phi^*} = \sup_{0 \leq s \leq S} \int_0^1 \int_{\mathbb{R}^k} |\theta|^2 \, \mathrm{d}\phi^*(\theta, t, s) \, \mathrm{d}t \leq C(S, \mathcal{R}_{\mathrm{ini}}, \mathcal{L}_{\mathrm{ini}}^{\mathrm{sup}}).$$

Denote by $\theta^*(s; t)$ the particle representation of $\phi^*$, meaning that $\theta^*(s; t)$ solves (54) with $\phi = \phi^*$. Since $\theta^*(s; t) \sim \phi^*(s; t)$,

$$\mathcal{L}_{S, \phi^*} = \sup_{0 \leq s \leq S} \int_0^1 \int_{\mathbb{R}^k} |\theta|^2 \, \mathrm{d}\phi^*(\theta, t, s) \, \mathrm{d}t = \sup_{0 \leq s \leq S} \int_0^1 \mathbb{E}\left( |\theta^*(s; t)|^2 \right) \, \mathrm{d}t.$$

Using (54), we obtain that

$$\frac{\mathrm{d}|\theta^*(s; t)|^2}{\mathrm{d}s} \leq |\theta^*(s; t)| \left| \nabla_\theta \frac{\delta E(\phi^*(s))}{\delta \rho} (\theta^*(s; t), t) \right|,$$

which gives

$$\frac{\mathrm{d} \int_0^1 \mathbb{E}\left( |\theta^*(s; t)|^2 \right) dt}{\mathrm{d}s}$$

$$\leq \left( \int_0^1 \mathbb{E}\left( |\theta^*(s; t)|^2 \right) dt \right)^{1/2} \left( \int_0^1 \mathbb{E}\left( \left| \nabla_\theta \frac{\delta E(\phi^*(s))}{\delta \rho} (\theta^*(s; t), t) \right|^2 \right) dt \right)$$

$$= \left( \int_0^1 \mathbb{E}\left( |\theta^*(s; t)|^2 \right) dt \right)^{1/2} \left| \frac{\mathrm{d}E(\phi^*(\theta, t, s))}{\mathrm{d}s} \right|,$$

where we use the Hölder inequality and (68) from Lemma D.1 in the last equality. Since

$$\int_0^1 \mathbb{E}\left( |\theta^*(0; t)|^2 \right) dt \leq \mathcal{L}_{\mathrm{ini}}^{\mathrm{sup}},$$

$$\int_0^S \left| \frac{\mathrm{d}E(\phi^*(\theta, t, s))}{\mathrm{d}s} \right| \, \mathrm{d}s \leq E(\rho_{\mathrm{ini}}(\theta, t)) - E(\phi^*(\theta, t, S)) \leq C(\mathcal{L}_{\mathrm{ini}}^{\mathrm{sup}}),$$

we obtain

$$\mathcal{L}_{S, \phi^*} = \sup_{0 \leq u \leq s} \int_0^1 \mathbb{E}\left( |\theta^*(u; t)|^2 \right) dt \leq C(S, \mathcal{L}_{\mathrm{ini}}^{\mathrm{sup}})$$

by Grönwall's inequality. This proves (70). ■

By contrast with (61), this estimate removes the dependence of the bound on $\mathcal{L}_{S, \phi}$. This improvement is important because it relaxes the fixed-point argument from the dependence on the initial guess $\phi$.

We are now ready to prove Theorem A.4.

**Proof** [Proof of Theorem A.4] From Corollary D.2, let $S_1$ be a constant satisfying the conditions in Proposition D.3. Then there is a local solution $\phi^* \in C([0, S_1]; \mathcal{C}([0, 1]; \mathcal{P}^2))$ to (9).

We now denote by $S^*$ the largest time within which the solution exists, where we denote this solution by $\phi^* \in C\left([0, S^*); \mathcal{C}([0,1]; \mathcal{P}^2)\right)$. We aim to show that $S^* = \infty$. According to Corollary D.3 (70), for any $s < S^*$ and $t \in [0, 1]$, we have

$$\int_{\mathbb{R}^k} |\theta|^2 \, \mathrm{d}\phi^*(\theta, t, s) \leq C(S^*, \mathcal{R}_{\mathrm{ini}}, \mathcal{L}_{\mathrm{ini}}^{\mathrm{sup}}),$$

$$\mathrm{supp}(\phi^*(t, s)) \subset \left\{\theta \big| |\theta_{[1]}| \leq C(S^*, \mathcal{R}_{\mathrm{ini}}, \mathcal{L}_{\mathrm{ini}}^{\mathrm{sup}})\right\},$$

Define $\mathcal{R}^* = \mathcal{R}_{S^*, \phi^*}$, $\mathcal{L}^* = \mathcal{L}_{S^*, \phi^*}^{\mathrm{sup}}$ according to (55). Since $\mathcal{R}^*, \mathcal{L}^* < \infty$, let us choose $\Delta_{S^*}$ small enough to satisfy

$$\exp(\Delta_{S^*} Q_1(4(\mathcal{L}^*+1)))\,(\mathcal{L}^* + 1) \leq 4(\mathcal{L}^*+1), \quad \exp(\Delta_{S^*} Q_1(4(\mathcal{L}^*+1)))\,(\mathcal{R}^* + 1) \leq 4(\mathcal{R}^*+1),$$

and

$$\Delta_{S^*} Q_2(4(\mathcal{L}^* + 1), 4(\mathcal{R}^* + 1), \Delta_{S^*}) \leq \frac{1}{2},$$

If $S^*$ is finite, then, using Proposition D.3 and Corollary D.2, we can further extend $\phi^*$ to be supported on $C\left([0, S^* + \Delta_{S^*}); \mathcal{C}([0,1]; \mathcal{P}^2)\right)$, giving a contradiction. If follows that $S^* = \infty$, as desired.

Finally, (21) is a direct result of Lemma D.1. ∎

## D.2 PROOF OF THEOREM A.3

This section is dedicated to Theorem A.3 — we show the well posedness of the gradient flow in the finite-layer case. We rewrite the gradient of (2) as follows

$$\frac{\partial E(\Theta_{L,M})}{\partial \theta_{l,m}} = \frac{1}{ML} \mathbb{E}_{x \sim \mu} \left( \partial_\theta f(Z_{\Theta_{L,M}}(l; x), \theta_{l,m}) p_{\Theta_{L,M}}(l; x) \right), \tag{71}$$

where $p_{\Theta_{L,M}}(l; x)$ solves:

$$\begin{cases} p_{\Theta_{L,M}}^\top(l; x) = p_{\Theta_{L,M}}^\top(l+1; x) \left( I + \frac{1}{ML} \sum_{m=1}^{M} \partial_z f\left( Z_{\Theta_{L,M}}(l+1; x), \theta_{l+1,i} \right) \right), \\ p_{\Theta_{L,M}}(L-1; x) = \left( g(Z_{\Theta_{L,M}}(L; x)) - y(x) \right) \nabla g(Z_{\Theta_{L,M}}(L; x)), \end{cases} \tag{72}$$

for $0 \leq l \leq L - 2$. We unify the space in a similar fashion to Definition 4.1.

**Definition D.1** $\Theta_{L,M} = \{\theta_{l,m}\}_{l=0,m=1}^{L-1,M} \in L_{L,M}^\infty$ *if and only if*

$$\sup_{l,m} |\theta_{l,m}| < \infty.$$

*The metric in $L_{L,M}^\infty$ is defined as*

$$d_{1,L,M}\left(\Theta_{L,M}, \widetilde{\Theta}_{L,M}\right) = \max_l \left( \frac{1}{M} \sum_{m=1}^{M} |\theta_{l,m} - \widetilde{\theta}_{l,m}|^2 \right)^{1/2}.$$

**Definition D.2** *For $s \geq 0$, we have $\Theta_{L,M}(s) = \{\theta_{l,m}(s)\}_{l=0,m=1}^{L-1,M} \in \mathcal{C}([0, \infty); L_{L,M}^\infty)$ if and only if*

1. *For fixed $s \in [0, \infty)$, $\Theta_{L,M}(s) \in L_{L,M}^\infty$.*

2. *For any $s_0 \in [0, \infty)$,*
$$\lim_{s \to s_0} d_{1,L,M}\left(\Theta_{L,M}(s), \Theta_{L,M}(s_0)\right) = 0,$$
*where $d_{1,L,M}$ is defined in Definition D.1.*

*The metric in $\mathcal{C}([0, \infty); L_{L,M}^\infty)$ is defined by*

$$d_{2,L,M}\left(\Theta_{L,M}, \widetilde{\Theta}_{L,M}\right) = \sup_s d_{1,L,M}(\Theta_{L,M}(s), \widetilde{\Theta}_{L,M}(s)).$$

Theorem A.3 is to say that the solution to (3) is unique in $\mathcal{C}([0,\infty); L^\infty_{L,M})$ if $\Theta_{L,M}(0) \in L^\infty_{L,M}$. Before proving the theorem, prepare some a-priori estimates of $Z_{\Theta_{L,M}}$ and $p_{\Theta_{L,M}}$.

**Lemma D.2** *Suppose that Assumption 4.1 holds and that $x$ is in the support of $\mu$. Let*

$$\Theta_{L,M} = \{\theta_{l,m}\}_{l=0,m=1}^{L-1,M}, \quad \text{and} \quad \widetilde{\Theta}_{L,M} = \left\{\widetilde{\theta}_{l,m}\right\}_{l=0,m=1}^{L-1,M},$$

*and denote*

$$\mathcal{L}_{\Theta_{L,M}} = \frac{1}{LM} \sum_{l=0}^{L-1} \sum_{m=1}^{M} |\theta_{l,m}|^2,$$

$$\mathcal{L}_{\widetilde{\Theta}_{L,M}} = \frac{1}{LM} \sum_{l=0}^{L-1} \sum_{m=1}^{M} |\widetilde{\theta}_{l,m}|^2,$$

$$\mathcal{R}_{L,M} = \sup_{l,m} \left\{ |\theta_{l,m}|, \left|\widetilde{\theta}_{l,m}\right| \right\}.$$

*Then for $0 \leq l \leq L - 1$, we have the following properties:*

- *Boundedness in $Z_{\Theta_{L,M}}$:*

$$\left| Z_{\Theta_{L,M}}(l+1;x) \right| \leq C(\mathcal{L}_{\Theta_{L,M}}), \tag{73}$$

- *Lipschitz in $Z_{\Theta_{L,M}}$:*

$$\left| Z_{\Theta_{L,M}}(l+1;x) - Z_{\widetilde{\Theta}_{L,M}}(l+1;x) \right| \leq C\left(\mathcal{L}_{\Theta_{L,M}}, \mathcal{L}_{\widetilde{\Theta}_{L,M}}\right) d_{1,L,M}\left(\Theta_{L,M}, \widetilde{\Theta}_{L,M}\right), \tag{74}$$

- *Boundedness in $p_{\Theta_{L,M}}$:*

$$\left| p_{\Theta_{L,M}}(l;x) \right| \leq C(\mathcal{L}_{\Theta_{L,M}}), \tag{75}$$

- *Lipschitz in $p_{\Theta_{L,M}}$:*

$$\left| p_{\Theta_{L,M}}(l;x) - p_{\widetilde{\Theta}_{L,M}}(l;x) \right| \leq C\left(\mathcal{R}_{L,M}\right) d_{1,L,M}\left(\Theta_{L,M}, \widetilde{\Theta}_{L,M}\right). \tag{76}$$

**Proof** From (1) and (12) we obtain

$$\left(\left| Z_{\Theta_{L,M}}(l+1;x) \right| + 1\right) \leq C_1 \left(1 + \frac{1}{LM} \sum_{m=1}^{M} (|\theta_{l,m}|^2 + 1)\right) \left(\left| Z_{\Theta_{L,M}}(l;x) \right| + 1\right)$$

$$\leq C_1 \exp\left(\frac{1}{LM} \sum_{m=1}^{M} (|\theta_{l,m}|^2 + 1)\right) \left(\left| Z_{\Theta_{L,M}}(l;x) \right| + 1\right),$$

which proves (73) by iteration on $l$.

From (13) and (73) we obtain

$$\frac{1}{ML} \sum_{m=1}^{M} |\partial_z f\left(Z_{\Theta_{L,M}}(l+1;x), \theta_{l,m}\right)| \leq \frac{C(\mathcal{L}_{\Theta_{L,M}})}{ML} \sum_{m=1}^{M} (|\theta_{l,m}|^2 + 1),$$

which by (72) implies

$$|p_{\Theta_{L,M}}(l;x)| \leq \left(1 + \frac{C(\mathcal{L}_{\Theta_{L,M}})}{ML} \sum_{m=1}^{M} (|\theta_{l,m}|^2 + 1)\right) |p_{\Theta_{L,M}}(l+1;x)|.$$

From this bound, together with $|p_{\Theta_{L,M}}(x, L-1)| \leq C|Z_{\Theta_{L,M}}(L;x)| \leq C(\mathcal{L}_{\Theta_{L,M}})$, we prove (75) by iteration on $l$.

To prove (74), we subtract the two updating formulas and split the estimate to obtain

$$
\left| Z_{\Theta_{L,M}}(l+1;x) - Z_{\widetilde{\Theta}_{L,M}}(l+1;x) \right|
$$

$$
\leq \left( 1 + \frac{C(\mathcal{L}_{\Theta_{L,M}})}{ML} \sum_{m=1}^{M} (|\theta_{l,m}|^2 + 1) \right) \left| Z_{\Theta_{L,M}}(l;x) - Z_{\widetilde{\Theta}_{L,M}}(l;x) \right|
$$

$$
+ \frac{C(\mathcal{L}_{\Theta_{L,M}}, \mathcal{L}_{\widetilde{\Theta}_{L,M}})}{L} \left( \frac{1}{M} \sum_{m=1}^{M} (|\theta_{l,m}|^2 + |\widetilde{\theta}_{l,m}|^2 + 1) \right) d_{1,L,M}\left( \Theta_{L,M}, \widetilde{\Theta}_{L,M} \right),
$$

where we use (1) together with the bounds (13), and (73). Noting that $|Z_{\Theta_{L,M}}(0;x) - Z_{\widetilde{\Theta}_{L,M}}(0;x)| = 0$, we prove (74) by iteration on $l$.

Finally, for (76), we subtract two equations in the form of (72), and use (72)-(75) together with Lipschitz continuity to obtain

$$
\left| p_{\Theta_{L,M}}(l;x) - p_{\widetilde{\Theta}_{L,M}}(l;x) \right|
$$

$$
\leq \left| (p_{\Theta_{L,M}}(l+1;x) - p_{\widetilde{\Theta}_{L,M}}(l+1;x))^\top \left( I + \frac{1}{ML} \sum_{m=1}^{M} \partial_z f\left( Z_{\Theta_{L,M}}(l+1;x), \theta_{l+1,m} \right) \right) \right|
$$

$$
+ \left| p_{\widetilde{\Theta}_{L,M}}^\top(l+1;x) \left( \frac{1}{ML} \sum_{m=1}^{M} \left( \partial_z f\left( Z_{\Theta_{L,M}}(l+1;x), \theta_{l+1,i} \right) - \partial_z f\left( Z_{\widetilde{\Theta}_{L,M}}(l+1;x), \widetilde{\theta}_{l+1,m} \right) \right) \right) \right|
$$

$$
\leq \left( 1 + \frac{C(\mathcal{R}_{L,M})}{ML} \sum_{m=1}^{M} (|\theta_{l,m}|^2 + 1) \right) \left| p_{\Theta_{L,M}}(l+1;x) - p_{\widetilde{\Theta}_{L,M}}(l+1;x) \right|
$$

$$
+ \frac{C(\mathcal{R}_{L,M})}{L} d_{1,L,M}\left( \Theta_{L,M}, \widetilde{\Theta}_{L,M} \right).
$$

(77)

The initial data is also controlled, as follows:

$$
|p_{\Theta_{L,M}}(L-1;x) - p_{\widetilde{\Theta}_{L,M}}(L-1;x)| \leq C|Z_{\Theta_{L,M}}(L;x) - Z_{\widetilde{\Theta}_{L,M}}(L;x)|
$$

$$
\leq C(\mathcal{R}_{L,M}) d_{1,L,M}\left( \Theta_{L,M}, \widetilde{\Theta}_{L,M} \right).
$$

By combining this with (77), we prove (76) by iteration on $l$. ∎

Lemma D.2 resembles Theorem A.2 and Lemma C.1. These estimates allow us to prove Theorem A.3. Since the proof strategy is exactly the same, we omit details. Essentially we define a map

$$
\widetilde{\Theta}(s) = \mathcal{T}_S^{L,M}(\Theta'(s)) : \mathcal{C}([0,\infty); L^\infty_{L,M}) \to , \mathcal{C}([0,\infty); L^\infty_{L,M}),
$$

where $\widetilde{\Theta}(s)$ solves:

$$
\frac{d\widetilde{\Theta}(s)}{ds} = -ML \nabla_\Theta E(\Theta'(s)), \quad \text{for } s \geq 0,
$$

where $\Theta$ defines the forcing term. The estimates above provide all the ingredients to show the map is well-defined, and for a small enough $S$, the map is also contracting, leading to the uniqueness of the solution to (3). Similar to Lemma D.1, one can also show $\frac{dE}{ds} = -ML|\nabla_\Theta E(\Theta_{L,M})|^2$, improving the estimates and removing the constants' dependence on the initial guess. This extends the local solution to the global one, as done in Step 3 for the continuous case.

# E  PROOF OF THEOREM 5.1

Theorem 5.1 links the cost defined by $\Theta_{L,M}(s)$ with that defined by $\rho(\theta, t, s)$ for all $s$. The continuous and mean-field limits are obtained, with both $L$ and $M$ sent to infinity. We decompose this result into two parts, discussing mean-field and continuous limits separately.

We start with the full definition of "limit-admissible" for a distribution $\rho$.

**Definition E.1** *For an admissible $\rho(\theta, t)$, we say $\rho(\theta, t)$ is* limit-admissible *if the average of a large number of particle presentations is bounded and Lipschitz with high probability. That is, for an admissible $\rho(\theta, t)$, there are two constants $C_3$ and $C_4$, both greater than $\sup_{t \in [0,1]} \int_{\mathbb{R}^k} |\theta|^2 d\rho(\theta, t)$ such that, for any $M$ stochastic process presentation $\{\theta_m(t)\}_{m=1}^M$ that are i.i.d. drawn from $\rho(\theta, t)$, the following properties are satisfied for any $\eta > 0$ and $M > \frac{C_3}{\eta}$:*

1. *Second moment boundedness in time:*

$$\mathbb{P}\left(\sup_{t \in [0,1]} \frac{1}{M} \sum_{m=1}^M |\theta_m(t)|^2 \leq C_4\right) \geq 1 - \eta. \tag{78}$$

2. *For all $L > 0$, we have*

$$\mathbb{P}\left(\frac{1}{M} \sum_{l=0}^{L-1} \sum_{m=1}^M \int_{\frac{l}{L}}^{\frac{l+1}{L}} \left|\theta_m(t) - \theta_m\left(\frac{l}{L}\right)\right|^2 \mathrm{d}t \leq \frac{C_4}{L^2}\right) \geq 1 - \eta. \tag{79}$$

We now state the two theorems that play complementary parts in Theorem 5.1. The first theorem addresses the limit in $M$ under the assumption that $L = \infty$. This is the mean-field part of the analysis.

**Theorem E.1** *Let Assumptions 4.1 and 4.2 hold with some $0 < k_1 \leq k$. Assume that $\rho_{\mathrm{ini}}(\theta, t)$ is limit-admissible and $\mathrm{supp}_\theta(\rho_{\mathrm{ini}}(\theta, t)) \subset \{\theta | \|\theta_{[1]}\| \leq R\}$ with some $R > 0$ for all $t \in [0, 1]$. Suppose that $\{\theta_m(0; t)\}_{m=1}^M$ are i.i.d drawn from $\rho_{\mathrm{ini}}(\theta, t)$. Suppose in addition that*

- *$\rho(\theta, t, s)$ solves (9) with the initial condition $\rho_{\mathrm{ini}}(\theta, t)$, and*

- *$\theta_m(s; t)$ solves (6) with the initial condition $\theta_m(0; t)$.*

*Then for any $\epsilon, \eta, S > 0$, there exists a constant $C(\rho_{\mathrm{ini}}(\theta, t), S) > 0$ depending on $\rho_{\mathrm{ini}}(\theta, t)$, $S$ such that when $M > \frac{C(\rho_{\mathrm{ini}}(\theta, t), S)}{\epsilon^2 \eta}$, we have*

$$\mathbb{P}\left(|E(\Theta(s; \cdot)) - E(\rho(\cdot, \cdot, s))| \leq \epsilon\right) \geq 1 - \eta, \quad \forall s < S.$$

**Proof** See Appendix F. ∎

The conclusion of this result suggests that for a $1 - \eta$ confidence of an $\epsilon$ accuracy, $M$ grows polynomially with respect to $1/\epsilon$ and $1/\eta$.

The second result considers the convergence of the parameter configuration for the discrete ResNet (1) to that for the continuous ResNet (4) as $L \to \infty$. This is the continuous-limit part of the analysis.

**Theorem E.2** *Let Assumptions 4.1 and 4.2 hold with some $0 < k_1 \leq k$. Assume that $\rho_{\mathrm{ini}}(\theta, t)$ is limit-admissible and $\mathrm{supp}_\theta(\rho_{\mathrm{ini}}(\theta, t)) \subset \{\theta | \|\theta_{[1]}\| \leq R\}$ with some $R > 0$ for all $t \in [0, 1]$. Suppose that $\{\theta_m(0; t)\}_{m=1}^M$ are i.i.d drawn from $\rho_{\mathrm{ini}}(\theta, t)$. Suppose in addition that*

- *$\theta_m(s; t)$ solves (6) with initial condition $\theta_m(0; t)$,*

- *$\theta_{l,m}(s)$ solves (3) with initial condition $\theta_m\left(0; \frac{l}{L}\right)$.*

*Then for any $\epsilon, \eta, S > 0$, there exists a constant $C(\rho_{\mathrm{ini}}(\theta, t), S) > 0$ depending on $\rho_{\mathrm{ini}}(\theta, t)$, $S$ such that when $M \geq \frac{C(\rho_{\mathrm{ini}}(\theta, t), S)}{\eta}$ and $L \geq \frac{C(\rho_{\mathrm{ini}}(\theta, t), S)}{\epsilon}$, we have for all $s < S$ that*

$$\mathbb{P}\left(|E(\Theta(s; \cdot)) - E(\Theta_{L,M}(s))| \leq \epsilon\right) \geq 1 - \eta.$$

**Proof** See Appendix G. ∎

This theorem shows that when the width is large enough, then with high probability, in the whole training process with $s < S$, the difference between the loss functions defined by the discrete ResNet and its continuous counterpart decreases to 0 as $L \to \infty$.

## F  CONVERGENCE TO THE MEAN-FIELD PDE

This section is dedicated to mean-field analysis and the proof of Theorem E.1. The intuition of this theorem is largely aligned with many other mean-field results, as demonstrated in (Ding et al., 2021).

As argued in Section 5, to show "equivalence" between (6) and (9), we can test them on the same smooth function $h(\theta)$. Testing (9) on amounts to multiplying $h$ on both sides of the equation by $h$. From integration by parts we have

$$\frac{\mathrm{d}}{\mathrm{d}s} \int_{\mathbb{R}^k} h \, \mathrm{d}\rho(\theta) = -\int_{\mathbb{R}^k} \nabla_\theta h \nabla_\theta \frac{\delta E(\rho(s))}{\delta \rho} \, \mathrm{d}\rho \,,$$

that is,

$$\frac{\mathrm{d}}{\mathrm{d}s} \mathbb{E}(h) = \mathbb{E}\left( \nabla_\theta h \nabla_\theta \frac{\delta E(\rho(s))}{\delta \rho} \right) \,.$$

To test (6) on $h$, we let $\rho = \frac{1}{M} \sum_{m=1}^M \delta_{\theta_m}$ and obtain

$$\frac{\mathrm{d}}{\mathrm{d}s} \mathbb{E}(h) = \frac{1}{M} \sum_{m=1}^M \nabla_\theta h(\theta_m) \frac{\mathrm{d}}{\mathrm{d}s} \theta_m = -\sum_{m=1}^M \nabla_\theta h(\theta_m) \frac{\delta E}{\delta \theta_m} \,.$$

We see that (9) and (6) are equivalent when tested by $h$, if and only if the right hand sides of the two equations above are the same, that is,

$$M \frac{\delta E}{\delta \theta_m} = \nabla_\theta \frac{\delta E(\rho)}{\delta \rho}(\theta_m, t) \,. \tag{80}$$

This claim can be established from the definitions of the Fréchet derivatives for $\frac{\delta E(\rho)}{\delta \rho}$ and $\frac{\delta E}{\delta \theta_m}$; see (Ding et al., 2021, Lemma 33).

To give a quantitative estimate on how quickly (6) converges to of (9), we utilize the particle method, a classical strategy for the mean-field limit. We sketch the proof here and will it more rigorous in the following subsections. We make use of two particle systems. In one system, the particles evolve themselves, while in the second, the particles are moved forward according to the underlying field constructed by the limit. In our situation, the former particle system consists of the $M$ stochastic processes $\{\theta_m(s;t)\}$ that descend according to $E(\Theta(s;\cdot))$. The latter particle system will be termed $\widetilde{\Theta}(s;t) = \{\widetilde{\theta}_m(s;t)\}_{m=1}^M$; it descends according to $E(\rho(\cdot,\cdot,s))$, the limiting cost function. Essentially, we prove

$$E(\Theta(s;\cdot)) \approx E\left( \widetilde{\Theta}(s;t) \right) \approx E(\rho(\cdot,\cdot,s)) \,.$$

The latter approximation arises roughly from the law of large numbers, but the former needs to be proved rigorously by tracing the two different evolving ODEs.

To be more specific, let $\rho(\theta, t, s)$ be the solution to (9) with admissible initial conditions $\rho_{\mathrm{ini}}(\theta, t)$, and let $\theta_m(s;t)$ be the solution to (6) with initial conditions $\{\theta_m(t,0)\}_{m=1}^M$ that are $i.i.d$ drawn from $\rho_{\mathrm{ini}}(\theta, t)$. Using the definition of $E$ in (8), we have

$$|E(\rho(\cdot,\cdot,s)) - E(\Theta(s;\cdot))|$$

$$\leq \mathbb{E}_{x\sim\mu} \left[ \left| \frac{1}{2} \left( g(Z_{\rho(s)}(1;x)) - y(x) \right)^2 - \frac{1}{2} \left( g(Z_{\Theta(s)}(1;x)) - y(x) \right)^2 \right| \right]$$

$$\leq \mathbb{E}_{x\sim\mu} \left( \left| g(Z_{\rho(s)}(1;x)) - g(Z_{\Theta(s)}(1;x)) \right| \left( \left| g(Z_{\rho(s)}(1;x)) + g(Z_{\Theta(s)}(1;x)) \right| + |y(x)| \right) \right) \tag{81}$$

$$\leq C(\mathcal{L}_s) \left| g(Z_{\rho(s)}(1;x)) - g(Z_{\Theta(s)}(1;x)) \right|$$

$$\leq C(\mathcal{L}_s) \left| Z_{\rho(s)}(1;x) - Z_{\Theta(s)}(1;x) \right| \,,$$

where $\mathcal{L}_s = \max \left\{ \int_0^1 \int_{\mathbb{R}^k} |\theta|^2 \, \mathrm{d}\rho(\theta, t, s) \, \mathrm{d}t, \frac{1}{M} \int_0^1 \sum_{m=1}^M |\theta_m(s;t)|^2 \, \mathrm{d}t \right\}$. In this derivation, we used the Lipschitz property of $g$, the boundedness of $y$ (required in Assumptions 4.1-4.2), and the boundedness of $Z_\rho$ and $Z_{\Theta(s)}$. Boundedness of $Z_\rho$ was shown in Theorem A.2, while the bound for $Z_{\Theta(s)}$ will be addressed in Lemma F.3. The constant $C$ depends on the support of $\rho_{\mathrm{ini}}$, as well as on

$s$ and the Lipschitz constant of $g$. It follows from (81) that to control $E(\rho(\cdot, \cdot, s)) - E(\Theta(s; \cdot))$, we need to control

$$\left| Z_{\rho(s)}(1; x) - Z_{\Theta(s)}(1; x) \right|. \tag{82}$$

To do so, we employ the particle method and invent a new particle system.

According to (80), we can reformulate the original particle system as

$$\frac{d\theta_m(s; t)}{ds} = -\nabla_\theta \frac{\delta E(\rho^{\mathrm{dis}}(s))}{\delta \rho}(\theta_m(s; t)), \quad \forall (t, s) \in [0, 1] \times [0, \infty), \tag{83}$$

where we denote

$$\rho^{\mathrm{dis}}(\theta, t, s) = \frac{1}{M} \sum_{m=1}^{M} \delta_{\theta_m(s;t)}(\theta). \tag{84}$$

We invent a new system that follows the underlying flow governed by the limit. Define $\widetilde{\Theta}(s) = \{\widetilde{\theta}_m(s; t)\}_{m=1}^{M}$, where $\widetilde{\theta}_m$ solves

$$\frac{d\widetilde{\theta}_m(s; t)}{ds} = -\nabla_\theta \frac{\delta E(\rho(s))}{\delta \rho} \left( \widetilde{\theta}_m(s; t) \right), \quad \forall (t, s) \in [0, 1] \times [0, \infty), \tag{85}$$

with initial condition

$$\widetilde{\theta}_m(0; t) = \theta_m(0; t). \tag{86}$$

As a consequence, we have $\widetilde{\theta}_m(s; t) \sim \rho(\theta, t, s)$ for all $(t, s)$. The corresponding ensemble distribution is

$$\widetilde{\rho}^{\mathrm{dis}}(\theta, t, s) = \frac{1}{M} \sum_{m=1}^{M} \delta_{\widetilde{\theta}_m(s;t)}(\theta). \tag{87}$$

Now we have available particle system $\Theta(s) = \{\theta_m(s)\}$, a newly invented particle system $\widetilde{\Theta}(s) = \{\widetilde{\theta}_m(s)\}$ and the mean-field flow $\rho$. Accordingly, there are three versions of $Z$: $Z_{\rho(s)}$ that solves (7) using $\rho(s)$ and $Z_{\Theta(s)}$ and $Z_{\widetilde{\Theta}(s)}$ that solve (4) using $\Theta(s)$ and $\widetilde{\Theta}(s)$, respectively. We use the following relabelling for convenience:

$$Z_s = Z_{\rho(s)}, \quad Z_s^{\mathrm{dis}} = Z_{\Theta(s)}, \quad \widetilde{Z}_s^{\mathrm{dis}} = Z_{\widetilde{\Theta}(s)}. \tag{88}$$

Similarly, there are three sets of $p$: $p_{\rho(s)}, p_{\rho^{\mathrm{dis}}(s)},$ and $p_{\widetilde{\rho}^{\mathrm{dis}}(s)}$ that solve (11) using $\rho(s), \rho^{\mathrm{dis}}(s),$ and $\widetilde{\rho}^{\mathrm{dis}}(s)$, respectively. We relabel similarly to (88) and write

$$p_s = p_{\rho(s)}, \quad p_s^{\mathrm{dis}} = p_{\rho^{\mathrm{dis}}(s)}, \quad \widetilde{p}_s^{\mathrm{dis}} = p_{\widetilde{\rho}^{\mathrm{dis}}(s)}. \tag{89}$$

Since $\widetilde{\Theta}(s)$ serves as a bridge, we translate the control of (82) to:

$$\left| Z_s(t; x) - Z_s^{\mathrm{dis}}(t; x) \right| \leq \left| Z_s(t; x) - \widetilde{Z}_s^{\mathrm{dis}}(t; x) \right| + \left| \widetilde{Z}_s^{\mathrm{dis}}(t; x) - Z_s^{\mathrm{dis}}(t; x) \right|. \tag{90}$$

Bounding $\left| Z_s(t; x) - \widetilde{Z}_s^{\mathrm{dis}}(t; x) \right|$ can be done using the law of large numbers. Bounding $\left| \widetilde{Z}_s^{\mathrm{dis}}(t; x) - Z_s^{\mathrm{dis}}(t; x) \right|$ translates to controlling $\sum_m \int_s^1 |\theta_m(s; t) - \widetilde{\theta}_m(s; t)| \, dt$, for which we will evaluate the difference between equations (83) and (85). Since these two equations have the same initial data, the difference between $\theta_m$ and $\widetilde{\theta}_m$ can then be controlled when the right-hand side forcing terms are close.

This approach divides the proof naturally into two components. In Section F.1, we give the rigorous bound of (90), while in Section F.2, we trace the evolution of the difference $\sum_m \int_0^1 |\theta_m(s; t) - \widetilde{\theta}_m(s; t)| \, dt$ in $s$, thus finalizing the proof for Theorem E.1.

### F.1 STABILITY IN THE MEAN-FIELD REGIME

Here we discuss control of the two terms on the right-hand side of (90). Recall that $\Theta(s)$ and $\widetilde{\Theta}(s)$ satisfy (83) and (85), and the two corresponding ensemble distribution are defined in (84) and (87), respectively. For any $S > 0$, define

$$
\mathcal{L}_S^{\sup} = \sup_{0 \le t \le 1, 0 \le s \le S} \left\{ \int_{\mathbb{R}^k} |\theta|^2 \, \mathrm{d}\rho(\theta, t, s), \frac{1}{M} \sum_{i=1}^M |\theta_m(s;t)|^2, \frac{1}{M} \sum_{i=1}^M |\widetilde{\theta}_m(s;t)|^2 \right\}
$$

$$
\mathcal{R}_S = \inf_{r > 0} \left\{ \mathrm{supp}(\rho(\theta, t, s)) \cup \{\theta_m(s;t)\}_{m=1}^M \subset \{\theta| |\theta_{[1]}| < r\}, \ \forall (t, s) \in [0, 1] \times [0, S] \right\},
$$
(91)

We have the following lemma:

**Lemma F.1** *For every fixed $s$, let $Z_s$ and $Z_s^{\mathrm{dis}}$ be as defined in* (88)*. Then there exists a constant $C(\mathcal{L}_s^{\sup})$ such that for all $t \in [0, 1], s \in [0, \infty)$, we have*

$$
\left| Z_s(t; x) - Z_s^{\mathrm{dis}}(t; x) \right|
$$

$$
\le C(\mathcal{L}_s^{\sup}) \left( \frac{1}{M} \sum_{m=1}^M \int_0^1 \left| \theta_m(s; \tau) - \widetilde{\theta}_m(s; \tau) \right|^2 \mathrm{d}\tau \right)^{1/2}
$$
(92)

$$
+ C(\mathcal{L}_s^{\sup}) \left( \int_0^1 \left| \int_{\mathbb{R}^k} f\left( Z_s(\tau; x), \theta \right) \mathrm{d}(\rho(\theta, \tau, s) - \widetilde{\rho}^{\mathrm{dis}}(\theta, \tau, s)) \right|^2 \mathrm{d}\tau \right)^{1/2}.
$$

**Proof** Since the statement holds for a fixed $s$, we eliminate all $s$ dependence in all calculations in the proof, for conciseness.

Recalling the definitions in (88), we denote

$$
\widetilde{\Delta}(t; x) = Z_s(t; x) - \widetilde{Z}_s^{\mathrm{dis}}(t; x), \quad \Delta(t; x) = \widetilde{Z}_s^{\mathrm{dis}}(t; x) - Z_s^{\mathrm{dis}}(t; x).
$$

It follows from the triangle inequality that

$$
\left| Z_s(t; x) - Z_s^{\mathrm{dis}}(t; x) \right| \le \left| \widetilde{\Delta}(t; x) \right| + |\Delta(t; x)|.
$$

We now bound these two terms. We first apply the same argument as in the proof of Theorem A.2 (see (26) in Appendix B) to obtain

$$
\frac{\mathrm{d} \left| \widetilde{\Delta}(t; x) \right|^2}{\mathrm{d}t} \le C(\mathcal{L}_s^{\sup}) \left| \widetilde{\Delta}(t; x) \right|^2 + \left| \int_{\mathbb{R}^k} f\left( Z_s(t; x), \theta \right) \mathrm{d}(\rho(\theta, t, s) - \widetilde{\rho}^{\mathrm{dis}}(\theta, t, s)) \right|^2.
$$

Using the Grönwall inequality and the fact that $\left| \widetilde{\Delta}(0; x) \right| = 0$, we have

$$
\left| \widetilde{\Delta}(t; x) \right| \le C(\mathcal{L}_s^{\sup}) \left( \int_0^1 \left| \int_{\mathbb{R}^k} f\left( Z_s(t; x), \theta \right) \mathrm{d}(\rho(\theta, t, s) - \widetilde{\rho}^{\mathrm{dis}}(\theta, t, s)) \right|^2 \mathrm{d}\tau \right)^{1/2}.
$$
(93)

for all $t \in [0, 1]$. Similarly, to bound $\Delta(t; x)$, we have

$$
\frac{\mathrm{d} |\Delta(t; x)|^2}{\mathrm{d}t} \le C(\mathcal{L}_s^{\sup}) |\Delta(t; x)|^2 + \left| \int_{\mathbb{R}^k} f\left( \widetilde{Z}_s^{\mathrm{dis}}(t; x), \theta \right) \mathrm{d}(\rho^{\mathrm{dis}}(\theta, t, s) - \widetilde{\rho}^{\mathrm{dis}}(\theta, t, s)) \right|^2.
$$

From Assumption 4.1 and the fact that $\widetilde{Z}_{\mathrm{dis}}$ is bounded in Theorem A.2, we have

$$
\left| \int_{\mathbb{R}^k} f\left( \widetilde{Z}_s^{\mathrm{dis}}(t; x), \theta \right) \mathrm{d}(\rho^{\mathrm{dis}}(\theta, t, s) - \widetilde{\rho}^{\mathrm{dis}}(\theta, t, s)) \right|^2 \le C(\mathcal{L}_s^{\sup}) \left( \frac{1}{M} \sum_{m=1}^M \left| \theta_m(s; t) - \widetilde{\theta}_m(s; t) \right| \right)^2.
$$

similar to the proof of Theorem A.2 (see (28) in Appendix B).

Using $\Delta(0; x) = 0$, we apply Grönwall's inequality to obtain

$$|\Delta(t; x)| \leq C(\mathcal{L}_s^{\text{sup}}) \left( \frac{1}{M} \sum_{m=1}^{M} \int_0^1 \left| \theta_m(s; \tau) - \widetilde{\theta}_m(s; \tau) \right|^2 \, \mathrm{d}\tau \right)^{1/2}. \tag{94}$$

The result is obtained from adding (93) and (94). ∎

The difference in $p_s$ and $p_s^{\text{dis}}$:

**Lemma F.2** *For every fixed $s \in [0, \infty)$, let $p_s$ and $p_s^{\text{dis}}$ be defined in (89). There exists a constant $C(\mathcal{R}_s)$ with $\mathcal{R}_s, \mathcal{L}_s^{\text{sup}}$ defined in (91) such that for all $t \in [0, 1]$:*

$$\left| p_s(t; x) - p_s^{\text{dis}}(t; x) \right|$$

$$\leq C(\mathcal{R}_s, \mathcal{L}_s^{\text{sup}}) \left( \frac{1}{M} \sum_{m=1}^{M} \int_0^1 \left| \theta_m(s; \tau) - \widetilde{\theta}_m(s; \tau) \right|^2 \, \mathrm{d}\tau \right)^{1/2}$$

$$+ C(\mathcal{R}_s, \mathcal{L}_s^{\text{sup}}) \left( \int_0^1 \left| \int_{\mathbb{R}^k} f(Z_s(\tau; x), \theta) \, \mathrm{d}(\rho(\theta, \tau, s) - \widetilde{\rho}^{\text{dis}}(\theta, \tau, s)) \right|^2 \, \mathrm{d}\tau \right)^{1/2} \tag{95}$$

$$+ C(\mathcal{R}_s, \mathcal{L}_s^{\text{sup}}) \left( \int_0^1 \left| \int_{\mathbb{R}^k} \partial_z f(Z_s(\tau; x), \theta) \, \mathrm{d}(\rho(\theta, \tau, s) - \widetilde{\rho}^{\text{dis}}(\theta, \tau, s)) \right|^2 \, \mathrm{d}\tau \right)^{1/2}.$$

**Proof** As in the previous proof, we eliminate dependence on $s$ in some notation, for conciseness. Denoting $\Delta_p(t; x) = p_s(t; x) - p_s^{\text{dis}}(t; x)$, we recall (11) to have:

$$\frac{\mathrm{d}|\Delta_p(t; x)|^2}{\mathrm{d}t}$$

$$\leq 2 \left| \int_{\mathbb{R}^k} \partial_z f(Z_s(t; x), \theta) \, \mathrm{d}\rho(\theta, t, s) \right| |\Delta_p(t; x)|^2$$

$$\quad + 2|\Delta_p(t; x)| \, |p_s^{\text{dis}}| \left| \int_{\mathbb{R}^k} \partial_z f(Z_s(t; x), \theta) \, \mathrm{d}\rho(\theta, t, s) - \int_{\mathbb{R}^k} \partial_z f(Z_s^{\text{dis}}(t; x), \theta) \, \mathrm{d}\rho^{\text{dis}}(\theta, t, s) \right|$$

$$\leq C(\mathcal{L}_s^{\text{sup}})|\Delta_p(t; x)|^2$$

$$\quad + 2 \left| \int_{\mathbb{R}^k} \partial_z f(Z_s(t; x), \theta) \, \mathrm{d}\rho(\theta, t, s) - \int_{\mathbb{R}^k} \partial_z f(Z_s^{\text{dis}}(t; x), \theta) \, \mathrm{d}\rho^{\text{dis}}(\theta, t, s) \right|^2$$

$$\leq C(\mathcal{L}_s^{\text{sup}})|\Delta_p(t; x)|^2$$

$$\quad + 6 \left| \int_{\mathbb{R}^k} \partial_z f(Z_s(t; x), \theta) \, \mathrm{d}\rho(\theta, t, s) - \int_{\mathbb{R}^k} \partial_z f(Z_s(t; x), \theta) \, \mathrm{d}\widetilde{\rho}^{\text{dis}}(\theta, t, s) \right|^2$$

$$\quad + 6 \underbrace{\left| \int_{\mathbb{R}^k} \partial_z f(Z_s(t; x), \theta) \, \mathrm{d}\widetilde{\rho}^{\text{dis}}(\theta, t, s) - \int_{\mathbb{R}^k} \partial_z f(Z_s(t; x), \theta) \, \mathrm{d}\rho^{\text{dis}}(\theta, t, s) \right|^2}_{\text{(I)}}$$

$$\quad + 6 \underbrace{\left| \int_{\mathbb{R}^k} \partial_z f(Z_s(t; x), \theta) \, \mathrm{d}\rho^{\text{dis}}(\theta, t, s) - \int_{\mathbb{R}^k} \partial_z f(Z_s^{\text{dis}}(t; x), \theta) \, \mathrm{d}\rho^{\text{dis}}(\theta, t, s) \right|^2}_{\text{(II)}},$$

where we use (13) from Assumption 4.1 together with (18) and (31) in the second inequality. To bound (I) on the right-hand side, we recall the definition (84) and (87), and use Assumption 4.1 (14)

along with the boundedness of $Z$ (as shown in (18)) to obtain

$$
\begin{aligned}
\text{(I)} &\le \left( \frac{1}{M} \sum_{m=1}^{M} \left| \partial_z f(Z_s(t;x), \theta_m(s;t)) - \partial_z f(Z_s(t;x), \widetilde{\theta}_m(s;t)) \right| \right)^2 \\
&\le C(\mathcal{R}_s, \mathcal{L}_s^{\mathrm{sup}}) \left( \frac{1}{M} \sum_{m=1}^{M} |\theta_m(s;t) - \widetilde{\theta}_m(s;t)| \right)^2 \\
&\le C(\mathcal{R}_s, \mathcal{L}_s^{\mathrm{sup}}) \left( \frac{1}{M} \sum_{m=1}^{M} \left| \theta_m(s;t) - \widetilde{\theta}_m(s;t) \right|^2 \right),
\end{aligned}
\tag{96}
$$

where we use Hölder's inequality in the last inequality. For (II), we have

$$
\begin{aligned}
\text{(II)} &\le \left( \frac{1}{M} \sum_{m=1}^{M} \left| \partial_z f(Z_s(t;x), \theta_m(s;t)) - \partial_z f(Z_s^{\mathrm{dis}}(t;x), \theta_m(s;t)) \right| \right)^2 \\
&\le C(\mathcal{R}_s, \mathcal{L}_s^{\mathrm{sup}}) |Z_s(t;x) - Z_s^{\mathrm{dis}}(t;x)|^2.
\end{aligned}
\tag{97}
$$

By substituting (96) and (97) into the bound above, we obtain

$$
\begin{aligned}
&\frac{\mathrm{d}|\Delta_p(t;x)|^2}{\mathrm{d}t} \\
={}& C(\mathcal{L}_s^{\mathrm{sup}}) |\Delta_p(t;x)|^2 + C(\mathcal{R}_s, \mathcal{L}_s^{\mathrm{sup}}) \left( \frac{1}{M} \sum_{m=1}^{M} \left| \theta_m(s;t) - \widetilde{\theta}_m(s;t) \right|^2 + |Z_s(t;x) - Z_s^{\mathrm{dis}}(t;x)|^2 \right) \\
&+ 6 \left| \int_{\mathbb{R}^k} \partial_z f(Z_s(t;x), \theta) \, \mathrm{d}\rho(\theta, t, s) - \int_{\mathbb{R}^k} \partial_z f(Z_s(t;x), \theta) \, \mathrm{d}\widetilde{\rho}^{\mathrm{dis}}(\theta, t, s) \right|^2.
\end{aligned}
\tag{98}
$$

The "initial condition" for $p_s$ and $p_s^{\mathrm{dis}}$ yields

$$
|\Delta_p(1;x)| \le C(\mathcal{L}_s^{\mathrm{sup}}) |Z_s(1;x) - Z_s^{\mathrm{dis}}(1;x)|.
$$

The result is obtained when we substitute (92) into (98) and use the Grönwall's inequality. ∎

### F.2 PROOF OF THEOREM E.1

With the quantitative description of (90) presented in Lemma F.1, we complete the proof for Theorem E.1 in this section. Recall from (55) that

$$
\mathcal{R}_{\mathrm{ini}} = \inf_{r>0} \left\{ \mathrm{supp}(\rho_{\mathrm{ini}}(t)) \subset \left\{ \theta \middle| |\theta_{[1]}| < r \right\}, \ \forall t \in [0,1] \right\}, \quad \mathcal{L}_{\mathrm{ini}}^{\mathrm{sup}} = \sup_{0 \le t \le 1} \int_{\mathbb{R}^k} |\theta|^2 \, \mathrm{d}\rho_{\mathrm{ini}}(\theta, t)
$$

and note that $\mathcal{R}_{\mathrm{ini}} \ge |\theta_{m,[1]}(0;t)|$ for all $m, t$. Define

$$
\mathcal{L}_{\mathrm{ini}}^{\mathrm{dis,sup}} = \sup_{0 \le t \le 1} \frac{1}{M} \sum_{m=1}^{M} |\theta_m(0;t)|^2,
$$

We note that when $M$ is large, $\mathcal{L}_{\mathrm{ini}}^{\mathrm{dis,sup}}$ is close to $\mathcal{L}_{\mathrm{ini}}^{\mathrm{sup}}$ (which has no randomness) with high probability. We have the following lemma:

**Lemma F.3** *For a given $S > 0$, there exists a constant $C(S, \mathcal{R}_{\mathrm{ini}})$ such that for any $t \in [0,1]$, $s \in [0, S]$, we have*

$$
\begin{aligned}
\frac{1}{M} \sum_{m=1}^{M} |\theta_m(s;t)|^2 &\le C(S, \mathcal{R}_{\mathrm{ini}}, \mathcal{L}_{\mathrm{ini}}^{\mathrm{dis,sup}}), \\
\{\theta_m(s;t)\}_{m=1}^{M} &\subset \left\{ \theta \middle| |\theta_{[1]}| \le C(S, \mathcal{R}_{\mathrm{ini}}, \mathcal{L}_{\mathrm{ini}}^{\mathrm{dis,sup}}) \right\},
\end{aligned}
\tag{99}
$$

*Furthermore, for any $x$ with $|x| < R_\mu$ (as in Assumption 4.2, item 4), and any $s \in [0, S]$ and $t \in [0, 1]$, the ODE solution is bounded as follows:*

$$\left| Z_s^{\mathrm{dis}}(t; x) \right| \leq C(S, \mathcal{R}_{\mathrm{ini}}, \mathcal{L}_{\mathrm{ini}}^{\mathrm{dis,sup}}), \tag{100}$$

*while the following bound holds on $p_s^{\mathrm{dis}}$:*

$$\left| p_s^{\mathrm{dis}}(t; x) \right| \leq C(S, \mathcal{R}_{\mathrm{ini}}, \mathcal{L}_{\mathrm{ini}}^{\mathrm{dis,sup}}). \tag{101}$$

The bound (99) is a result of Corollary D.3. The bounds (100) and (101) are obtained using the same arguments as in Theorem A.2 and Lemma C.1.

The next lemma bounds the support and second moment of $\widetilde{\rho}^{\mathrm{dis}}(\theta, t, s)$.

**Lemma F.4** *Under conditions of Theorem E.1, for any $S > 0$, there is a constant $C(S, \mathcal{R}_{\mathrm{ini}}, \mathcal{L}_{\mathrm{ini}}^{\mathrm{dis,sup}})$ depending only on $\mathcal{R}_{\mathrm{ini}}, \mathcal{L}_{\mathrm{ini}}^{\mathrm{dis,sup}}, S$ such that for any $t \in [0, 1], s \in [0, S]$, we have*

$$\frac{1}{M} \sum_{m=1}^{M} |\widetilde{\theta}_m(s; t)|^2 \leq C(S, \mathcal{R}_{\mathrm{ini}}, \mathcal{L}_{\mathrm{ini}}^{\mathrm{dis,sup}}),$$
$$\{\widetilde{\theta}_m(s; t)\}_{m=1}^{M} \subset \{\theta | |\theta_{[1]}| \leq C(S, \mathcal{R}_{\mathrm{ini}}, \mathcal{L}_{\mathrm{ini}}^{\mathrm{dis,sup}})\}, \tag{102}$$

**Proof** Similar to the proof of Proposition D.1, we multiply (85) by $\widetilde{\theta}_m(s; t)$ on both sides and utilize the bound (41) from Lemma C.2, where $\mathcal{L}$ in (41) is replaced by $C(S, \mathcal{R}_{\mathrm{ini}}, \mathcal{L}_{\mathrm{ini}}^{\mathrm{dis,sup}})$ according to Corollary D.3. We thus obtain

$$|\widetilde{\theta}_m(s; t)| \leq C(S, \mathcal{R}_{\mathrm{ini}}, \mathcal{L}_{\mathrm{ini}}^{\mathrm{dis,sup}}) \left( |\widetilde{\theta}_m(0; t)| + 1 \right),$$

and

$$|\widetilde{\theta}_{m,[1]}(s; t)| \leq C(S, \mathcal{R}_{\mathrm{ini}}, \mathcal{L}_{\mathrm{ini}}^{\mathrm{dis,sup}}) \left( |\widetilde{\theta}_{m,[1]}(0; t)| + 1 \right)$$

which implies (102) by (86). ∎

Denote $\mathcal{L}_0^{\mathrm{sup}} = \max\{\mathcal{L}_{\mathrm{ini}}^{\mathrm{dis,sup}}, \mathcal{L}_{\mathrm{ini}}^{\mathrm{sup}}\}$. According to Definition E.1 (78), $\mathcal{L}_0^{\mathrm{sup}}$ should be bound by $C_4$ with high probability. We are now ready for the main proof of this section.

**Proof** [Proof of Theorem E.1] First, using Lemmas F.3, F.4, there is a constant $C(S, \mathcal{R}_{\mathrm{ini}}, \mathcal{L}_0^{\mathrm{sup}})$ depending only on $\mathcal{R}_{\mathrm{ini}}, \mathcal{L}_0^{\mathrm{sup}}, S$ such that for any $t \in [0, 1], s \in [0, S]$

$$\max \left\{ \int_{\mathbb{R}^k} |\theta|^2 \, \mathrm{d}\rho(\theta, t, s), \; \frac{1}{M} \sum_{m=1}^{M} \left| \widetilde{\theta}_m(s; t) \right|^2, \; \frac{1}{M} \sum_{m=1}^{M} |\theta_m(s; t)|^2 \right\} \leq C(S, \mathcal{R}_{\mathrm{ini}}, \mathcal{L}_0^{\mathrm{sup}}),$$

$$\mathrm{supp}(\rho(\theta, t, s)) \cup \{\theta_m(s; t)\}_{m=1}^{M} \cup \left\{ \widetilde{\theta}_m(s; t) \right\}_{m=1}^{M} \subset \{\theta | |\theta_{[1]}| \leq C(S, \mathcal{R}_{\mathrm{ini}}, \mathcal{L}_0^{\mathrm{sup}})\}, \tag{103}$$

Recalling (81), we have $\mathcal{L}_s \leq C(S, \mathcal{R}_{\mathrm{ini}}, \mathcal{L}_0^{\mathrm{sup}})$ and

$$|E(\rho(\cdot, \cdot, s)) - E(\Theta(s; \cdot))| \leq C(S, \mathcal{R}_{\mathrm{ini}}, \mathcal{L}_0^{\mathrm{sup}}) \left| Z_s(1; x) - Z_s^{\mathrm{dis}}(1; x) \right|. \tag{104}$$

Furthermore, according to Lemma F.1 (92), from (103)

$$\left| Z_s(t; x) - Z_s^{\mathrm{dis}}(t; x) \right|$$

$$\leq C(S, \mathcal{R}_{\mathrm{ini}}, \mathcal{L}_0^{\mathrm{sup}}) \left( \frac{1}{M} \sum_{m=1}^{M} \int_0^1 \left| \theta_m(s; \tau) - \widetilde{\theta}_m(s; \tau) \right|^2 \mathrm{d}\tau \right)^{1/2} \tag{105}$$

$$+ C(S, \mathcal{R}_{\mathrm{ini}}, \mathcal{L}_0^{\mathrm{sup}}) \left( \int_0^1 \left| \int_{\mathbb{R}^k} f\left(Z_s(\tau; x), \theta\right) \mathrm{d}(\rho(\theta, \tau, s) - \widetilde{\rho}^{\mathrm{dis}}(\theta, \tau, s)) \right|^2 \mathrm{d}\tau \right)^{1/2}.$$

The second term in this bound can be treated using the law of large numbers. We focus on controlling the first term.

Step 1: Estimating $\frac{1}{M}\sum_{m=1}^{M}\int_0^1\left|\theta_m(s;t)-\widetilde{\theta}_m(s;t)\right|^2\,\mathrm{d}t$. Defining

$$\Delta_{t,m}(s)=\theta_m(s;t)-\widetilde{\theta}_m(s;t)\,,$$

we note that $|\Delta_{t,m}(0)|=0$. By taking the difference of (83) and (85) and multiplying both sides by $\Delta_{t,m}(s)$, we obtain

$$
\begin{aligned}
&\frac{\mathrm{d}|\Delta_{t,m}(s)|^2}{\mathrm{d}s}\\
={}&-2\left\langle\Delta_{t,m}(s),\mathbb{E}_{x\sim\mu}\left(\partial_\theta f(Z_s(t;x),\widetilde{\theta}_m)p_s(t;x)-\partial_\theta f(Z_s^{\mathrm{dis}}(t;x),\theta_m)p_s^{\mathrm{dis}}(t;x)\right)\right\rangle\\
={}&-2\left\langle\Delta_{t,m}(s),\underbrace{\mathbb{E}_{x\sim\mu}\left(\partial_\theta f(Z_s(t;x),\widetilde{\theta}_m)p_s(t;x)-\partial_\theta f(Z_s(t;x),\widetilde{\theta}_m)p_s^{\mathrm{dis}}(t;x)\right)}_{\text{(I)}}\right\rangle\\
&-2\left\langle\Delta_{t,m}(s),\underbrace{\mathbb{E}_{x\sim\mu}\left(\partial_\theta f(Z_s(t;x),\widetilde{\theta}_m)p_s^{\mathrm{dis}}(t;x)-\partial_\theta f(Z_s^{\mathrm{dis}}(t;x),\theta_m)p_s^{\mathrm{dis}}(t;x)\right)}_{\text{(II)}}\right\rangle\,.
\end{aligned}
\tag{106}
$$

For (I), we have from the bounds of $Z_s$ in (18), respectively, that

$$|\text{(I)}|\le C(S,\mathcal{R}_{\mathrm{ini}},\mathcal{L}_0^{\mathrm{sup}})\left(|\widetilde{\theta}_m|+1\right)\mathbb{E}_{x\sim\mu}\left(\left|p_s(t;x)-p_s^{\mathrm{dis}}(t;x)\right|\right)\,,$$

which can be controlled using Lemma F.2 (95). For (II), we have

$$
\begin{aligned}
|\text{(II)}|\le{}&C(S,\mathcal{R}_{\mathrm{ini}},\mathcal{L}_0^{\mathrm{sup}})\mathbb{E}_{x\sim\mu}\left(\left|\partial_\theta f(Z_s(t;x),\widetilde{\theta}_m)-\partial_\theta f(Z_s^{\mathrm{dis}}(t;x),\theta_m)\right|\right)\\
\le{}&C(S,\mathcal{R}_{\mathrm{ini}},\mathcal{L}_0^{\mathrm{sup}})\left[\mathbb{E}_{x\sim\mu}\left(\left|Z_s(t;x)-Z_s^{\mathrm{dis}}(t;x)\right|\right)+\left|\widetilde{\theta}_m-\theta_m\right|\right]\,,
\end{aligned}
$$

where the first term can be controlled using Lemma F.1 (92). In both estimates, we used the property of $f$ in Assumption 4.1 and bounds on $Z$, $p$, $\theta_{m,[1]}$, and $\widetilde{\theta}_{m,[1]}$. By substituting these estimates into (106), we obtain

$$
\begin{aligned}
\frac{\mathrm{d}|\Delta_{t,m}(s)|^2}{\mathrm{d}s}\le{}&C(S,\mathcal{R}_{\mathrm{ini}},\mathcal{L}_0^{\mathrm{sup}})|\Delta_{t,m}(s)|^2\\
&+C(S,\mathcal{R}_{\mathrm{ini}},\mathcal{L}_0^{\mathrm{sup}})|\Delta_{t,m}(s)|\mathbb{E}_{x\sim\mu}\left(\left|Z_s(t;x)-Z_s^{\mathrm{dis}}(t;x)\right|\right)\\
&+C(S,\mathcal{R}_{\mathrm{ini}},\mathcal{L}_0^{\mathrm{sup}})|\Delta_{t,m}(s)|\left(|\widetilde{\theta}_m|+1\right)\mathbb{E}_{x\sim\mu}\left(\left|p_s(t;x)-p_s^{\mathrm{dis}}(t;x)\right|\right)\,,
\end{aligned}
$$

which implies that

$$
\begin{aligned}
&\frac{1}{M}\sum_{m=1}^{M}\frac{\mathrm{d}|\Delta_{t,m}(s)|^2}{\mathrm{d}s}\\
\le{}&C(S,\mathcal{R}_{\mathrm{ini}},\mathcal{L}_0^{\mathrm{sup}})\left(\frac{1}{M}\sum_{m=1}^{M}|\Delta_{t,m}(s)|^2\right)\\
&+C(S,\mathcal{R}_{\mathrm{ini}},\mathcal{L}_0^{\mathrm{sup}})\left(\frac{1}{M}\sum_{m=1}^{M}|\Delta_{t,m}(s)|\right)\mathbb{E}_{x\sim\mu}\left(\left|Z_s(t;x)-Z_s^{\mathrm{dis}}(t;x)\right|\right)\\
&+C(S,\mathcal{R}_{\mathrm{ini}},\mathcal{L}_0^{\mathrm{sup}})\left(\frac{1}{M}\sum_{m=1}^{M}|\Delta_{t,m}(s)|\left(|\widetilde{\theta}_m|+1\right)\right)\mathbb{E}_{x\sim\mu}\left(\left|p_s(t;x)-p_s^{\mathrm{dis}}(t;x)\right|\right)\\
\le{}&C(S,\mathcal{R}_{\mathrm{ini}},\mathcal{L}_0^{\mathrm{sup}})\left(\frac{1}{M}\sum_{m=1}^{M}|\Delta_{t,m}(s)|^2\right)+C(S,\mathcal{R}_{\mathrm{ini}},\mathcal{L}_0^{\mathrm{sup}})\mathbb{E}_{x\sim\mu}\left(\left|p_s(t;x)-p_s^{\mathrm{dis}}(t;x)\right|^2\right)\\
&+C(S,\mathcal{R}_{\mathrm{ini}},\mathcal{L}_0^{\mathrm{sup}})\mathbb{E}_{x\sim\mu}\left(\left|Z_s(t;x)-Z_s^{\mathrm{dis}}(t;x)\right|^2\right)\,,
\end{aligned}
\tag{107}
$$

where in the last inequality we use Hölder's inequality

$$
\left( \frac{1}{M} \sum_{m=1}^{M} |\Delta_{t,m}(s)| \left( |\widetilde{\theta}_m| + 1 \right) \right) \mathbb{E}_{x \sim \mu} \left( \left| p_s(t;x) - p_s^{\mathrm{dis}}(t;x) \right| \right)
$$

$$
\leq \left( \frac{1}{M} \sum_{m=1}^{M} |\Delta_{t,m}(s)|^2 \right)^{1/2} \left( \frac{1}{M} \sum_{m=1}^{M} \left( |\widetilde{\theta}_m| + 1 \right)^2 \right)^{1/2} \mathbb{E}_{x \sim \mu} \left( \left| p_s(t;x) - p_s^{\mathrm{dis}}(t;x) \right| \right)
$$

$$
\leq C(S, \mathcal{R}_{\mathrm{ini}}, \mathcal{L}_0^{\mathrm{sup}}) \left( \frac{1}{M} \sum_{m=1}^{M} |\Delta_{t,m}(s)|^2 \right)^{1/2} \mathbb{E}_{x \sim \mu} \left( \left| p_s(t;x) - p_s^{\mathrm{dis}}(t;x) \right| \right)
$$

$$
\leq C(S, \mathcal{R}_{\mathrm{ini}}, \mathcal{L}_0^{\mathrm{sup}}) \left[ \left( \frac{1}{M} \sum_{m=1}^{M} |\Delta_{t,m}(s)|^2 \right) + \mathbb{E}_{x \sim \mu} \left( \left| p_s(t;x) - p_s^{\mathrm{dis}}(t;x) \right|^2 \right) \right]
$$

Noting the estimate in Lemma F.1-F.2, we obtain

$$
\frac{\mathrm{d} \frac{1}{M} \sum_{m=1}^{M} \int_0^1 |\Delta_{t,m}(s)|^2 \, \mathrm{d}t}{\mathrm{d}s}
$$

$$
\leq C(S, \mathcal{R}_{\mathrm{ini}}, \mathcal{L}_0^{\mathrm{sup}}) \left( \frac{1}{M} \sum_{m=1}^{M} \int_0^1 |\Delta_{t,m}(s)|^2 \, \mathrm{d}t \right)
$$

$$
+ C(S, \mathcal{R}_{\mathrm{ini}}, \mathcal{L}_0^{\mathrm{sup}}) \mathbb{E}_{x \sim \mu} \left( \int_0^1 \left| \int_{\mathbb{R}^k} f\left( Z_s(\tau;x), \theta \right) \, \mathrm{d}(-\widetilde{\rho}^{\mathrm{dis}}(\theta, \tau, s)) \right|^2 \, \mathrm{d}\tau \right)
$$

$$
+ C(S, \mathcal{R}_{\mathrm{ini}}, \mathcal{L}_0^{\mathrm{sup}}) \mathbb{E}_{x \sim \mu} \left( \int_0^1 \left| \int_{\mathbb{R}^k} \partial_z f\left( Z_s(\tau;x), \theta \right) \, \mathrm{d}(-\widetilde{\rho}^{\mathrm{dis}}(\theta, \tau, s)) \right|^2 \, \mathrm{d}\tau \right),
$$

which implies, using Grönwall's inequality, that

$$
\frac{1}{M} \sum_{m=1}^{M} \int_0^1 |\Delta_{t,m}(s)|^2 \, \mathrm{d}t
$$

$$
\leq C(S, \mathcal{R}_{\mathrm{ini}}, \mathcal{L}_0^{\mathrm{sup}}) \mathbb{E}_{x \sim \mu} \left( \int_0^S \int_0^1 \left| \int_{\mathbb{R}^k} f\left( Z_s(t;x), \theta \right) \, \mathrm{d}(\rho(\theta, \tau, s) - \widetilde{\rho}^{\mathrm{dis}}(\theta, \tau, s)) \right|^2 \, \mathrm{d}\tau \, \mathrm{d}s \right)
$$

$$
+ C(S, \mathcal{R}_{\mathrm{ini}}, \mathcal{L}_0^{\mathrm{sup}}) \mathbb{E}_{x \sim \mu} \left( \int_0^S \int_0^1 \left| \int_{\mathbb{R}^k} \partial_z f\left( Z_s(t;x), \theta \right) \, \mathrm{d}(\rho(\theta, \tau, s) - \widetilde{\rho}^{\mathrm{dis}}(\theta, \tau, s)) \right|^2 \, \mathrm{d}\tau \, \mathrm{d}s \right),
$$

$$\tag{108}$$

where we use $|\Delta_{t,m}(0)| = 0$.

Step 2: Completing the proof. By substituting (108) into (92), noticing $\mathcal{R}_s \leq C(\mathcal{R}_{\text{ini}}, S)$, we obtain

$$
\left| Z_s(t;x) - Z_s^{\text{dis}}(t;x) \right|
$$

$$
\leq C(S, \mathcal{R}_{\text{ini}}, \mathcal{L}_0^{\text{sup}}) \left( \underbrace{\mathbb{E}_{x \sim \mu} \left( \int_0^S \int_0^1 \left| \int_{\mathbb{R}^k} f\left(Z_s(\tau;x), \theta\right) \, \mathrm{d}(\rho(\theta, \tau, s) - \widetilde{\rho}^{\text{dis}}(\theta, \tau, s)) \right|^2 \mathrm{d}\tau \, \mathrm{d}s \right)}_{\text{(I)}} \right)^{1/2}
$$

$$
+ C(S, \mathcal{R}_{\text{ini}}, \mathcal{L}_0^{\text{sup}}) \left( \underbrace{\mathbb{E}_{x \sim \mu} \left( \int_0^S \int_0^1 \left| \int_{\mathbb{R}^k} \partial_z f\left(Z_s(\tau;x), \theta\right) \, \mathrm{d}(\rho(\theta, \tau, s) - \widetilde{\rho}^{\text{dis}}(\theta, \tau, s)) \right|^2 \mathrm{d}\tau \, \mathrm{d}s \right)}_{\text{(II)}} \right)^{1/2}
$$

$$
+ C(S, \mathcal{R}_{\text{ini}}, \mathcal{L}_0^{\text{sup}}) \left( \underbrace{\int_0^1 \left| \int_{\mathbb{R}^k} \partial_z f\left(Z_s(\tau;x), \theta\right) \, \mathrm{d}(\rho(\theta, \tau, s) - \widetilde{\rho}^{\text{dis}}(\theta, \tau, s)) \right|^2 \mathrm{d}\tau}_{\text{(III)}} \right)^{1/2} ,
$$

$$
\tag{109}
$$

All three terms in (109) can be controlled in expectation. Here we take the expectation with respect to the randomness initial drawing of $\{\theta_m(0;t)\}_{m=1}^M$. For (I), we have

$$
\mathbb{E}(\mathrm{I}) = \mathbb{E} \left( \mathbb{E}_{x \sim \mu} \left( \int_0^S \int_0^1 \left| \int_{\mathbb{R}^k} f\left(Z_s(\tau;x), \theta\right) \, \mathrm{d}(\rho(\theta, \tau, s) - \widetilde{\rho}^{\text{dis}}(\theta, \tau, s)) \right|^2 \mathrm{d}\tau \, \mathrm{d}s \right) \right)
$$

$$
= \mathbb{E}_{x \sim \mu} \left( \int_0^S \int_0^1 \mathbb{E} \left( \left| \int_{\mathbb{R}^k} f\left(Z_s(\tau;x), \theta\right) \, \mathrm{d}(\rho(\theta, \tau, s) - \widetilde{\rho}^{\text{dis}}(\theta, \tau, s)) \right|^2 \right) \mathrm{d}\tau \, \mathrm{d}s \right)
$$

$$
\leq \frac{C(S, \mathcal{R}_{\text{ini}}, \mathcal{L}_{\text{ini}}^{\text{sup}})}{M} \mathbb{E}_{x \sim \mu} \left( \int_0^S \int_0^1 \int_{\mathbb{R}^k} |f\left(Z_s(\tau;x), \theta\right)|^2 \, \mathrm{d}\rho(\theta, \tau, s) \, \mathrm{d}\tau \, \mathrm{d}s \right)
$$

$$
\leq \frac{C(S, \mathcal{R}_{\text{ini}}, \mathcal{L}_{\text{ini}}^{\text{sup}})}{M} ,
$$

where we use $\widetilde{\theta}_m(s;t) \sim \rho(\theta, t, s)$ in the first inequality. In second inequality, if $k_1 < k$, we use first inequality of Assumption 4.1 (15) with $\left| \theta_{[1]} \right| \leq C(S, \mathcal{R}_{\text{ini}}, \mathcal{L}_{\text{ini}}^{\text{sup}})$ and $|Z_s| \leq C(S, \mathcal{R}_{\text{ini}}, \mathcal{L}_{\text{ini}}^{\text{sup}})$. If $k_1 = k$, we use (12) and $|\theta| = \left| \theta_{[1]} \right| \leq C(S, \mathcal{R}_{\text{ini}}, \mathcal{L}_{\text{ini}}^{\text{sup}})$ and $|Z_s| \leq C(S, \mathcal{R}_{\text{ini}}, \mathcal{L}_{\text{ini}}^{\text{sup}})$ in the second inequality. By similar reasoning, we obtain

$$
\mathbb{E}(\mathrm{II}) \leq \frac{C(S, \mathcal{R}_{\text{ini}}, \mathcal{L}_{\text{ini}}^{\text{sup}})}{M}, \quad \mathbb{E}(\mathrm{III}) \leq \frac{C(S, \mathcal{R}_{\text{ini}}, \mathcal{L}_{\text{ini}}^{\text{sup}})}{M} .
$$

From Markov's inequality, these bounds imply that when $M > \frac{C(\mathcal{R}_{\text{ini}}, S, \mathcal{L}_{\text{ini}}^{\text{sup}})}{\epsilon^2 \eta}$, we have

$$
\mathbb{P}\left( \left\{ (\mathrm{I}) < \epsilon^2 \right\} \cap \left\{ (\mathrm{II}) < \epsilon^2 \right\} \cap \left\{ (\mathrm{III}) < \epsilon^2 \right\} \right) > 1 - \eta/2 .
\tag{110}
$$

Finally, using Definition E.1 (78), when $M > \frac{2C_3}{\eta}$,

$$
\mathbb{P}\left( \mathcal{L}_0^{\text{sup}} \leq C_4 \right) \geq 1 - \eta/2 .
\tag{111}
$$

By substituting (110) and (111) into (109), we see that there exists a constant $C(\mathcal{R}_{\text{ini}}, C_3, C_4, S)$ such that for any $\epsilon, \eta > 0$, when $M > \frac{C(\mathcal{R}_{\text{ini}}, C_3, C_4, S)}{\epsilon^2 \eta}$ we obtain that

$$
\mathbb{P}\left( \left| Z_s(1;x) - Z_s^{\text{dis}}(1;x) \right| < \epsilon \right) > 1 - \eta .
$$

By using this result and (111) in conjunction with (104), we complete the proof.

$\blacksquare$

## G CONVERGENCE TO THE CONTINUOUS LIMIT

This section is dedicated to the continuous limit and, in particular, the proof of Theorem E.2.

### G.1 STABILITY WITH DISCRETIZATION

Before proving Theorem E.2, and similarly to Appendix F.1, we first consider the stability of $Z$ and $p$ under discretization. Defining the path of parameters $\Theta(t) = \{\theta_m(t)\}_{m=1}^{M}$ and the set of parameters $\Theta_{L,M} = \{\theta_{l,m}\}_{l=0,m=1}^{L-1,M}$, we have the following lemma.

**Lemma G.1** *Suppose that Assumption 4.1 holds and that $x$ is in the support of $\mu$. Denoting*

$$\mathcal{L}^{\mathrm{sup}} = \sup_{0 \le t \le 1, l} \left\{ \frac{1}{M} \sum_{i=1}^{M} |\theta_{l,m}|^2, \frac{1}{M} \sum_{i=1}^{M} |\theta_m(t)|^2 \right\} \tag{112}$$

$$\mathcal{R} = \inf_{r>0} \left\{ \{\theta_{l,m}\}_{m=1,l=1}^{M,L} \cup \{\theta_m(t)\}_{m=1}^{M} \subset \{\theta \, | \, |\theta_{[1]}| < r\}, \, \forall t \in [0,1] \right\},$$

*there exists a constant $C(\mathcal{R}, \mathcal{L}^{\mathrm{sup}})$ depending only on $\mathcal{R}, \mathcal{L}^{\mathrm{sup}}$ such that for any $0 \le l \le L - 1$, we have*

$$\sup_{\frac{l}{L} \le t \le \frac{l+1}{L}} \left\{ \left| Z_\Theta(t;x) - Z_{\Theta_{L,M}}(l;x) \right|, \left| Z_\Theta(t;x) - Z_{\Theta_{L,M}}(l+1;x) \right| \right\}$$

$$\le C(\mathcal{R}, \mathcal{L}^{\mathrm{sup}}) \left( \frac{1}{M} \sum_{l=0}^{L-1} \sum_{m=1}^{M} \int_{\frac{l}{L}}^{\frac{l+1}{L}} |\theta_{l,m} - \theta_m(\tau)|^2 \, \mathrm{d}\tau \right)^{1/2} + \frac{C(\mathcal{R}, \mathcal{L}^{\mathrm{sup}})}{L}, \tag{113}$$

*and*

$$\sup_{\frac{l}{L} \le t \le \frac{l+1}{L}} \left| p_\Theta(t;x) - p_{\Theta_{L,M}}(l;x) \right|$$

$$\le C(\mathcal{R}, \mathcal{L}^{\mathrm{sup}}) \left( \frac{1}{M} \sum_{l=0}^{L-1} \sum_{m=1}^{M} \int_{\frac{l}{L}}^{\frac{l+1}{L}} |\theta_{l,m} - \theta_m(\tau)|^2 \, \mathrm{d}\tau \right)^{1/2} + \frac{C(\mathcal{R}, \mathcal{L}^{\mathrm{sup}})}{L}. \tag{114}$$

**Proof** Define

$$Z(t;x) = Z_\Theta(t;x), \quad p(t;x) = p_\Theta(t;x),$$

and

$$\widetilde{Z}(t;x) = \sum_{l=0}^{L-1} Z_{\Theta_{L,M}}(l;x) \mathbf{1}_{\frac{l}{L} \le t < \frac{l+1}{L}}, \quad \widetilde{p}(t;x) = \sum_{l=0}^{L-1} p_{\Theta_{L,M}}(l;x) \mathbf{1}_{\frac{l}{L} < t \le \frac{l+1}{L}}, \tag{115}$$

with

$$\widetilde{Z}(1;x) = Z_{\Theta_{L,M}}(L;x), \quad \widetilde{p}(0;x) = p_{\Theta_{L,M}}(0;x). \tag{116}$$

Using (1), (72), Assumption 4.1, and Lemma D.2 (73) and (75), we obtain for all $l = 0, 1, \dots, L - 1$ that

$$\left| Z_{\Theta_{L,M}}(l+1;x) - Z_{\Theta_{L,M}}(l;x) \right| < \frac{C(\mathcal{L}^{\mathrm{sup}})}{L},$$

$$\left| p_{\Theta_{L,M}}(l+1;x) - p_{\Theta_{L,M}}(l;x) \right| < \frac{C(\mathcal{L}^{\mathrm{sup}})}{L}. \tag{117}$$

Now define $\Delta_t$ by

$$\Delta_t = Z(t;x) - \widetilde{Z}(t;x).$$

For $t \in [\frac{l}{L}, \frac{l+1}{L}]$, we have from (4) that

$$|\Delta_t| \le \left| \Delta_{\frac{l}{L}} \right| + \frac{1}{M} \sum_{m=1}^{M} \int_{\frac{l}{L}}^{t} |f(Z(\tau;x), \theta_m(\tau))| \, \mathrm{d}\tau$$

$$\le \left| \Delta_{\frac{l}{L}} \right| + \frac{C(\mathcal{L}^{\mathrm{sup}})}{M} \sum_{m=1}^{M} \int_{\frac{l}{L}}^{\frac{l+1}{L}} (|\theta_m(\tau)|^2 + 1) \, \mathrm{d}\tau \tag{118}$$

$$\le \left| \Delta_{\frac{l}{L}} \right| + \frac{C(\mathcal{L}^{\mathrm{sup}})}{L},$$

where we use (12), (18), and (112) in the last two inequalities. From (1) and (4), we obtain further that

$$
\left| \Delta_{\frac{l+1}{L}} \right| = \left| \Delta_{\frac{l}{L}} \right| + \left| \frac{1}{M} \sum_{m=1}^{M} \int_{\frac{l}{L}}^{\frac{l+1}{L}} f(Z(\tau; x), \theta_m(\tau)) - f\left( \widetilde{Z}(\tau; x), \theta_{l,m} \right) \, \mathrm{d}\tau \right|
$$

$$
\leq \left| \Delta_{\frac{l}{L}} \right| + \left| \frac{1}{M} \sum_{m=1}^{M} \int_{\frac{l}{L}}^{\frac{l+1}{L}} f(Z(\tau; x), \theta_m(\tau)) - f\left( Z(\tau; x), \theta_{l,m} \right) \, \mathrm{d}\tau \right|
$$

$$
+ \left| \frac{1}{M} \sum_{m=1}^{M} \int_{\frac{l}{L}}^{\frac{l+1}{L}} f(Z(\tau; x), \theta_{l,m}) - f\left( \widetilde{Z}(\tau; x), \theta_{l,m} \right) \, \mathrm{d}\tau \right|
$$

$$
\overset{(I)}{\leq} \left| \Delta_{\frac{l}{L}} \right| + C(\mathcal{L}^{\mathrm{sup}}) \left( \frac{1}{M} \sum_{m=1}^{M} \int_{\frac{l}{L}}^{\frac{l+1}{L}} (|\theta_m(\tau)| + |\theta_{l,m}| + 1)|\theta_m(\tau) - \theta_{l,m}| \, \mathrm{d}\tau \right)
$$

$$
+ C(\mathcal{L}^{\mathrm{sup}}) |\Delta_\xi| \left( \frac{1}{ML} \sum_{m=1}^{M} (|\theta_{l,m}|^2 + 1) \right)
$$

$$
\overset{(II)}{\leq} \left( 1 + \frac{C(\mathcal{L}^{\mathrm{sup}})}{L} \right) \left| \Delta_{\frac{l}{L}} \right| + C(\mathcal{L}^{\mathrm{sup}}) \left( \frac{1}{M} \sum_{m=1}^{M} \int_{\frac{l}{L}}^{\frac{l+1}{L}} (|\theta_m(\tau)| + |\theta_{l,m}| + 1)|\theta_m(\tau) - \theta_{l,m}| \, \mathrm{d}\tau \right)
$$

$$
+ \frac{C(\mathcal{L}^{\mathrm{sup}})}{L^2},
$$

where $\xi \in [\frac{l}{L}, \frac{l+1}{L}]$, and we used the mean-value theorem with (13), (18), (73) in (I) and (112), (118) in (II). By applying this bound iteratively, we obtain

$$
\left| \Delta_{\frac{l}{L}} \right| \leq C(\mathcal{L}^{\mathrm{sup}}) |\Delta_0| + C(\mathcal{L}^{\mathrm{sup}}) \left( \frac{1}{M} \sum_{j=0}^{l-1} \sum_{m=1}^{M} \int_{\frac{j}{L}}^{\frac{j+1}{L}} (|\theta_m(\tau)| + |\theta_{l,m}| + 1)|\theta_m(\tau) - \theta_{l,m}| \, \mathrm{d}\tau \right) + \frac{C(\mathcal{L}^{\mathrm{sup}})}{L},
$$

where $|\Delta_0| = 0$. By combining this bound with (118) and using Hölder's inequality with (112), we obtain that

$$
|\Delta_t| \leq C(\mathcal{L}^{\mathrm{sup}}) \left( \frac{1}{M} \sum_{l=0}^{L-1} \sum_{m=1}^{M} \int_{\frac{l}{L}}^{\frac{l+1}{L}} |\theta_{l,m} - \theta_m(\tau)|^2 \, \mathrm{d}\tau \right)^{1/2} + \frac{C(\mathcal{L}^{\mathrm{sup}})}{L}. \tag{119}
$$

By combining (119) with (117), we prove (113).

To prove (114), we define
$$
\Delta_p(t; x) = p(t; x) - \widetilde{p}(t; x).
$$

Similarly to (34), we obtain

$$
|\Delta_p(1; x)| \leq C(\mathcal{L}^{\mathrm{sup}}) \left| \widetilde{Z}(1; x) - Z(1; x) \right|
$$

$$
\leq C(\mathcal{L}^{\mathrm{sup}}) \left( \frac{1}{M} \sum_{l=0}^{L-1} \sum_{m=1}^{M} \int_{\frac{l}{L}}^{\frac{l+1}{L}} |\theta_{l,m} - \theta_m(\tau)|^2 \, \mathrm{d}\tau \right)^{1/2} + \frac{C(\mathcal{L}^{\mathrm{sup}})}{L}. \tag{120}
$$

For $t \in \left( \frac{l}{L}, \frac{l+1}{L} \right]$ and using (11), we obtain that

$$
|\Delta_p(t; x)| \leq \left| \Delta_p\left( \frac{l+1}{L}; x \right) \right| + \frac{1}{M} \sum_{m=1}^{M} \int_{t}^{\frac{l+1}{L}} |\partial_z f(Z(\tau; x), \theta_m(\tau))| \, |p(\tau; x)| \, \mathrm{d}\tau
$$

$$
\leq \left| \Delta_p\left( \frac{l+1}{L}; x \right) \right| + \frac{C(\mathcal{L}^{\mathrm{sup}})}{M} \sum_{m=1}^{M} \int_{\frac{l}{L}}^{\frac{l+1}{L}} (|\theta_m(\tau)|^2 + 1) \, \mathrm{d}\tau \tag{121}
$$

$$
\leq \left| \Delta_p\left( \frac{l+1}{L}; x \right) \right| + \frac{C(\mathcal{L}^{\mathrm{sup}})}{L},
$$

where we use (13), (18), and (31) in the second inequality and the definition of $\mathcal{L}^{\mathrm{sup}}$ from (112) in the last line.

From (11), we obtain that

$$p^\top \left(\frac{l}{L}; x\right) = p^\top \left(\frac{l+1}{L}; x\right) + \frac{1}{M} \sum_{m=1}^{M} \int_{\frac{l}{L}}^{\frac{l+1}{L}} p^\top (\tau; x) \, \partial_z f(Z(\tau; x), \theta_m(\tau)) \, \mathrm{d}\tau \,,$$

while (72) implies that

$$\widetilde{p}^\top \left(\frac{l}{L}; x\right) = \widetilde{p}^\top \left(\frac{l+1}{L}; x\right) + \frac{1}{M} \sum_{m=1}^{M} \int_{\frac{l}{L}}^{\frac{l+1}{L}} \widetilde{p}^\top \left(\frac{l+1}{L}; x\right) \partial_z f(Z_{\Theta_{L,M}}(l; x), \theta_{l,m}) \, \mathrm{d}\tau \,,$$

where $\widetilde{p}$ is defined in (115), (116).

By bounding differences of these two expressions, we have

$$\left|\Delta_p \left(\frac{l}{L}; x\right)\right| \leq \left|\Delta_p \left(\frac{l+1}{L}; x\right)\right|$$

$$+ \underbrace{\frac{1}{M} \sum_{m=1}^{M} \int_{\frac{l}{L}}^{\frac{l+1}{L}} \left| p^\top (\tau; x) \, \partial_z f(Z(\tau; x), \theta_m(\tau)) \, \mathrm{d}\tau - \widetilde{p}^\top \left(\frac{l+1}{L}; x\right) \partial_z f(Z(\tau; x), \theta_m(\tau)) \right| \, \mathrm{d}\tau}_{\text{(I)}}$$

$$+ \underbrace{\frac{1}{M} \sum_{m=1}^{M} \int_{\frac{l}{L}}^{\frac{l+1}{L}} \left| \widetilde{p}^\top \left(\frac{l+1}{L}; x\right) \partial_z f(Z(\tau; x), \theta_m(\tau)) - \widetilde{p}^\top \left(\frac{l+1}{L}; x\right) \partial_z f(Z_{\Theta_{L,M}}(l; x), \theta_m(\tau)) \right| \, \mathrm{d}\tau}_{\text{(II)}}$$

$$+ \underbrace{\frac{1}{M} \sum_{m=1}^{M} \int_{\frac{l}{L}}^{\frac{l+1}{L}} \left| \widetilde{p}^\top \left(\frac{l+1}{L}; x\right) \partial_z f(Z_{\Theta_{L,M}}(l; x), \theta_m(\tau)) - \widetilde{p}^\top \left(\frac{l+1}{L}; x\right) \partial_z f(Z_{\Theta_{L,M}}(l; x), \theta_{l,m}) \right| \, \mathrm{d}\tau}_{\text{(III)}} \,.$$

$$(122)$$

We bound (I), (II), and (III) in turn.

(I): Using (13), (18), and (112), we obtain that

$$\text{(I)} \leq \frac{C(\mathcal{L}^{\mathrm{sup}})}{L} |\Delta_p(t; x)|$$

(II): Using Assumption 4.1 (14) together with (18), (73), (75), and (112), we obtain that

$$\text{(II)} \leq \frac{C(\mathcal{R}, \mathcal{L}^{\mathrm{sup}})}{M} \sum_{m=1}^{M} \int_{\frac{l}{L}}^{\frac{l+1}{L}} |Z(\tau; x) - Z_{\Theta_{L,M}}(l; x)| \, \mathrm{d}\tau$$

$$\leq \frac{C(\mathcal{R}, \mathcal{L}^{\mathrm{sup}})}{L} \left( \left( \frac{1}{M} \sum_{l'=0}^{L-1} \sum_{m=1}^{M} \int_{\frac{l'}{L}}^{\frac{l'+1}{L}} |\theta_{l',m} - \theta_m(\tau)|^2 \, \mathrm{d}\tau \right)^{1/2} + \frac{1}{L} \right) \,,$$

where we make use of (113) in the last inequality.

(III): Using Assumption 4.1 item 3 together with (73), (75), and (112), we obtain that

$$\text{(III)} \leq \frac{C(\mathcal{R}, \mathcal{L}^{\mathrm{sup}})}{M} \sum_{m=1}^{M} \int_{\frac{l}{L}}^{\frac{l+1}{L}} |\theta_m(\tau) - \theta_{l,m}| \, \mathrm{d}\tau$$

$$\leq C(\mathcal{R}, \mathcal{L}^{\mathrm{sup}}) \int_{\frac{l}{L}}^{\frac{l+1}{L}} \left( \frac{1}{M} \sum_{m=1}^{M} |\theta_m(\tau) - \theta_{l,m}|^2 \right)^{1/2} \mathrm{d}\tau$$

By substituting these three bounds and (121) into (122), we obtain

$$
\left| \Delta_p \left( \frac{l}{L}; x \right) \right| \leq \left( 1 + \frac{C(\mathcal{R}, \mathcal{L}^{\text{sup}})}{L} \right) \left| \Delta_p \left( \frac{l+1}{L}; x \right) \right|
$$

$$
+ \frac{C(\mathcal{R}, \mathcal{L}^{\text{sup}})}{L} \left( \left( \frac{1}{M} \sum_{l'=0}^{L-1} \sum_{m=1}^{M} \int_{\frac{l'}{L}}^{\frac{l'+1}{L}} |\theta_{l',m} - \theta_m(\tau)|^2 \, \mathrm{d}\tau \right)^{1/2} + \frac{1}{L} \right)
$$

$$
+ C(\mathcal{R}, \mathcal{L}^{\text{sup}}) \int_{\frac{l}{L}}^{\frac{l+1}{L}} \left( \frac{1}{M} \sum_{m=1}^{M} |\theta_m(\tau) - \theta_{l,m}|^2 \right)^{1/2} \, \mathrm{d}\tau + \frac{C(\mathcal{R}, \mathcal{L}^{\text{sup}})}{L^2} \, .
$$

By applying this bound iteratively, and using (120) and (121), we obtain

$$
|\Delta_p(t; x)| \leq C(\mathcal{R}, \mathcal{L}^{\text{sup}}) \left( \left( \frac{1}{M} \sum_{l=0}^{L-1} \sum_{m=1}^{M} \int_{\frac{l}{L}}^{\frac{l+1}{L}} |\theta_{l,m} - \theta_m(\tau)|^2 \, \mathrm{d}\tau \right)^{1/2} + \frac{1}{L} \right) , \tag{123}
$$

where we also use Hölder's inequality to write

$$
C(\mathcal{R}, \mathcal{L}^{\text{sup}}) \sum_{l=0}^{L-1} \int_{\frac{l}{L}}^{\frac{l+1}{L}} \left( \frac{1}{M} \sum_{m=1}^{M} |\theta_m(\tau) - \theta_{l,m}|^2 \right)^{1/2} \, \mathrm{d}\tau
$$

$$
\leq C(\mathcal{R}, \mathcal{L}^{\text{sup}}) \left( \frac{1}{M} \sum_{l=0}^{L-1} \sum_{m=1}^{M} \int_{\frac{l}{L}}^{\frac{l+1}{L}} |\theta_{l,m} - \theta_m(\tau)|^2 \, \mathrm{d}\tau \right)^{1/2} \, .
$$

We obtain (114) by combining (123) with (121). ∎

### G.2 PROOF OF THEOREM E.2

We denote by $\theta_m(s; t)$ the solution to (6) with initial $\{\theta_m(0; t)\}_{m=1}^{M}$ that are $i.i.d$ drawn from $\rho_{\text{ini}}(\theta, t)$. Further, $\theta_{l,m}(s)$ is a solution to (3) with initial value $\theta_{l,m}(0) = \theta_m \left( 0; \frac{l}{L} \right)$ for $0 \leq l \leq L - 1$ and $1 \leq i \leq M$. Define

$$
\mathcal{L}_{\text{ini}}^{\text{dis,sup}} = \sup_{0 \leq t \leq 1} \left\{ \frac{1}{M} \sum_{i=1}^{M} |\theta_m(0; t)|^2 \right\}
$$

and recall $\mathcal{R}_{\text{ini}}$ defined in (55). According to Definition E.1, $\mathcal{L}_{\text{ini}}^{\text{dis,sup}}$ is bounded with high probability. Then, we have the following lemma:

**Lemma G.2** *For fixed* $S > 0$ *and any* $s \in [0, S]$*, there exists a constant* $C(S, \mathcal{R}_{\text{ini}}, \mathcal{L}_{\text{ini}}^{\text{dis,sup}})$ *depending only on* $S, \mathcal{R}_{\text{ini}}, \mathcal{L}_{\text{ini}}^{\text{dis,sup}}$ *such that*

$$
\frac{1}{M} \sum_{m=1}^{M} |\theta_{l,m}(s)|^2 \leq C(S, \mathcal{R}_{\text{ini}}, \mathcal{L}_{\text{ini}}^{\text{dis,sup}}) ,
$$
$$
\{\theta_{l,m}(s)\}_{m=1, l=1}^{M, L} \subset \left\{ \theta \, \middle| \, |\theta_{[1]}| \leq C(S, \mathcal{R}_{\text{ini}}, \mathcal{L}_{\text{ini}}^{\text{dis,sup}}) \right\} , \tag{124}
$$

*Furthermore, the ODE solution and* $p_{\Theta_{L,M}(s)}$ *are bounded as follows, for any* $x$ *in the support of* $\mu$ *and* $l = 0, 1, \ldots, L - 1$:

$$
\left| Z_{\Theta_{L,M}(s)}(l+1; x) \right| \leq C \left( S, \mathcal{L}_{\text{ini}}^{\text{dis,sup}}, \mathcal{R}_{\text{ini}} \right) , \tag{125}
$$

*and*

$$
\left| p_{\Theta_{L,M}(s)}(l; x) \right| \leq C \left( S, \mathcal{L}_{\text{ini}}^{\text{dis,sup}}, \mathcal{R}_{\text{ini}} \right) . \tag{126}
$$

The proof is quite similar to that of Lemma F.3, so we omit the details.

We are now ready to prove Theorem E.2.

**Proof** [Proof of Theorem E.2] From (99) and (124) we obtain for all $t \in [0,1]$, $s \in [0,S]$,

$$\max\left\{\frac{1}{M}\sum_{m=1}^{M}|\theta_{l,m}(s)|^2, \frac{1}{M}\sum_{m=1}^{M}|\theta_m(s;t)|^2\right\} \leq C(S,\mathcal{R}_{\text{ini}},\mathcal{L}_{\text{ini}}^{\text{dis,sup}}),$$

$$\{\theta_m(s;t)\}_{m=1}^{M} \cup \{\theta_{l,m}(s)\}_{m=1,l=1}^{M,L} \subset \left\{\theta \Big| |\theta_{[1]}| \leq C(S,\mathcal{R}_{\text{ini}},\mathcal{L}_{\text{ini}}^{\text{dis,sup}})\right\}, \tag{127}$$

where $C(S,\mathcal{R}_{\text{ini}},\mathcal{L}_{\text{ini}}^{\text{dis,sup}})$ is a constant depending on $S, \mathcal{R}_{\text{ini}}$, and $\mathcal{L}_{\text{ini}}^{\text{dis,sup}}$.

From a similar derivation to (81), we obtain

$$|E(\Theta(s;\cdot)) - E(\Theta_{L,M}(s))| \leq C(S,\mathcal{R}_{\text{ini}},\mathcal{L}_{\text{ini}}^{\text{dis,sup}})\left|Z_{\Theta(s)}(1;x) - Z_{\Theta_{L,M}(s)}(L;x)\right|. \tag{128}$$

Thus, to prove the theorem, it suffices to prove that $\left|Z_{\Theta_{L,M}(s)}(L;x) - Z_{\Theta(s)}(1;x)\right|$ is small. For this purpose, according to (113), we need to bound the quantity

$$\frac{1}{M}\sum_{l=0}^{L-1}\sum_{m=1}^{M}\int_{\frac{l}{L}}^{\frac{l+1}{L}}|\Delta_{t,m}(s)|^2\,\mathrm{d}t, \tag{129}$$

where $\Delta_{t,m}(s) = \theta_{l,m}(s) - \theta_m(s;t)$. The next part of the proof contains the required bound.

First, using (3) and (6), we obtain that

$$\frac{\mathrm{d}|\Delta_{t,m}(s)|^2}{\mathrm{d}s}$$

$$= -2\left\langle\Delta_{t,m}(s), \mathbb{E}_{x\sim\mu}\left(\partial_\theta f(Z_{\Theta_{L,M}(s)}(l;x),\theta_{l,m}(s))p_{\Theta_{L,M}(s)}(l;x) - \partial_\theta f(Z_{\Theta(s)}(t;x),\theta_m(s;t))p_{\Theta(s)}(t;x)\right)\right\rangle$$

$$= -2\left\langle\Delta_{t,m}(s), \underbrace{\mathbb{E}_{x\sim\mu}\left(\partial_\theta f(Z_{\Theta_{L,M}(s)}(l;x),\theta_{l,m}(s))p_{\Theta_{L,M}(s)}(l;x) - \partial_\theta f(Z_{\Theta(s)}(t;x),\theta_m(s;t))p_{\Theta_{L,M}(s)}(l;x)\right)}_{\text{(I)}}\right\rangle$$

$$- 2\left\langle\Delta_{t,m}(s), \underbrace{\mathbb{E}_{x\sim\mu}\left(\partial_\theta f(Z_{\Theta(s)}(t;x),\theta_m(s;t))p_{\Theta_{L,M}(s)}(l;x) - \partial_\theta f(Z_{\Theta(s)}(t;x),\theta_m(s;t))p_{\Theta(s)}(t;x)\right)}_{\text{(II)}}\right\rangle. \tag{130}$$

To bound (I), we use (126) to obtain

$$|(\text{I})| \leq C(S,\mathcal{R}_{\text{ini}},\mathcal{L}_{\text{ini}}^{\text{dis,sup}})\mathbb{E}_{x\sim\mu}\left(\left|\partial_\theta f(Z_{\Theta_{L,M}(s)}(l;x),\theta_{l,m}(s)) - \partial_\theta f(Z_{\Theta(s)}(t;x),\theta_m(s;t))\right|\right)$$

$$\leq C(S,\mathcal{R}_{\text{ini}},\mathcal{L}_{\text{ini}}^{\text{dis,sup}})\left[\mathbb{E}_{x\sim\mu}\left(\left|Z_{\Theta_{L,M}(s)}(l;x) - Z_{\Theta(s)}(t;x)\right| + |\theta_{l,m}(s) - \theta_m(s;t)|\right)\right], \tag{131}$$

where we use Assumption 4.1 (14), (100), (125), and (127) in the second inequality. To bound (II), we use (13), (100), and (127) to obtain

$$|(\text{II})| \leq C(S,\mathcal{R}_{\text{ini}},\mathcal{L}_{\text{ini}}^{\text{dis,sup}})(|\theta_m(s;t)| + 1)\mathbb{E}_{x\sim\mu}\left(\left|p_{\Theta_{L,M}(s)}(l;x) - p_{\Theta(s)}(t;x)\right|\right). \tag{132}$$

By substituting (132) and (131) into (130), we obtain

$$\frac{\mathrm{d}|\Delta_{t,m}(s)|^2}{\mathrm{d}s}$$

$$\leq C(S,\mathcal{R}_{\text{ini}},\mathcal{L}_{\text{ini}}^{\text{dis,sup}})|\Delta_{t,m}(s)|^2$$

$$+ C(S,\mathcal{R}_{\text{ini}},\mathcal{L}_{\text{ini}}^{\text{dis,sup}})|\Delta_{t,m}(s)|\mathbb{E}_{x\sim\mu}\left(\left|Z_{\Theta_{L,M}(s)}(l;x) - Z_{\Theta(s)}(t;x)\right|\right)$$

$$+ C(S,\mathcal{R}_{\text{ini}},\mathcal{L}_{\text{ini}}^{\text{dis,sup}})|\Delta_{t,m}(s)|(|\theta_m(s;t)| + 1)\mathbb{E}_{x\sim\mu}\left(\left|p_{\Theta_{L,M}(s)}(l;x) - p_{\Theta(s)}(t;x)\right|\right).$$

Using Hölder's inequality similar to (107), we obtain

$$
\begin{aligned}
\frac{\mathrm{d}\frac{1}{M}\sum_{m=1}^{M}|\Delta_{t,m}(s)|^2}{\mathrm{d}s}
&\leq C(S,\mathcal{R}_{\mathrm{ini}},\mathcal{L}_{\mathrm{ini}}^{\mathrm{dis,sup}})\left(\frac{1}{M}\sum_{m=1}^{M}|\Delta_{t,m}(s)|^2\right)\\
&\quad + C(S,\mathcal{R}_{\mathrm{ini}},\mathcal{L}_{\mathrm{ini}}^{\mathrm{dis,sup}})\mathbb{E}_{x\sim\mu}\left(\left|p_{\Theta_{L,M}(s)}(t;x)-p_{\Theta(s)}(t;x)\right|^2\right)\\
&\quad + C(S,\mathcal{R}_{\mathrm{ini}},\mathcal{L}_{\mathrm{ini}}^{\mathrm{dis,sup}})\mathbb{E}_{x\sim\mu}\left(\left|Z_{\Theta_{L,M}(s)}(t;x)-Z_{\Theta(s)}(t;x)\right|^2\right),
\end{aligned}
\tag{133}
$$

By substituting (113) and (114) into (133), we obtain

$$
\frac{\mathrm{d}\frac{1}{M}\sum_{l=0}^{L-1}\sum_{m=1}^{M}\int_{\frac{l}{L}}^{\frac{l+1}{L}}|\Delta_{t,m}(s)|^2\,\mathrm{d}t}{\mathrm{d}s}\leq C(S,\mathcal{R}_{\mathrm{ini}},\mathcal{L}_{\mathrm{ini}}^{\mathrm{dis,sup}})\left(\frac{1}{M}\sum_{l=0}^{L-1}\sum_{m=1}^{M}\int_{\frac{l}{L}}^{\frac{l+1}{L}}|\Delta_{t,m}(s)|^2\,\mathrm{d}t+\frac{1}{L^2}\right),
$$

which implies, from Grönwall's inequality, that

$$
\frac{1}{M}\sum_{l=0}^{L-1}\sum_{m=1}^{M}\int_{\frac{l}{L}}^{\frac{l+1}{L}}|\Delta_{t,m}(s)|^2\,\mathrm{d}t\leq C(S,\mathcal{R}_{\mathrm{ini}},\mathcal{L}_{\mathrm{ini}}^{\mathrm{dis,sup}})\left(\frac{1}{M}\sum_{l=0}^{L-1}\sum_{m=1}^{M}\int_{\frac{l}{L}}^{\frac{l+1}{L}}|\Delta_{t,m}(0)|^2\,\mathrm{d}t+\frac{1}{L^2}\right).
\tag{134}
$$

We have thus established the bound (129). We also have

$$
\frac{1}{M}\sum_{l=0}^{L-1}\sum_{m=1}^{M}\int_{\frac{l}{L}}^{\frac{l+1}{L}}|\Delta_{t,m}(0)|^2\,\mathrm{d}t=\frac{1}{M}\sum_{l=0}^{L-1}\sum_{m=1}^{M}\int_{\frac{l}{L}}^{\frac{l+1}{L}}\left|\theta_m\left(0;\frac{l}{L}\right)-\theta_m(0;t)\right|^2\,\mathrm{d}t.
\tag{135}
$$

To complete the proof, we use (79) and take $M\geq\frac{8C_3}{\eta}$ to obtain

$$
\mathbb{P}\left(\frac{1}{M}\sum_{l=0}^{L-1}\sum_{m=1}^{M}\int_{\frac{l}{L}}^{\frac{l+1}{L}}|\Delta_{t,m}(0)|^2\,\mathrm{d}t\leq\frac{C_4}{L^2}\right)\geq 1-\frac{\eta}{8}.
$$

According to (78), when $M>\frac{8C_3}{\eta}$,

$$
\mathbb{P}\left(\mathcal{L}_{\mathrm{ini}}^{\mathrm{dis,sup}}\leq C_4\right)\geq 1-\frac{\eta}{8}.
\tag{136}
$$

By using these expressions to substitute $\frac{1}{M}\sum_{l=0}^{L-1}\sum_{m=1}^{M}\int_{\frac{l}{L}}^{\frac{l+1}{L}}|\Delta_{t,m}(0)|^2$ and $\mathcal{L}_{\mathrm{ini}}^{\mathrm{dis,sup}}$ into (134), we find that there exists a constant $C'(C_4,\mathcal{R}_{\mathrm{ini}},S)$ depending on $C_4$, $\mathcal{R}_{\mathrm{ini}}$, and $S$ such that if

$$
M\geq\frac{8C_3}{\eta},\quad L\geq\frac{C'(C_4,\mathcal{R}_{\mathrm{ini}},S)}{\epsilon},
$$

then we have

$$
\mathbb{P}\left(\frac{1}{M}\sum_{l=0}^{L-1}\sum_{m=1}^{M}\int_{\frac{l}{L}}^{\frac{l+1}{L}}|\Delta_{t,m}(s)|^2\,\mathrm{d}t\leq\epsilon^2\right)\geq 1-\frac{\eta}{4}.
\tag{137}
$$

Using (137), (136), and (127) to bound the right hand side of (113), we find that there exists another constant $C''(C_4,\mathcal{R}_{\mathrm{ini}},S)$ depending on $C_4$, $\mathcal{R}_{\mathrm{ini}}$, and $S$ such that if

$$
M\geq\frac{8C_3}{\eta},\quad L\geq\frac{C''(C_4,\mathcal{R}_{\mathrm{ini}},S)}{\epsilon},
$$

then we have

$$
\mathbb{P}\left(\left|Z_{\Theta(s)}(1;x)-Z_{\Theta_{L,M}(s)}(L;x)\right|\leq\epsilon\right)\geq 1-\frac{\eta}{2},
$$

By using this result and (136) in conjunction with (128), we complete the proof. ∎

# H    PROOF OF GLOBAL CONVERGENCE RESULT

Intuitively, if the equation (9) converges to a stationary point, denote by $\rho_\infty$, so that $\partial_s \rho_\infty = 0$, then

$$\nabla_\theta \left. \frac{\delta E}{\delta \rho} \right|_{\rho_\infty(\cdot,\cdot)} (\theta, t) = 0, \quad \rho_\infty(\theta, t)\text{-a.e. } \theta \in \mathbb{R}^k, \quad \text{a.e. } t \in [0, 1].$$

The rest of the analysis shows that $E(\rho_\infty) = 0$ when this happens. However, it is not direct because the condition above only suggests the fact that $\frac{\delta E}{\delta \rho}|_{\rho_\infty(;)}$ is a piecewise constant function. We need a stronger result that shows this constant has to be zero. This is achieved by Proposition 6.1. To show this proposition, we follow the proof in (Lu et al., 2020) that explores the expressive power of $f(x, \theta)$, particularly the universal kernel property of Assumption 4.1. It is this proposition that identifies stationary points with the global minimizer.

We should mention that the zero loss was demonstrated by Chizat & Bach (2018) for the 2-layer problem where the stability equates to zero-loss due to convexity. The extension to the multi-layer case is more difficult since convexity is not present.

## H.1    PROOF OF PROPOSITION 6.1

We first prove a lower bound for $p_\rho$ in the following lemma.

**Lemma H.1** *Suppose that $\rho \in C([0, 1]; \mathcal{P}^2)$ and that $p_\rho$ is a solution to (11). Denoting*

$$\mathcal{L}_\rho = \int_0^1 \int_{\mathbb{R}^k} |\theta|^2 d\rho(\theta, t),$$

*then for any $t \in [0, 1]$ we have that*

$$\mathbb{E}_{x \sim \mu} \left( |p_\rho(t; x)|^2 \right) \geq Q(\mathcal{L}_\rho) E(\rho), \tag{138}$$

*where $Q : \mathbb{R}_+ \to \mathbb{R}_+$ is a decreasing function.*

**Proof**   Recall that the initial condition for $p_\rho$ in (11) is:

$$p_\rho(1; x) = (g(Z_\rho(1; x)) - y(x)) \nabla g(Z_\rho(1; x)),$$

so from Assumption 4.1, we have

$$\mathbb{E}_{x \sim \mu} \left( |p_\rho(1; x)|^2 \right) \geq \left( \inf_{x \in \mathbb{R}^d} |\nabla g(x)| \right)^2 E(\rho).$$

Further, since the equation is linear, we have

$$\frac{\partial p_\rho^\top}{\partial t} = -p_\rho^\top \int_{\mathbb{R}^k} \partial_z f(Z_\rho, \theta) d\rho(\theta, t).$$

According to equation (13) in Assumption 4.1, we obtain

$$\left| \int_{\mathbb{R}^k} \partial_z f(Z_\rho(t; x), \theta) \, \mathrm{d}\rho_1(\theta, t) \right| \leq C(Z_\rho(t; x)) \int_{\mathbb{R}^k} (|\theta|^2 + 1) \, \mathrm{d}\rho(\theta, t)$$

$$\leq C(\mathcal{L}_\rho) \int_{\mathbb{R}^k} (|\theta|^2 + 1) \, \mathrm{d}\rho_1(\theta, t),$$

where we use (18) in the second inequality. By combining the last two bounds in the usual way, we obtain

$$\frac{\mathrm{d}|p_\rho(t; x)|^2}{\mathrm{d}t} \leq \left( 2C(\mathcal{L}_\rho) \int_{\mathbb{R}^k} (|\theta|^2 + 1) d\rho(\theta, t) \right) |p_\rho(t; x)|^2,$$

By solving the equation, we have

$$|p_\rho(t; x)|^2 \geq |p_\rho(1; x)|^2 \exp\left( -2C(\mathcal{L}_\rho) \int_t^1 \int_{\mathbb{R}^k} (|\theta|^2 + 1) d\rho(\theta, t) \right) \geq C(\mathcal{L}_\rho) |p_\rho(1; x)|^2.$$

The proof is finalized by taking expectation on both sides, and note that monotonicity comes from the format of the exponential term. ∎

We are now ready to prove Proposition 6.1.

**Proof** [Proof of Proposition 6.1] Denote

$$\mathcal{L}_\rho = \int_0^1 \int_{\mathbb{R}^k} |\theta|^2 \, \mathrm{d}\rho(\theta, t) \, \mathrm{d}t \,.$$

According to existence and uniqueness of the solution to (7), for any $t \in [0, 1]$, we can construct a map $\mathcal{Z}_t$ such that

$$\mathcal{Z}_t(x) = Z_\rho(t; x) \,.$$

Since the trajectory can be computed backwards in time, $\mathcal{Z}_t^{-1}$ is well defined. Further, we denote $\mu_t^* = (\mathcal{Z}_t)_\sharp \mu$ to be the pushforward of $\mu$ under map $\mathcal{Z}_t$ and let

$$p^*(t; x) = p_\rho\left(t; \mathcal{Z}_t^{-1}(x)\right) \,.$$

By Assumption 4.1 and classical ODE theory, $\mathcal{Z}_t$ and $\mathcal{Z}_t^{-1}$ are both continuous maps in $x$, and so are $p_\rho(t; x)$ and $p^*(t; x)$. With the change of variables, we have for all $t \in [0, 1]$ that

$$\frac{\delta E(\rho)}{\delta \rho}(\theta, t) = \int_{\mathbb{R}^d} p_\rho^\top(t; x) f(\mathcal{Z}_t(x), \theta) \, \mathrm{d}\mu = \int_{\mathbb{R}^d} (p^*(t; x))^\top f(x, \theta) \, \mathrm{d}\mu_t^* \,. \tag{139}$$

For a fixed $t_0 \in [0, 1]$, we have boundedness of the Jacobian from Lemma C.1, meaning that

$$\sup_{x \in \mathrm{supp}(\mu)} \left\| \frac{\mathrm{d}\mu_{t_0}^*(\mathcal{Z}_t^{-1}(x))}{\mathrm{d}\mu(x)} \right\|_2 \leq C(\mathcal{L}_\rho).$$

As a consequence, $\mu_{t_0}^*(x)$ has a compact support since $\mu(x)$ does. We denote the size of the support by $R^*$, defined to be a real number such that $\mathrm{supp}\left(\mu_{t_0}^*(x)\right) \subset \{x : |x| < R^*\}$.

We now derive a general formula for $\int \frac{\delta E(\rho)}{\delta \rho}(\theta, t) \, \mathrm{d}\nu$. Recalling (139), we have

$$\int_{\mathbb{R}^k} \frac{\delta E(\rho)}{\delta \rho}(\theta, t_0) \, \mathrm{d}\nu(\theta) = \int_{\mathbb{R}^d} (p^*(t_0; x))^\top \left( \int_{\mathbb{R}^k} f(x, \theta) \, \mathrm{d}\nu(\theta) \right) \mathrm{d}\mu_{t_0}^*(x)$$

$$= \int_{\mathbb{R}^d} (p^*(t_0; x))^\top \left( \int_{\mathbb{R}^k} f(x, \theta) \, \mathrm{d}\nu(\theta) + p^*(x, t_0) \right) \mathrm{d}\mu_{t_0}(x) \tag{140}$$

$$- \int_{\mathbb{R}^d} (p^*(t_0; x))^\top p^*(x, t_0) \, \mathrm{d}\mu_{t_0}^*(x) \,.$$

Noticing that according to Lemma H.1, if $E(\rho) \neq 0$, the second term above is strictly negative (less than $-Q(\mathcal{L}_\rho)E(\rho)$), the goal then is to find $\nu$ for which $\int \mathrm{d}\nu = 0$ that makes the first term small, so that the right-hand side in (140) is negative. Defining the continuous function $h$ to be

$$h(x) = p^*(t_0; x) + \int_{\mathbb{R}^k} f(x, \theta) \, \mathrm{d}\rho(\theta, t_0) \,,$$

then according to Assumption 4.1, for arbitrarily small $\epsilon$, there is a $\hat{\nu}$ so that $\int \hat{\nu} = 0$ and

$$\left\| h(x) - \int_{\mathbb{R}^k} f(x, \theta) \, \mathrm{d}\hat{\nu}(\theta) \right\|_{L^\infty_{|x| < R^*}} \leq \epsilon \,.$$

Setting $\nu = \rho - \hat{\nu}$ and substituting into the first term of (140), we obtain

$$\int_{\mathbb{R}^d} (p^*(t_0; x))^\top \left( \int_{\mathbb{R}^k} f(x, \theta) \, \mathrm{d}\nu(\theta) + p^*(x, t_0) \right) \mathrm{d}\mu_{t_0}(x)$$

$$\leq \int_{\mathbb{R}^d} |p^*(t_0; x)| \left| h(x) - \int_{\mathbb{R}^k} f(x, \theta) \, \mathrm{d}\hat{\nu}(\theta) \right| \mathrm{d}\mu_{t_0}^*(x) \tag{141}$$

$$\leq \|p^*(t_0; x)\|_{L^\infty_{|x| < R^*}} \left\| h(x) - \int_{\mathbb{R}^k} f(x, \theta) \, \mathrm{d}\hat{\nu}(\theta) \right\|_{L^\infty_{|x| < R^*}} \,.$$

By choosing $\epsilon$ small enough that (141) is less than $\frac{1}{2}Q(\mathcal{L}_\rho)E(\rho)$, we have from (140) $\int \frac{\delta E(\rho)}{\delta \rho}(\theta, t_0)\,\mathrm{d}\nu(\theta) < 0$, completing the proof. ∎

## H.2   2-HOMOGENEOUS CASE: PROOF OF THEOREM 6.1

We first give an example of 2-homogeneous activation function that satisfy Assumption 4.1, and 6.1.

**Remark H.1** *A function that satisfies Assumption 4.1 and the 2-homogeneous property of Assumption 6.1 is* $f(x,\theta) = f(x, \theta_{[1]}, \theta_{[2]}) = \sigma(\theta_{[1]}x + \theta_{[2]})\exp(-|x|^2)$, *where* $\theta_{[1]} \in \mathbb{R}^{d \times d}$, $\theta_{[2]} \in \mathbb{R}^d$, *and* $\sigma(x) = |\max\{x,0\}|^2$ *applied componentwise.*

Before proving the Theorem 6.1, we first introduce the following lemma, which shows that the separation property is preserved in the training process. Our proof of this result is adapted from (Chizat & Bach, 2018).

**Lemma H.2** *Let Assumptions 4.1 and 4.2 hold, and suppose that* $\rho_{\mathrm{ini}}(\theta, t)$ *is admissible with compact support. Let* $\rho(\theta, t, s) \in \mathcal{C}([0,\infty); \mathcal{C}([0,1]; \mathcal{P}^2))$ *solve (9). If there exists* $t_0 \in [0,1]$, *so that the initial condition* $\rho_{\mathrm{ini}}(\theta, t_0)$ *separates the spheres* $r_a \mathbb{S}^{k-1}$ *and* $r_b \mathbb{S}^{k-1}$ *for some* $0 < r_a < r_b$, *then for any* $s_0 \in [0,\infty)$, $\rho(\theta, t_0, s_0)$ *separates the spheres* $r'_a \mathbb{S}^{k-1}$ *and* $r'_b \mathbb{S}^{k-1}$ *for some* $0 < r'_a < r'_b$.

**Proof** For every fixed $0 < s_0 < S < \infty$, we note that the particle representation $\theta_\rho(s; t_0)$ of $\rho(\theta, t_0, s)$ updates the following equation:

$$\begin{cases} \dfrac{\mathrm{d}\theta_\rho(s; t_0)}{\mathrm{d}s} = -\nabla_\theta \dfrac{\delta E(\rho(s))}{\delta \rho}(\theta_\rho(s; t_0), t_0)\,, & s \in (0, S) \\ \theta_\rho(0; t_0) = \theta\,. \end{cases} \tag{142}$$

Define the map $\mathcal{P}_s(\theta)$ to be the solution map that solves the equation above for given initial condition $\theta$ up to time $s$. Our proof amounts to showing that this map preserves the separation property. According to (Chizat & Bach, 2018, Proposition C.11), we need only show that the inverse map of $\mathcal{P}_s(\theta)$ is stable near 0 for any fixed $0 < s < S$. That is, for any $\epsilon > 0$, we need to identify $\eta > 0$ such that

$$\mathcal{P}_s^{-1}(\theta) \subset \mathcal{B}_\epsilon(0)\,, \quad \forall \theta \in \mathcal{B}_\eta(0)\,, \tag{143}$$

where $\mathcal{B}_\eta(0)$ is the $k$-dimensional ball around original 0 with radius $\eta$.

Since $f$ is 2-homogeneous in $\theta$, we have that $|\partial_\theta f(z, 0)| = 0$ for all $z$. Thus, from (10),

$$\left| \nabla_\theta \frac{\delta E(\rho(s))}{\delta \rho}(0, t_0) \right| = 0.$$

Using estimate (49) from Lemma C.4, we obtain

$$\left| \nabla_\theta \frac{\delta E(\rho(s))}{\delta \rho}(\theta, t_0) \right| \le C(\mathcal{L}_S^{\mathrm{sup}})|\theta|\,,$$

where $\mathcal{L}_S^{\mathrm{sup}} = \sup_{0 \le s \le S, t \in [0,1]} \int_{\mathbb{R}^k} |\theta|^2 \,\mathrm{d}\rho(\theta, t, s)\,\mathrm{d}t$. This upper bound on the velocity implies in particular that

$$|\mathcal{P}_s^{-1}(\theta)| \le |\theta| \exp(C(\mathcal{L}_S^{\mathrm{sup}})s)\,,$$

which establishes (143) when we choose $\eta$ to satisfy $\eta < \epsilon \exp(-C(\mathcal{L}_S^{\mathrm{sup}})s)$, concluding the proof. ∎

We are now ready to prove Theorem 6.1.

**Proof** [Proof of Theorem 6.1] Since $\rho(\theta, t, s)$ converges to $\rho_\infty(\theta, t)$ in $\mathcal{C}([0,1]; \mathcal{P}^2)$, we have for any $t_0$ that

$$\sup_{s \ge 0} \int_{\mathbb{R}^k} |\theta|^2 \,\mathrm{d}\rho(\theta, t_0, s) < \infty\,. \tag{144}$$

According to Proposition 6.1, it suffices to prove that

$$\frac{\delta E(\rho_\infty)}{\delta \rho}(\theta, t_0) = 0, \quad \forall \theta \in \mathbb{R}^k\,. \tag{145}$$

We use a contradiction argument: We will assume that (145) is not satisfied and show that $\int_{\mathbb{R}^k} |\theta|^2 \, \mathrm{d}\rho(\theta, t_0, s)$ blows up to infinity as $s \to \infty$, contradicting (144). In particular, we will use homogeneity to construct a set in which the second moment blows up.

Define the functions $h_\infty$ and $h_s$ as follows:

$$h_\infty(\theta) = \frac{\delta E(\rho_\infty)}{\delta \rho}(\theta, t_0), \quad h_s(\theta) = \frac{\delta E(\rho(s))}{\delta \rho}(\theta, t_0) \, .$$

Recall from (10) that

$$\frac{\delta E(\rho)}{\delta \rho}(\theta, t_0) = \mathbb{E}_\mu \left( p_\rho^\top(t_0, x) f(Z(t_0; x), \theta) \right) \, . \tag{146}$$

Since (145) is not satisfied, there exists a $\theta^*$ such that $\frac{\delta E(\rho_\infty)}{\delta \rho}(\theta^*, t_0) \neq 0$. From (146), by Hölder's inequality,

$$0 < \left| \frac{\delta E(\rho_\infty)}{\delta \rho}(\theta^*, t_0) \right| \leq \left( \mathbb{E}_{x \sim \mu} \left( |p_{\rho_\infty}(t_0; x)|^2 \right) \right)^{1/2} \left( \mathbb{E}_{x \sim \mu} \left( |f(Z(t_0; x), \theta^*)|^2 \right) \right)^{1/2} \, ,$$

which implies

$$\mathbb{E}_{x \sim \mu} \left( |p_{\rho_\infty}(t_0; x)|^2 \right) > 0 \, .$$

Then, Since $f$ is a universal kernel according to Assumption 4.1, we can find $\nu$ such that $\int f(Z(t_0; x), \theta) \, \mathrm{d}\nu$ approximates $-p_{\rho_\infty}(t_0, x)$. leading to

$$\int_{\mathbb{R}^k} h_\infty(\theta) \, \mathrm{d}\nu(\theta) = \int_{\mathbb{R}^k} \frac{\delta E(\rho_\infty)}{\delta \rho}(\theta, t_0) \, \mathrm{d}\nu(\theta) < -\frac{1}{2} \mathbb{E}_{x \sim \mu} \left( |p_{\rho_\infty}(t_0; x)|^2 \right) < 0 \, .$$

As a consequence, there exists at least one point $\theta_0 \in \mathbb{R}^k$ such that $h_\infty(\theta_0) < 0$. Since $f$ is 2-homogeneous, by (10), $h$ is also 2-homogeneous, so that

$$h_\infty(\theta_0/|\theta_0|) < 0 \, .$$

Because of continuity, there is a small neighborhood around $\theta_0/|\theta_0|$ in $\mathbb{S}^{k-1}$ where $h$ is strictly negative. Moreover, since $h$ is Sard-type regular, there exist $\epsilon > 0$ and $\eta > 0$ such that

$$\begin{cases} A = \left\{ \widetilde{\theta} \in \mathbb{S}^{k-1} \middle| h_\infty|_{\mathbb{S}^{k-1}} \left( \widetilde{\theta} \right) < -\epsilon \right\} \neq \emptyset \, , \\ \nabla_{\widetilde{\theta}} h_\infty|_{\mathbb{S}^{k-1}} \left( \widetilde{\theta} \right) \cdot n_{\widetilde{\theta}} > \eta \, , \quad \forall \widetilde{\theta} \in \partial A \, , \end{cases}$$

where $h_\infty|_{\mathbb{S}^{k-1}}$ is the confinement of $h_\infty$ on $\mathbb{S}^{k-1}$, and $n_{\widetilde{\theta}}$ is the outer normal vector to $\partial A$.

This statement of $h_\infty$ can be extended to $h_s$ for sufficiently larger $s$ as well. Using estimate (48) from Lemma C.4, we obtain that

$$h_s(\theta) \to h_\infty(\theta) \quad \text{in} \quad \mathcal{C}^1_{\text{loc}}(\mathbb{R}^k), \quad \text{as } s \to \infty \, ,$$

meaning there exists $S > 0$ such that for any $s \geq S$, we have

$$\begin{cases} h_s|_{\mathbb{S}^{k-1}} \left( \widetilde{\theta} \right) < -\epsilon/2 \, , & \forall \widetilde{\theta} \in A \, , \\ \nabla_{\widetilde{\theta}} h_s|_{\mathbb{S}^{k-1}} \left( \widetilde{\theta} \right) \cdot n_{\widetilde{\theta}} > \frac{1}{2}\eta \, , & \forall \widetilde{\theta} \in \partial A \, . \end{cases}$$

Extending this patch on the unit sphere to the whole domain, we define the cone:

$$\mathcal{A} = \left\{ \theta \in \mathbb{R}^k \middle| |\theta| > 0, \, \theta/|\theta| \in A \right\} \, .$$

Using the 2-homogeneous property of $h_s$, we have for $s \geq S$ that

$$\begin{cases} h_s(\theta) < -\frac{\epsilon|\theta|^2}{2} \, , & \forall \theta \in \mathcal{A} \, , \\ \nabla_\theta h_s(\theta) \cdot \vec{n}_\theta > 0 \, , & \forall \theta \in \partial \mathcal{A} \cap \{|\theta| > 0\} \, , \end{cases} \tag{147}$$

where $\vec{n}_\theta$ is the outer normal vector to $\partial \mathcal{A}$.

We now define a new system that follows the gradient flow corresponding to $\rho_s$. Denote by $\widehat{\theta}(s; \alpha)$ the solution to the following ODE:

$$
\begin{cases}
\dfrac{d\widehat{\theta}(s; \alpha)}{ds} = -\nabla_\theta \dfrac{\delta E(\rho(s))}{\delta\rho}\left(\widehat{\theta}(s; \alpha), t_0\right) = -\nabla_\theta h_s\left(\widehat{\theta}(s; \alpha)\right), & s > S \\
\widehat{\theta}(S; \alpha) = \alpha,
\end{cases}
\tag{148}
$$

where $\alpha \in \mathbb{R}^k$. According to (147), when $\theta \in \partial\mathcal{A} \cap \{|\theta| > 0\}$, $\nabla_\theta h_s(\theta)$ points outwards, away from $\mathcal{A}$. We also notice that $\widehat{\theta}\left(s; \vec{0}\right) = \vec{0}$. Thus if the ODE starts with from some $\alpha \in \mathcal{A}$, then for any $s \geq S$, the particle stays within $\mathcal{A}$, that is,

$$
\widehat{\theta}(s; \alpha) \in \mathcal{A}.
\tag{149}
$$

As a consequence, we have

$$
\frac{d\left|\widehat{\theta}(s; \alpha)\right|^2}{ds} = -2\left\langle \widehat{\theta}(s; \alpha), \nabla_\theta h_s\left(\widehat{\theta}(s; \alpha)\right)\right\rangle = -4h_s\left(\widehat{\theta}(s; \alpha)\right) > 2\epsilon\left|\widehat{\theta}(s; \alpha)\right|^2,
\tag{150}
$$

where we use the 2-homogeneous property of $h_s$ in the second equality and (147) in the final inequality.

According to Lemma H.2, there exist two spheres separated by $\rho(\theta, t_0, S)$, meaning that there exist $\beta > 0$ and $\gamma > 0$ relatively small (for example, with $\beta < r'_a$) such that

$$
\int_{\mathcal{A} \cap (\mathcal{B}_\beta(\vec{0}))^c} d\rho(\theta, t_0, S) > \gamma.
\tag{151}
$$

By tracing the trajectory of (148), we have

$$
\int_{\mathcal{A} \cap (\mathcal{B}_\beta(\vec{0}))^c} d\rho(\theta, t_0, s) = \int_{\mathbb{R}^k} \mathbf{1}_{\widehat{\theta}(s;\alpha) \in \mathcal{A} \cap (\mathcal{B}_\beta(\vec{0}))^c}\, d\rho(\alpha, t_0, S)
$$

$$
\geq \int_{\mathcal{A} \cap (\mathcal{B}_\beta(\vec{0}))^c} d\rho(\alpha, t_0, S) > \gamma, \quad s \geq S,
$$

where in the first inequality we also use $\frac{d|\widehat{\theta}(s;\alpha)|^2}{ds} \geq 0$ when $\alpha \in \mathcal{A}$. Further, we have

$$
\frac{d\int_{\mathcal{A} \cap (\mathcal{B}_\beta(\vec{0}))^c} \mathbf{1}_{\widehat{\theta}(s;\alpha) \in \mathcal{A} \cap (\mathcal{B}_\beta(\vec{0}))^c}\left|\widehat{\theta}(s; \alpha)\right|^2 d\rho(\alpha, t_0, S)}{ds}
$$

$$
= \frac{d\int_{\mathcal{A} \cap (\mathcal{B}_\beta(\vec{0}))^c}\left|\widehat{\theta}(s; \alpha)\right|^2 d\rho(\alpha, t_0, S)}{ds}
$$

$$
\geq 2\epsilon \int_{\mathcal{A} \cap (\mathcal{B}_\beta(\vec{0}))^c}\left|\widehat{\theta}(s; \alpha)\right|^2 d\rho(\alpha, t_0, S)
$$

$$
\geq 2\epsilon\gamma\beta^2
$$

where we use (150) in the second inequality and (151) in the final inequality. It follows that

$$
\lim_{s \to \infty} \int_{\mathcal{A} \cap (\mathcal{B}_\beta(\vec{0}))^c} \mathbf{1}_{\widehat{\theta}(s;\alpha) \in \mathcal{A} \cap (\mathcal{B}_\beta(\vec{0}))^c}\left|\widehat{\theta}(s; \alpha)\right|^2 d\rho(\alpha, t_0, S) = \infty.
$$

It follows from this result that

$$
\lim_{s \to \infty} \int_{\mathbb{R}^k} |\theta|^2 d\rho(\theta, t_0, s) = \infty,
$$

contradicting (144). Therefore, we must have

$$
\frac{\delta E(\rho_\infty)}{\delta\rho}(\theta, t_0) = 0, \quad \forall \theta \in \mathbb{R}^k,
$$

which completes the proof. ∎

## H.3 PARTIALLY 1-HOMOGENEOUS CASE: PROOF OF THEOREM 6.2

We first give an example of partially 1-homogeneous activation function that satisfy Assumption 4.1, and 6.2.

**Remark H.2** *The following function satisfies Assumptions 4.1 and 6.2: Let $\theta = (\theta_{[1]}, \theta_{[2]}, \theta_{[3]})$ with $\theta_{[1]} \in \mathbb{R}^d$, $\theta_{[2]} \in \mathbb{R}^{d \times d}$, $\theta_{[3]} \in \mathbb{R}^d$. Define $f(x, \theta) = f(x, \theta_{[1]}, \theta_{[2]}, \theta_{[3]}) = \theta_{[1]} \sigma \left( \frac{\theta_{[2]} \sigma_2(|\theta_{[2]}|)}{|\theta_{[2]}|} x + \frac{\theta_{[3]} \sigma_2(|\theta_{[3]}|)}{|\theta_{[3]}|} \right)$, where $\sigma(x)$ is a regularized ReLU activation function, and $\sigma_2, \sigma_2(x)/x : \mathbb{R}_+ \to \mathbb{R}_+$ are bounded, Lipschitz, and differentiable with Lipschitz differential. One way (of many) to define a regularized ReLU activation function is $\sigma(x) = (x + \eta)^2/(4\eta) \mathbf{1}_{x \in [-\eta, \eta]} + x \mathbf{1}_{x \in (\eta, \infty)}$, for some small $\eta$.*

As in the previous theorem, we prove a lemma, adapted from (Chizat & Bach, 2018, Lemma C.13), that asserts preservation of the separation property.

**Lemma H.3** *Let Assumptions 4.1 and 4.2 with $k_1 = 1$. Suppose that $\rho_{\mathrm{ini}}(\theta, t)$ is admissible and $\mathrm{supp}_\theta(\rho_{\mathrm{ini}}(\theta, t)) \subset \{\theta \| \theta_{[1]} | \leq R\}$ with some $R > 0$ for all $t \in [0, 1]$. Let $\rho(\theta, t, s) \in \mathcal{C}([0, \infty); \mathcal{C}([0, 1]; \mathcal{P}^2))$ solve (9). Suppose in addition that*

- *$f$ satisfies the partial 1-homogeneous condition (see Assumption 6.2),*

- *The initial conditions satisfy the separation condition, that is, there exists $t_0 \in [0, 1]$ such that $\rho_{\mathrm{ini}}(\theta_{[1]}, \theta_{[2]}, t_0)$ separates the spheres $\{-r_0\} \times \mathbb{R}^{k-1}$ and $\{r_0\} \times \mathbb{R}^{k-1}$ for some $r_0 > 0$.*

*Then for any $s_0 \in [0, \infty)$, $\rho(\theta_{[1]}, \theta_{[2]}, t_0, s_0)$ separates $\{-r'\} \times \mathbb{R}^{k-1}$ and $\{r'\} \times \mathbb{R}^{k-1}$ for some $r' > 0$.*

**Proof** Note that the particle representation $\theta_\rho(s; t_0)$ of $\rho(\theta, s, t_0)$ satisfies

$$\begin{cases} \frac{\mathrm{d}\theta_\rho(s; t_0)}{\mathrm{d}s} &= -\nabla_\theta \frac{\delta E(\rho(s))}{\delta \rho} (\theta_\rho(s; t_0), t_0), \\ \theta_\rho(0; t_0) &= \theta. \end{cases} \tag{152}$$

Define a continuous map $\mathcal{P} : \mathbb{R}^k \times [0, \infty) \to \mathbb{R}^k$ as the solution to (152), that is, $\mathcal{P}(\theta, s)$ is the solution to (152) with initial condition $\theta_\rho(0; t_0) = \theta$, where $t_0$ is fixed. Define a diffeomorphism $\psi : \mathbb{R} \times \mathbb{R}^{k-1} \to \mathbb{R} \times \mathcal{B}_1(0)$ as follows:

$$\psi(\theta_{[1]}, \theta_{[2]}) = \begin{cases} \left( \theta_{[1]}, \left( \theta_{[2]}/|\theta_{[2]}| \right) \cdot \tanh \left( |\theta_{[2]}| \right) \right), & \theta_{[2]} \neq 0 \\ (\theta_{[1]}, 0), & \theta_{[2]} = 0, \end{cases}$$

where $\theta_{[1]} \in \mathbb{R}$ is the first component of $\theta$ and $\theta_{[2]} \in \mathbb{R}^{k-1}$ contains the remaining components. This map keeps the first component of $\theta_{[1]}$ intact and shrinks $\theta_{[2]}$ to push its amplitude below 1. This diffeomorphism preserves the connection/separation property.

Define the continuous map $\mathcal{Q}$ as follows:

$$\mathcal{Q}(\theta, s) = \psi \circ \mathcal{P}(\psi^{-1}(\theta), s) : \mathbb{R} \times \mathcal{B}_1(0) \times [0, \infty) \to \mathbb{R} \times \mathcal{B}_1(0).$$

Since $\psi$ preserves the connection property, the lemma is proved if we can show $\psi \circ \mathcal{P}(\mathrm{supp}(\rho_{\mathrm{ini}}), s)$ separates $\{-r'\} \times \mathcal{B}_1(0)$ and $\{r'\} \times \mathcal{B}_1(0)$ for some $r' > 0$. Since $\psi \circ \mathcal{P}(\mathrm{supp}(\rho_{\mathrm{ini}}), s) = \mathcal{Q}(\psi(\mathrm{supp}(\rho_{\mathrm{ini}})), s)$, we trace the evolution of $\mathcal{Q}(\theta, s)$ for $\theta \in \mathbb{R} \times \mathcal{B}_1(0)$. According to (Chizat & Bach, 2018, Proposition C.14), this translates to showing $\mathcal{Q}(\theta, s)$ can be continuously extended to $\mathbb{R} \times \overline{\mathcal{B}_1(0)} \times [0, S] \to \mathbb{R} \times \overline{\mathcal{B}_1(0)}$, with the extension satisfying

$$\mathcal{Q}(\theta, s) \in \mathbb{R} \times \partial \mathcal{B}_1 \left( \vec{0} \right), \quad \forall \theta \in \mathbb{R} \times \partial \mathcal{B}_1 \left( \vec{0} \right), \quad s \in [0, \infty), \tag{153}$$

meaning that the extension $\mathcal{Q}(\cdot, s)$ maps $\mathbb{R} \times \partial \mathcal{B}_1(\vec{0})$ to itself for all $s \in [0, \infty)$.

Denoting $\mathcal{Q}_s(\theta) = \mathcal{Q}(\theta, s)$, we consider the velocity field of this flow (similar to the proof of (Chizat & Bach, 2018, Lemma C.13)):

$$
\begin{aligned}
\frac{\mathrm{d}\mathcal{Q}_s}{\mathrm{d}s} &= \left( \nabla_\theta \psi \left( \mathcal{P}_s \circ \psi^{-1}(\theta) \right) \right) \frac{\mathrm{d}\mathcal{P}_s \circ \psi^{-1}}{\mathrm{d}s} \\
&= - \left( \nabla_\theta \psi \left( \mathcal{P}_s \circ \psi^{-1}(\theta) \right) \right) \left( \nabla_\theta \frac{\delta E(\rho(s))}{\delta \rho} \left( \mathcal{P}_s \circ \psi^{-1}(\theta), t_0 \right) \right) \\
&= - \left( \nabla_\theta \psi \left( \psi^{-1}(\mathcal{Q}_s) \right) \right) \left( \nabla_\theta \frac{\delta E(\rho(s))}{\delta \rho} \left( \psi^{-1}(\mathcal{Q}_s), t_0 \right) \right) = V(\mathcal{Q}_s, s) \, .
\end{aligned}
$$

From the fourth condition of Theorem 6.2, the velocity field $V(\theta, s)$ can be continuously extended to $\mathbb{R} \times \partial \mathcal{B}_1(0)$ as follows:

$$
V(\theta, s) = V(\theta_{[1]}, \theta_{[2]}, s) = \begin{cases} - \left( \nabla_\theta \psi \left( \psi^{-1}(\theta) \right) \right) \left( \nabla_\theta \frac{\delta E(\rho(s))}{\delta \rho} \left( \psi^{-1}(\theta), t_0 \right) \right), & |\theta_{[2]}| < 1 \\ \left( -H_{\infty, \rho(s)}(\theta_{[2]}), 0 \right), & |\theta_{[2]}| = 1 \, , \end{cases}
$$

where $H_{\infty, \rho(s)}$ is the limit of $\frac{\delta E(\rho(s))}{\delta \rho} \left( 1, r\theta_{[2]}, t_0 \right)$ as $r \to \infty$. Within this velocity field, $\mathcal{Q}_s$ can be continuously extended to $\mathbb{R} \times \overline{\mathcal{B}_1(0)} \times [0, \infty) \to \mathbb{R} \times \overline{\mathcal{B}_1(0)}$ with the extension satisfying (153). This completes the proof. ∎

We are now ready to prove Theorem 6.2.

**Proof** [Proof of Theorem 6.2] The technique of proof is similar to the 2-homogeneous case. Since $\rho(\theta, t, s)$ converges to $\rho_\infty(\theta, t)$ in $\mathcal{C}([0, 1]; \mathcal{P}^2)$, we have

$$
\sup_{s \geq 0} \int_{\mathbb{R}^k} |\theta|^2 d\rho(\theta, t_0, s) < \infty \, . \tag{154}
$$

According to Proposition 6.1, it suffices to prove that

$$
\frac{\delta E(\rho_\infty)}{\delta \rho}(\theta, t_0) = \frac{\delta E(\rho_\infty)}{\delta \rho}(\theta_{[1]}, \theta_{[2]}, t_0) = \theta_{[1]} \frac{\delta E(\rho_\infty)}{\delta \rho}(1, \theta_{[2]}, t_0) = 0, \quad \forall \theta \in \mathbb{R}^k \, . \tag{155}
$$

In the following, we will show that $\int_{\mathbb{R}^k} |\theta|^2 d\rho(\theta, t_0, s)$ blows up as $s \to \infty$ if (155) fails to hold, in contradiction to (154).

Denote

$$
h_\infty(\theta_{[2]}) = \frac{\delta E(\rho_\infty)}{\delta \rho}(1, \theta_{[2]}, t_0), \quad h_s(\theta_{[2]}) = \frac{\delta E(\rho(s))}{\delta \rho}(1, \theta_{[2]}, t_0) \, . \tag{156}
$$

Then if (155) is not satisfied, without loss of generality, we can assume that there exists $\theta_{[2]}$ such that $h_\infty(\theta_{[2]}) < 0$. Since $h_\infty$ satisfies Sard-type regularity, there exist $\epsilon > 0$ and $\eta > 0$ such that

$$
A = \left\{ \theta_{[2]} \in \mathbb{R}^{k-1} \middle| h_\infty \left( \theta_{[2]} \right) < -\epsilon \right\} \neq \emptyset; \quad \nabla_{\theta_{[2]}} h_\infty(\theta_{[2]}) \cdot \vec{n}_{\theta_{[2]}} > \eta, \quad \forall \theta_{[2]} \in \partial A \, ,
$$

where $n_{\theta_{[2]}}$ is the outer normal vector on $\partial A$.

Using the definition of $\frac{\delta E}{\delta \rho}$ as in (10), we have

$$
\begin{aligned}
&\left| \nabla_{\theta_{[2]}} \frac{\delta E(\rho(s))}{\delta \rho}(1, \theta_{[2]}, t) - \nabla_{\theta_{[2]}} \frac{\delta E(\rho_\infty)}{\delta \rho}(1, \theta_{[2]}, t) \right| \\
&= \mathbb{E}_{x \sim \mu} \left( \partial_{\theta_{[2]}} \hat{f}(Z_{\rho(s)}(t; x), \theta_{[2]}) p_{\rho(s)}(t; x) - \partial_{\theta_{[2]}} \hat{f}(Z_{\rho_\infty}(t; x), \theta_{[2]}) p_{\rho_\infty}(t; x) \right) \\
&\leq \mathbb{E}_{x \sim \mu} \left( \left| \partial_{\theta_{[2]}} \hat{f}(Z_{\rho(s)}(t; x), \theta_{[2]}) - \partial_{\theta_{[2]}} \hat{f}(Z_{\rho_\infty}(t; x), \theta_{[2]}) \right| |p_{\rho(s)}(t; x)| \right) \\
&\quad + \mathbb{E}_{x \sim \mu} \left( \left| \partial_{\theta_{[2]}} \hat{f}(Z_{\rho_\infty}(t; x), \theta_{[2]}) \right| |p_{\rho(s)}(t; x) - p_{\rho_\infty}(t; x)| \right) \\
&\leq C \left( \mathbb{E}_{x \sim \mu} \left( |Z_{\rho(s)}(t; x) - Z_{\rho_\infty}(t; x)| \right) + \mathbb{E}_{x \sim \mu} \left( |p_{\rho(s)}(t; x) - p_{\rho_\infty}(t; x)| \right) \right) \\
&\leq C d_1(\rho(s), \rho_\infty) \, ,
\end{aligned}
$$

where we have used the Lipschitz continuity of $f$ and its derivatives as in Assumption 6.2 and the estimates (18), (31), and (47). We thus have

$$h_s(\theta_{[2]}) \to h_\infty(\theta_{[2]}) \quad \text{in} \quad \mathcal{C}^1(\mathbb{R}^{k-1}), \quad \text{as } s \to \infty. \tag{157}$$

As a consequence, there exists $S > 0$ such that for any $s \geq S$

$$\begin{cases} h_s\left(\theta_{[2]}\right) < -\epsilon/2, & \forall \theta_{[2]} \in A, \\ \nabla_{\theta_{[2]}} h_s(\theta_{[2]}) \cdot \vec{n}_{\theta_{[2]}} > \frac{1}{2}\eta, & \forall \theta_{[2]} \in \partial A. \end{cases} \tag{158}$$

Extending this set to the whole space, we define

$$\mathcal{A} = \left\{ (\theta_{[1]}, \theta_{[2]}) \in (0, \infty) \times A \right\}.$$

Since $\partial\mathcal{A} = \left\{ (\theta_{[1]}, \theta_{[2]}) \in (0, \infty) \times \partial A \right\} \cup \{\theta_{[1]} = 0, \theta_{[2]} \in \overline{A}\}$, from (158) and the definition of $h_s$, we have

$$\nabla_\theta \frac{\delta E(\rho(s))}{\delta\rho}(\theta, t_0) \cdot \vec{n}_\theta = \theta_{[1]} \left( \nabla_{\theta_{[2]}} h_s(\theta_{[2]}) \cdot \vec{n}_{\theta_{[2]}} \right) > 0, \quad \forall \theta \in (0, \infty) \times \partial A, \tag{159}$$

where $\vec{n}_\theta$ is the outer normal direction on $\partial\mathcal{A}$, and $\vec{n}_{\theta_{[2]}}$ is the outer normal vector on $\partial A$. When $\theta \in \{\theta_{[1]} = 0, \theta_{[2]} \in \overline{A}\}$,

$$\left[ \nabla_\theta \frac{\delta E(\rho(s))}{\delta\rho}(\theta, t_0) \right]_1 = h_s(\theta_{[2]}) < 0, \quad \left[ \nabla_\theta \frac{\delta E(\rho(s))}{\delta\rho}(\theta, t_0) \right]_i = 0, \quad i = 2, \ldots, k, \tag{160}$$

where $[\cdot]_i$ means the $i$-component of the vector. This implies that $\nabla_\theta \frac{\delta E(\rho(s))}{\delta\rho}(\theta, t_0)$ points strictly downward when $\theta \in \{\theta_{[1]} = 0, \theta_{[2]} \in \overline{A}\}$.

As in the 2-homogeneous case, we consider the gradient flow corresponding to $\rho_s$. Denote by $\widehat{\theta}(s; \alpha)$ the solution to the following ODE:

$$\begin{cases} \frac{d\widehat{\theta}(s;\alpha)}{ds} = -\nabla_\theta \frac{\delta E(\rho(s))}{\delta\rho}\left(\widehat{\theta}(s;\alpha), t_0\right) = -\nabla_\theta \left(\theta_{[1]} h_s\left(\theta_{[2]}\right)\right)\left(\widehat{\theta}(s;\alpha)\right), & s > S \\ \widehat{\theta}(S;\alpha) = \alpha, \end{cases} \tag{161}$$

where $\alpha \in \mathbb{R}^k$. Since the minus gradient is pointing inward to $\mathcal{A}$, as stated in (159) and (160), $\widehat{\theta}(s;\alpha)$ stays in $\mathcal{A}$ if $\widehat{\theta}(s';\alpha) \in \mathcal{A}$ for some $s' \in [S, s]$. Moreover, using (161), if $\widehat{\theta}(s;\alpha) \in \mathcal{A}$, we have

$$\frac{d|\widehat{\theta}_{[1]}(s;\alpha)|^2}{ds} = -2\widehat{\theta}_{[1]}(s;\alpha)\, h_s\left(\widehat{\theta}_{[2]}(s;\alpha)\right) > \epsilon\widehat{\theta}_{[1]}(s;\alpha), \tag{162}$$

where the last inequality uses (158).

Similar to (Chizat & Bach, 2018, Proposition C.4), we claim that there exists $S_1 \geq S$, $\beta > 0$, and $\gamma > 0$ such that (see detailed proof in Appendix H.3.1)

$$\int_{(\beta,\infty)\times A} d\rho(\theta, t_0, S_1) > \gamma. \tag{163}$$

Then

$$\frac{d \int_{\mathbb{R}^k} \mathbf{1}_{\widehat{\theta}(S_1;\alpha)\in(\beta,\infty)\times A} \left|\widehat{\theta}_{[1]}(s;\alpha)\right|^2 d\rho(\alpha, t_0, S)}{ds}$$
$$\geq \epsilon \int_{\mathbb{R}^k} \mathbf{1}_{\widehat{\theta}(S_1;\alpha)\in(\beta,\infty)\times A} \widehat{\theta}_{[1]}(s;\alpha)\, d\rho(\alpha, t_0, S)$$
$$\geq \epsilon\beta \int_{\mathbb{R}^k} \mathbf{1}_{\widehat{\theta}(S_1;\alpha)\in(\beta,\infty)\times A}\, d\rho(\alpha, t_0, S) \tag{164}$$
$$= \epsilon\beta \int_{(\beta,\infty)\times A} d\rho(\theta, t_0, S_1) \geq \epsilon\beta\gamma.$$

Since

$$\lim_{s\to\infty} \int_{(\beta,\infty)\times A} |\theta_{[1]}|^2 d\rho(\theta, t_0, s) \geq \lim_{s\to\infty} \int_{\mathbb{R}^k} \mathbf{1}_{\widehat{\theta}(S_1;\alpha)\in(\beta,\infty)\times A} \left|\widehat{\theta}_{[1]}(s;\alpha)\right|^2 d\rho(\alpha, t_0, S) = \infty,$$

we finally obtain, using (164), that

$$\lim_{s \to \infty} \int_{\mathbb{R}^k} |\theta|^2 \, \mathrm{d}\rho(\theta, t_0, s) \geq \lim_{s \to \infty} \int_{(\beta, \infty) \times A} |\theta_{[1]}|^2 \, \mathrm{d}\rho(\theta, t_0, s) = \infty \,.$$

This limit contradicts (154), implying that (155) must hold, as claimed. ∎

### H.3.1    Claim in the proof of Theorem 6.2

In this section, we prove the statement in (163), meaning that we need to find $S_1 \geq S$, $\beta > 0$, and $\gamma > 0$ such that

$$\int_{(\beta, \infty) \times A} \mathrm{d}\rho(\theta, t_0, S_1) > \gamma \,. \tag{165}$$

Supposing that $\int_A \mathrm{d}\rho(\theta, t_0, S) > 0$, then by making $\beta$ and $\gamma$ small enough, (165) is satisfied naturally.

If $\int_A \mathrm{d}\rho(\theta, t_0, S) = 0$, then it suffices to show that there exists $S_1 > S$ such that

$$\int_A \mathrm{d}\rho(\theta, t_0, S_1) > 0 \,. \tag{166}$$

Define $h_s(\theta_{[2]})$ and $h_\infty(\theta_{[2]})$ as in (156). Because of the fourth condition in Theorem 6.2, there exists a function $h'(\widetilde{\theta})$ on $\mathcal{C}^1\left(\mathbb{S}^{k-2}\right)$ such that

$$h_\infty(r\widetilde{\theta}) \xrightarrow{r \to \infty} h'(\widetilde{\theta}), \quad \text{in } \mathcal{C}^1(\mathbb{S}^{k-2}) \,.$$

Combining this with (157), there exists $h^* > \epsilon/2$ such that $\|h_s\|_{L_A^\infty} \leq h^*$. Further, for any $\xi > 0$, we can find $\theta_{[2]}^* \in A$ and $S'$ large enough such that

$$|\nabla_{\theta_{[2]}^*} h_s(\theta_{[2]}^*)| < \xi, \quad \forall s \geq S' \,. \tag{167}$$

According to Lemma H.3, there exists $r_{S'} > 0$ such that $\rho(\theta_{[1]}, \theta_{[2]}, t_0, S')$ separates $\{-r_{S'}\} \times \mathbb{R}^{k-1}$ and $\{r_{S'}\} \times \mathbb{R}^{k-1}$. Considering the set $[-r_{S'}, r_{S'}] \times \{\theta_{[2]}^*\}$, it must intersect the support of $\rho(\theta_{[1]}, \theta_{[2]}, t_0, S')$ due to the separation property. Thus, any open set that contains $[-r_{S'}, r_{S'}] \times \{\theta_{[2]}^*\}$ must have a positive measure in $\rho(\theta_{[1]}, \theta_{[2]}, t_0, S')$. Because $\int_A \mathrm{d}\rho(\theta, t_0, S') = 0$ and $(-r_{S'} - 1, \infty) \times A$ is a open set that covers $[-r_{S'}, r_{S'}] \times \{\theta_{[2]}^*\}$, there exists a open set $U \subset (-r_{S'} - 1, 0] \times A$ such that

$$\int_U \mathrm{d}\rho(\theta_{[1]}, \theta_{[2]}, t_0, S') > 0 \,.$$

Thus, we can find a point $0 < r^* \leq r_{S'} + 1$ and an arbitrary small $\sigma > 0$ such that

$$\mathcal{B}_\sigma(-r^*, \theta_{[2]}^*) \subset (-r_{S'} - 1 - \sigma, \sigma) \times A, \quad \text{and} \quad \int_{\mathcal{B}_\sigma(-r^*, \theta_{[2]}^*)} \mathrm{d}\rho(\theta, t_0, S') > 0 \,.$$

Recalling the system (161), we claim the following:

> *When $\xi, \sigma$ are small enough, there exists $S_1 > S'$ such that $\widehat{\theta}(S_1; \alpha) \in \mathcal{A}$ for any $\alpha \in \mathcal{B}_\sigma(-r^*, \theta_{[2]}^*)$.*

If this claim is true, then

$$\int_A \mathrm{d}\rho(\theta, t_0, S_1) \geq \int_{\mathcal{B}_\sigma(-r^*, \theta_{[2]}^*)} \mathrm{d}\rho(\theta, t_0, S') > 0 \,.$$

which proves (166) and the lemma.

Now, we prove the claim. Because $f$ satisfies Assumption 6.2 and the second moment of $\rho$ is uniformly bounded in $s$, for all $s > 0$, we have

$$\left| \frac{\delta E(\rho(s))}{\delta \rho}(1, \theta_{[2]}, t_0) - \frac{\delta E(\rho(s))}{\delta \rho}(1, \theta'_{[2]}, t_0) \right| \leq L |\theta_{[2]} - \theta'_{[2]}|,$$

$$\left| \nabla_\theta \frac{\delta E(\rho(s))}{\delta \rho}(1, \theta_{[2]}, t_0) - \nabla_\theta \frac{\delta E(\rho(s))}{\delta \rho}(1, \theta'_{[2]}, t_0) \right| \leq L |\theta_{[2]} - \theta'_{[2]}|, \tag{168}$$

for some constant $L$. According to (161), we have

$$\begin{cases} \dfrac{\mathrm{d}\widehat{\theta}_{[1]}(s;\alpha)}{\mathrm{d}s} = -h_s\left(\widehat{\theta}_{[2]}(s;\alpha)\right) \\[4mm] \dfrac{\mathrm{d}\left|\widehat{\theta}_{[2]}(s;\alpha) - \theta^*_{[2]}\right|^2}{\mathrm{d}s} \leq 2\left|\widehat{\theta}_{[2]}(s;\alpha) - \theta^*_{[2]}\right|\left|\widehat{\theta}_{[1]}(s;\alpha)\right|\left|\nabla_{\theta_{[2]}} h_s\left(\widehat{\theta}_{[2]}(s;\alpha)\right)\right| \end{cases}, \quad s \geq S' \tag{169}$$

where

$$-r^* - \sigma \leq \widehat{\theta}_{[1]}(S';\alpha) \leq \sigma, \quad \text{and} \quad \left|\widehat{\theta}_{[2]}(S';\alpha) - \theta^*_{[2]}\right| \leq \sigma. \tag{170}$$

To prove the claim, it suffices to show that there exists $S_1 > S'$ such that

$$\widehat{\theta}_{[1]}(S_1;\alpha) > 1 \quad \text{and} \quad \widehat{\theta}_{[2]}(S_1;\alpha) \in A. \tag{171}$$

We first show that $\widehat{\theta}_{[1]}$ increases and the right hand-side of (169) is bounded. Since $A$ is a open set, there exists an arbitrary small $\Sigma > 0$ such that $\mathcal{B}_\Sigma(\theta^*_{[2]}) \subset A$. We first choose $\sigma < \min\{\Sigma, 2\}$. When

$$\left|\widehat{\theta}_{[1]}(s;\alpha)\right| \leq \frac{2r_{S'} + 2}{\epsilon} h^* + 2, \quad \left|\widehat{\theta}_{[2]}(s;\alpha) - \theta^*_{[2]}\right| \leq \Sigma, \tag{172}$$

we have from (158), (167), and (168) that

$$\frac{\mathrm{d}\widehat{\theta}_{[1]}(s;\alpha)}{\mathrm{d}s} = -h_s\left(\widehat{\theta}_{[2]}(s;\alpha)\right) < h^*, \quad \frac{\mathrm{d}\widehat{\theta}_{[1]}(s;\alpha)}{\mathrm{d}s} = -h_s\left(\widehat{\theta}_{[2]}(s;\alpha)\right) > \epsilon/2 \tag{173}$$

and

$$\frac{\mathrm{d}\left|\widehat{\theta}_{[2]}(s;\alpha) - \theta^*_{[2]}\right|^2}{\mathrm{d}s}$$

$$\leq 2\left(\frac{2r_{S'} + 2}{\epsilon} h^* + 2\right)\left|\widehat{\theta}_{[2]}(s;\alpha) - \theta^*_{[2]}\right|\left(L\left|\widehat{\theta}_{[2]}(s;\alpha) - \theta^*_{[2]}\right| + \xi\right) \tag{174}$$

$$\leq 2L\left(\frac{2r_{S'} + 2}{\epsilon} h^* + 2\right)\left|\widehat{\theta}_{[2]}(s;\alpha) - \theta^*_{[2]}\right|^2 + 2\left(\frac{2r_{S'} + 2}{\epsilon} h^* + 2\right)\Sigma\xi.$$

When $s = S'$ and $\alpha \in \mathcal{B}_\sigma(-r^*, \theta^*_{[2]})$, we have from (170) that

$$|\widehat{\theta}_{[1]}(S';\alpha)| \leq r_{S'} + 1 + \sigma < \frac{2r_{S'} + 2}{\epsilon} h^* + 2, \quad |\widehat{\theta}_{[2]}(S';\alpha) - \theta^*_{[2]}| \leq \sigma < \Sigma.$$

Thus, for $s$ slightly larger than $S'$, we still have that

$$\left|\widehat{\theta}_{[1]}(s;\alpha)\right| < \frac{2r_{S'} + 2}{\epsilon} h^* + 2 \quad \text{and} \quad \left|\widehat{\theta}_{[2]}(s;\alpha) - \theta^*_{[2]}\right| < \Sigma.$$

Denote by $S^*$ the first time that

$$\left|\widehat{\theta}_{[1]}(S^*;\alpha)\right| \geq \frac{2r_{S'} + 2}{\epsilon} h^* + 2 \quad \text{or} \quad \left|\widehat{\theta}_{[2]}(S^*;\alpha) - \theta^*_{[2]}\right| \geq \Sigma.$$

Then we show that there exists $S_1 \in [S', S^*]$ such that (171) is satisfied when $\sigma$, $\Sigma$, and $\xi$ are small enough. From (170), we have for $s \in (S', S^*)$ that

$$\widehat{\theta}_{[1]}(s;\alpha) \leq \sigma + (s - S')h^*$$

$$\widehat{\theta}_{[1]}(s;\alpha) > -r_{S'} - \sigma + (s - S')\epsilon/2,$$

$$\left|\widehat{\theta}_{[2]}(s;\alpha) - \theta^*_{[2]}\right| < \exp\left(L\left(\frac{2r_{S'} + 2}{\epsilon} h^* + 2\right)(s - S')\right)\left(\sigma^2 + 2(s - S')\left(\frac{2r_{S'} + 2}{\epsilon} h^* + 2\right)\Sigma\xi\right)^{1/2},$$

where the first two inequalties come from (173) and last inequality comes from (174) via Grönwall's inequality. Defining

$$S_1 = \frac{2r_{S'} + 2}{\epsilon} + S',$$

we can choose the positive values $\sigma$, $\Sigma$, and $\xi$ small enough that

$$\widehat{\theta}_{[1]}(S_1, \theta^{S'}) > -r_{S'} + (S_1 - S')\epsilon/2 = 1,$$

and for $s \in [S', S_1]$

$$\left| \widehat{\theta}_{[1]}(s; \alpha) \right| < \frac{2r_{S'} + 2}{\epsilon} h^* + 2, \quad \left| \widehat{\theta}_{[2]}(s; \alpha) - \theta^*_{[2]} \right| < \Sigma. \tag{175}$$

According to (175), the bounds (172) are satisfied for $s \in [S', S_1]$, which implies that $S_1 < S^*$ and $\widehat{\theta}_{[2]}(S_1, \theta^{S'}) \in A$. Further, we have $\widehat{\theta}_{[1]}(S_1, \theta^{S'}) > 1$. By combining these two results, we conclude that (171) is satisfied with the chosen values of $\sigma$, $\Sigma$, $\xi$, and $S_1$. Thus, we have $\widehat{\theta}(S_1; \alpha) \in (1, \infty) \times A \subset \mathcal{A}$ for any $\alpha \in \mathcal{B}_\sigma(-r^*, \theta^*_{[2]})$, which proves the claim.

