# OpenReview forum: "On the Global Convergence of Gradient Descent for multi-layer ResNets in the mean-field regime"
_ICLR.cc/2022/Conference — ICLR 2022 Submitted_

### Official Review · Reviewer_YcBa · 2021-11-02

**Correctness:** 3
**Technical Novelty And Significance:** 2
**Empirical Novelty And Significance:** Not applicable
**Recommendation:** 8
**Confidence:** 5

**Main Review:**

Many thanks to the authors for addressing my concerns and correcting the mistake.

As said in my review, there are reasons to believe why the contribution of this paper is a clearly meaningful one on top of prior works in similar resnet mean field setups, namely the removal of the full support assumption on $\rho_{t=\infty}$. Though again the technicality is not necessarily surprising, this is hard work. My final recommendation would have been 7 if possible; in support of this paper, I'm making 8.

==============================

There is a lot of interests in the mean field analysis for neural networks, so the subject of the paper is timely. Before this work, Lu et al 2020 attempted to analyze a very similar problem setup, but had to make the full support assumption of $\rho_{t=\infty}$ for their global convergence result. The current paper fills in this gap via Theorems 6.1 and 6.2, along with Theorem 5.1 which shows that the wide and deep resnet can be approximated well by the corresponding continuous MF model.

It should be stressed that this assumption on $\rho_{t=\infty}$ is a very strong one (as pointed out in Pham and Nguyen 2021) and is unrealistic in certain situations. Removal of this assumption tends to be highly technical; one can just look at the length of the proof of this paper, compared to that in Lu et al 2020. As such, the paper has good and meaningful technical contents.

In that sense, my overall evaluation is that the paper gives completion to Lu et al 2020, though I am not surprised by the result nor the techniques. In particular, while I cannot check the lengthy proof details, I do not doubt the results, and the arguments in the presented proof are within expectation. Let me go into further details of the global convergence proof of Theorems 6.1 and 6.2:
- Compared to Lu et al 2020, perhaps the one piece that differs is Proposition 6.1, which recognizes that the condition of global optimum can be checked at any residual block, unlike what was done in Lu et al 2020 which sort of integrates over all blocks. The key is the appearance of $g(Z_\rho) - y$ term in the gradient update of every residual block, and any place where this term appears can be exploited. I must say, though, this observation is not new and has been used elsewhere in the mean field literature.
- Since the problem comes down to any of the residual blocks, which are just two-layer MF neural nets, the entire stability argument from Chizat and Bach 2018 can be replicated. Another piece of technical work is in proving the support property of $\rho(s)$ for any time $s$, and this is again a direct application of the topological degree lemma in Chizat and Bach 2018.

To add to this point, there are results which are proven with similar analyses (with the Sard-type regularity and the stability argument) on setups other than two-layer nets; examples include Theorem 50 in Nguyen and Pham 2020 for deep feedforward nets and Theorem 11 in Agazzi and Lu 2020 for temporal difference dynamics. In other words, this analysis can be carried out (with some technical confidence, admittedly) for the right assumptions.

One major issue is that Theorem 5.1 and Theorem 6.2 cannot be merged to claim Theorem 3.1 the main result. As far as I understand, Theorem 5.1 assumes compactly supported initialization distributions. This compact support assumption makes it possible to prove well-posedness and eases various parts of the proof of Theorem 5.1. However it is important NOT to have initializations with a finite support for the global convergence results (Theorem 6.2) to hold. Due to this mismatch, one cannot claim Theorem 3.1. To be fair, the development in Chizat and Bach 2018 also goes along the same line; the issue can be avoided by either:

- assuming a compact domain for the first layer weight with an appropriate version of GD (which Chizat and Bach 2018 covers, but the current paper does not),

- working directly with projection onto the sphere in the 2-homogenous case (which Chizat and Bach 2018 does, but the current paper does not).

This mismatch in the conditions is technical and should be an interesting mathematical problem to fill in, but does not seem to pose any conceptual issue in machine learning. As such, in light of the established literature, I would not evaluate strongly against the paper, but I ask that the paper remove Theorem 3.1 and make a clear acknowledgment of the mismatch.

Another request is that the paper should also make more efforts in discussing the last Sard-type regularity condition in Theorem 6.2. While the paper claims this is a very mild assumption and refers to Remark 4.1, this remark only concerns with the third Sard-type condition and not the last one. In my opinion, it is this last condition that is extremely difficult to verify (the third one can be satisfied for sufficiently smooth activation functions). This is because it involves dealing with points that are at infinity. Again since this assumption is adopted from Chizat and Bach 2018, I would not evaluate strongly against the paper.

Minor comments:
- $\epsilon$ and $\eta$ are not explained in the first page.
- The equation before (3) seems to be missing a factor of $ML$.
- The paper should clarify what “regularized ReLU” is; this is not a standard term.
- The example activations in Remarks 6.1 and 6.2 are somewhat contrived; can the paper give more familiar examples?

References:

L. Chizat, F. Bach, On the global convergence of gradient descent for over-parameterized models using optimal transport, NeurIPS 2018.

Y. Lu, C. Ma, Y. Lu, J. Lu, L. Ying, A mean field analysis of deep resnet and beyond: Towards provable optimization via overparameterization from depth, ICML 2020.

H. T. Pham, P.-M. Nguyen. Global convergence of three-layer neural networks in the mean field regime, ICLR 2021.

P.-M. Nguyen, H. T. Pham. A rigorous framework for the mean field limit of multilayer neural networks, 2020

A. Agazzi, J. Lu, Temporal-difference learning with nonlinear function approximation: lazy training and mean field regime, 2020.


**Summary Of The Paper:**

The paper proves a global convergence result for GD in resnets, whose residual blocks are mean field two-layer neural nets. This is taken with a double limit, where the depth limit is the neural ODE and the width limit is the two-layer MF limit. To prove global convergence of the infinite-depth/width limit, the paper employs previous techniques that assume Sard-type regularities and the accompanying stability argument from Chizat and Bach 2018.

**Summary Of The Review:**

This is a technical paper that removes a strong assumption from the analysis of Lu et al 2020, though the techniques are not surprising. A number of edits are necessary. Though giving a confirmation to a result that is expected, the paper is still a good theoretical contribution, in light of the heavy technical works that are required.

---

> ### Author Response · Authors · 2021-11-16
> **Response to Reviewer YcBa**
>
> We thank the reviewer for the kind words.  While we agree with some of the reviewer's comments, we have a different view of others.  We note first that the full-support assumption is not used at all in this paper. (In fact, we view the dropping of this assumption as the main achievement of the paper.) Upon obtaining the mean-field limit (Theorem 5.1), we go on to show global convergence in Theorems 6.1 (6.2) using the homogeneity assumption. The conditions for Theorem 5.1 and 6.1 (6.2) match each other fully: both require the initial condition to be admissible (Definition 4.2), meaning that the initial condition for $\rho$ is compactly supported.
> The argument that we use to show the global convergence, exactly as the reviewer suggests, employs the homogeneity assumption explored in [3]. (This assumption is critical to  getting rid of the full-support condition assumed in [2].)
> In summary, our result Theorem 6.1 and 6.2 can be viewed as the extension of [3] from 2 layers to multiple layers.
>
> We agree with the reviewer that [1] is a great paper in this direction. It contains some beautiful mathematical derivations that are useful in our paper. Our setup is, however, quite different, as the reviewer mentioned. In [1], the blocks are all integrated out.
> This is a natural consequence of assuming that $\rho$ is a probability measure on the full $(\theta,t)$-space.
> This assumption certainly simplifies the proof, but we cannot find an NN that can be represented by this structure.
> Instead, we assume $\rho$ to be a probability measure only on $\theta$-space for each  $t$: This is the true mean-field limit of the conventionally used ResNets. From this perspective, we have a different (more practical) setup than [1], even though both deal with multi-layer NNs.
>
> We also agree with the reviewer's comment on the Sard-type regularity. This assumption appears because we adopted the techniques from [3] and thus naturally inherited their assumptions. Dropping the Sard-type condition is an interesting future research direction.
>
> We will correct all notations that are inconsistent in the paper. We thank the reviewer for pointing them out.
>
> $\textbf{References:}$
>
> [1] Yiping Lu, Chao Ma, Yulong Lu, Jianfeng Lu, Lexing Ying. A Mean-field analysis of deep ResNet and beyond: Towards provable optimization via overparameterization From depth, ICML, 2020.
>
> [2] Zhiyan Ding, Shi Chen, Qin Li, Stephen Wright, Overparameterization of deep ResNet: zero loss and mean-field analysis, arXiv, 2021.
>
> [3] Lenaic Chizat, Francis Bach, On the global convergence of gradient descent for over-parameterized models using optimal transport, NeurIPS, 2018.

---

> > ### Comment · Reviewer_YcBa · 2021-11-16
> > **reply**
> >
> > Dear authors,
> >
> > A quick question. In, say, Theorem 6.2, it is assumed that $\rho_{ini}$ has compact support, but at the same time, it also separates the two sets $(-r_0)\times R^{k-1}$ and $(+r_0)\times R^{k-1}$. Doesn't the latter imply that at initialization the first layer's weight distribution has full support? If so, the two assumptions collide, which is the problem I mentioned in my review and which renders Theorem 6.2 and Theorem 5.1 incompatible.
> >
> > Let me know if I misunderstand something.

---

> > > ### Author Response · Authors · 2021-11-16
> > > **Response to the question**
> > >
> > > Thank the reviewer for pointing out this mistake. We admit that we miss this part. The current Theorem 6.2 and Theorem 5.1 can not be combined together.
> > >
> > > However, we have a way to fix it fast. According to Assumption 6.2 issue 2, $\hat{f}(x,\theta_2)$ is a nice bounded function. Then, even $\theta_2$ is not in a compact set, we can still prove a similar result like Theorem 3.1. We will fix it in the revision.
> > >
> > > Besides, Theorem 6.1 and Theorem 5.1 are compatible. And the proof of Theorem 6.1 is what the reviewer suggests in the comments.

---

> > > > ### Comment · Reviewer_YcBa · 2021-11-16
> > > > **reply**
> > > >
> > > > Thanks for the quick reply. If the activation is bounded then I agree, it can be fixed. I imagine some subgaussian assumption to be made on the initial distribution of the first layer for this to work. Maybe some changes to the proofs to handle the technicality, not a simple tweak but not much more than some exercise. And some changes to the description in the main text, since $\rho_{ini}$ is no longer admissible in this case.
> > > >
> > > > Upon another quick glance, I've also realized some references that mainly concern with NTK being placed in the text that concerns with MF in Section 3, and Fang et al being placed in the text that concerns with NTK. Also some typos to names in the references: B. Poczos --> B. Póczos, P. Nguyen --> P.-M. Nguyen.

---

> > > > > ### Author Response · Authors · 2021-11-18
> > > > > **Response to Reviewer YcBa**
> > > > >
> > > > > We appreciate the reviewer for pointing out the hidden inconsistency in the statement of Theorem 5.1 and Theorem 6.2. In the revision, we have fixed this problem. In particular, we now rephrase the locally Lipschitz condition of $\nabla f$ (in original Assumption 4.1 item 2) to formulate a general condition that fits both the situation in Theorem 6.1 and the situation in Theorem 6.2. In particular, we assume: There exists $k_1$ with $0<k_1\leq k$ with the following property: Denoting $\theta=(\theta_{[1]},\theta_{[2]})$, $r=\max\{|x|,|\theta_{[1]}|\}$,
> > > > > where $\theta_{[1]}\in\mathbb{R}^{k_1}$, $\theta_{[2]}\in\mathbb{R}^{k-k_1}$, we have for $C_3(r)$ monotonically increasing with respect to $r$ that the following bounds hold:
> > > > > \begin{equation}
> > > > > \left|\partial^i_x\partial^j_\theta f(x,\theta)\right|\leq C_3(r),\quad i+j=2,\ i,j \geq0\,.
> > > > > \end{equation}
> > > > > When $k_1<k$, we have in addition that
> > > > > \begin{equation}
> > > > > \max\{|\partial_xf|,|f|\}\leq C_3(r)\left(|\theta_{[2]}|+1\right),\ \left|\partial_{\theta_{[1]}}f\right|\leq C_3(|x|)\left(|\theta_{[1]}|+1\right)\,.
> > > > > \end{equation}
> > > > > When the parameter $k_1$ is chosen to be $k$, the dimension of $\theta$, it covers the case (the 2-homogeneous condition) in Theorem 6.1, and when it is chosen to be $1$, it covers the partially 1-homogeneous case in Theorem 6.2. This condition essentially separates $\theta$ into $\theta_{[1]}$ and $\theta_{[2]}$ components, with the $\theta_{[2]}$ component bounded in $f$ (roughly). This boundedness allows us to relax the compactly supported condition required for this component in the initial data in Theorem 5.1. We will upload our revision shortly.

---

> > > > > > ### Comment · Reviewer_YcBa · 2021-11-21
> > > > > > **reply**
> > > > > >
> > > > > > Thanks for the update. It's a bit unclear to me: doesn't the condition for $k_1<k$ also allow the 2-homogeneous case? Assuming $|x|\leq const$ for simplicity, if $f(\theta, x) = \theta_1 ReLU(\theta_2 \cdot x)$, then this condition still seems to hold.

---

> > > > > > > ### Author Response · Authors · 2021-11-21
> > > > > > > **Response to Reviewer YcBa**
> > > > > > >
> > > > > > > We think that 2-homogeneous case can’t satisfy the condition (Assumption 4.1 item 3) with $k_1<k$.  For this example, $f = \theta_1 ReLU(\theta_2 \cdot x)$ doesn’t satisfy the second inequality in (15):
> > > > > > > $
> > > > > > > |\partial_{\theta_1}f|\leq C(|x|)(1+|\theta_1|)
> > > > > > > $
> > > > > > >
> > > > > > > Since $\partial_{\theta_1}f=ReLu(\theta_2 \cdot x)$, it will blow up with $\theta_2$. A correct example is $f = \theta_1 g(\theta_2 \cdot x)$, where $g$ is a bounded function. This is the example we gave in Remark H.2. However, in this setting, $f$ is only partially 1-homogeneous. Actually, for any 2-homogeneous activation function, it doesn't satisfy the second inequality in (15) for any $k_1<k$. Thus, for 2-homogeneous case, technically, we need initial condition to be compactly supported to prove the wellposedness.

---

> > > > > > > > ### Comment · Reviewer_YcBa · 2021-11-21
> > > > > > > > **reply**
> > > > > > > >
> > > > > > > > Apologies, I misread the second condition. Indeed it doesn't satisfy for ReLU. It's also not surprising that for 2-homogeneous case, one requires the compact support condition for the first layer's initial weight distribution for the usual proof to hold.

---

### Official Review · Reviewer_pGat · 2021-11-02

**Correctness:** 3
**Technical Novelty And Significance:** 3
**Empirical Novelty And Significance:** Not applicable
**Recommendation:** 6
**Confidence:** 3

**Main Review:**

This work reports a new mean-field explanation for an infinitely deep and infinitely wide neural network with skip connection and provides global convergence results with the framework of (Chizat & Bach, 2018) under standard assumptions and the universal kernel property exploited in (Lu et al., 2020). My main comments are as follows:

* In P4, the literature review paper list for mean-field limit perspective is not correct, e.g., (Allen-Zhu et al., 2019), (Zhang et al., 2019) are not in that regime.

* In Assumption 4.1, the activation f is required to be C^2. In P6, just below Assumption 4.2, they claim \sigma can be regularized ReLU, which is seems not introduced. Please explain.

* In Theorem 5.1, the final bound seems not uniform for 0<=s<S, which is a much weaker nonasymptotic result than that in (Mei et al., 2018) for two-layer NN.

* The sequential limit, as shown in Theorem E.1 and E.2, is taking limit for L before increasing M, which is similar to the layer-wise scaling used in (Sirignano & Spiliopoulos, 2018). Thus, the M and L would have to grow up at specific speeds simultaneously. So, for curiosity, can we exchange the order of limits of M and L in an asymptotic version of Theorem 3.1?


**Summary Of The Paper:**

This paper studies the gradient flow training dynamics for deep and wide ResNet with the mean-field scaling. For a similar setting as (Lu et al., 2020), they report a limiting model with finite-dimensional approximation, and its global convergence result. The authors also justify the well-posedness of the resulting distributional dynamics, which seems only heuristically defined in prior work.

**Summary Of The Review:**

Overall, I think this paper has useful ideas for understanding the training dynamics of ResNet in the nonlinear regime, and it complements the existing research in the literature.

---

> ### Author Response · Authors · 2021-11-16
> **Response to Reviewer pGat**
>
> We thank the reviewer for the positive feedback and the comments. Our responses to reviewer's main comments are listed in the following:
>
> $\textbf{Regarding the reference problem:}$
>
> Thanks for the good catch! Some references were placed incorrectly. We will fix it.
>
> $\textbf{Regarding the example of Regularized ReLU:}$
>
> We apologize for omitting details of the regularization of ReLU in the paper; we will add them in the revision. One way to construct regularized ReLU is to set
>
> $\sigma(x)=0$, for $x<-\eta$
>
> $\sigma(x)=(x+\eta)^2/(4\eta)$, for $x\in[-\eta,\eta]$
>
> $\sigma(x)=x$, for $x>\eta$
>
> for a small $\eta$,
> which smooths the transition between $0$ and $x$.
>
> $\textbf{Regarding the uniform bound in $s$ and the non-asymptotic result, compared with [1]:}$
>
> We agree that we do not have a uniform bound on $s$, but we believe that we may have a different understanding from the reviewer of the results in [1]. To our knowledge, there has {\bf not} been any result in the mean-field regime (including ours and [1]) proving non-asymptotic global convergence in training time $T$. From what we can see,  [1, Theorem 3] also does not have a uniform bound on the mean-field convergence rate. The convergence rate depends on the training time $T$ (see Theorems 1,2 of [1]), similar to ours. Though we do not have proof, we suspect it is not possible to make this bound uniform in $T$ (our $s$). We believe that most mean-field results rely on the coupling method, to derive an equation that traces the propagation of error in time. When certain types of regularity are used (for example, Lipschitz condition), Gronwall's inequality usually gives the error term growing in time. However, we do not see this property as a severe drawback. Indeed, in order to achieve the ultimate convergence of the cost function, we first tune $s$ (or $T$) to be large enough so the convergence of the mean-field limiting system is achieved. Then we tune $M$ and $L$ to ensure the discrete system (NN) is close to the mean-field limiting system. In a sense, $s$ is already determined by the time that  $L$ and $M$ are set.
>
> $\textbf{Regarding exchanging the order of limits of $L$ and $M$:}$
>
> We thank the reviewer for this very interesting question, which we have also considered. We believe the answer to be positive, in the sense that one can pass the $M$ limit before the $L$ limit. The techniques would be similar but the derivation would be somewhat different. Given that the paper is already long, we did not pursue this new direction.
>
> $\textbf{References:}$
>
> [1]. Song Mei, Andrea Montanari, Phan-Minh Nguyen. A mean field view of the landscape of two-layer neural
> networks. Proceedings of the National Academy of Sciences, 115(33), 2018.

---

### Official Review · Reviewer_Q3Rw · 2021-11-03

**Correctness:** 4
**Technical Novelty And Significance:** 3
**Empirical Novelty And Significance:** Not applicable
**Recommendation:** 5
**Confidence:** 3

**Main Review:**

This work analyzes infinitely deep and infinitely wide residual networks. Overall, I found the writing to be quite good, and these types of results are of interest to the community (at least they interest me). I have not been able to check the details of the proofs, but they seem plausible.

The limit of viewing infinitely deep ResNets as an ODE is due to the Neural ODE paper, and further taking the infinite width limit to consider the mean-field regime was considered in [1]. Therefore, the main contribution of this work is in analyzing the training dynamics of regular gradient flow on this network ([1] considered a different, unusual training mechanism).

However, I would like to get more clarity on the incremental novelty. As the authors acknowledge, the paper is very similar to [2]. Since this paper is meant to be a standalone paper distinct from [2], I would like to get some clarification as to what are the specific contributions not present in [2].

The results of section 6 seem to be the global convergence of the continuous limit, is the core contribution. The technical challenges that had to be overcome appear to be quite similar to those considered in [3]. I would like to hear from the authors about the novelty of the proof technique compared to that presented and used in [3].


[1] Yiping Lu, Chao Ma, Yulong Lu, Jianfeng Lu, Lexing Ying. A Mean-field Analysis of Deep ResNet and Beyond: Towards Provable Optimization Via Overparameterization From Depth, ICML, 2020.
[2] Zhiyan Ding, Shi Chen, Qin Li, Stephen Wright, Overparameterization of deep ResNet: zero loss and mean-field analysis, arXiv, 2021.
[3] Huy Tuan Pham, Phan-Minh Nguyen, Global Convergence of Three-layer Neural Networks in the Mean Field Regime, ICLR, 2021.


**Summary Of The Paper:**

This work analyzes infinitely deep and infinitely wide residual networks. The infinite depth limit is the neural ODE. The infinite width limit is the mean-field limit. The main contribution establishing global convergence of the infinitely wide and deep ResNet. Finally, the global convergence of the infinite limit is translated to the finite but large setup.

**Summary Of The Review:**

Interesting results, but contribution should be clarified.

I am willing to raise the score if the issue of novelty is clarified.

---

> ### Author Response · Authors · 2021-11-16
> **Response to Reviewer Q3Rw**
>
> We thank the reviewer for their constructive and helpful comments. However, we feel that we may have misled the reviewer regarding the contribution of our paper. We plan to clarify our contributions compared with previous work in the revision. We give further details below.
>
> $\textbf{Compare with [1], [2]:}$
>
> We thank the reviewer for mentioning these two papers, both of which we cite. In particular, we are very well aware of the technicalities in the second paper and have full knowledge of the similarities and differences between this paper and ours. As discussed in our response to Reviewer 1, we plan to rewrite the text before and after Theorem 3.1 to describe our contributions more clearly. In particular, with the elimination of the ``full-support'' condition on $\rho$, the current paper is significantly more challenging from a technical viewpoint than is [1].
>
> 1. In [1], the authors need to make a full-support assumption on $\rho$ (see [1, Theorem 7]).  As discussed in [1], this assumption is quite strict and not practical. We are able to dispense with this assumption in the current paper, at the expense of considerable technical complications. Without the full support condition, there is little structure in the gradient flow that can be used to prove convergence, so the 2-homogeneity assumption on the activation function (adopted from [5]) becomes important. The new proof (considerably longer than one page!) is found in Appendix H.2 and H.3 of the current paper.
>
> 2. In [2], the authors also study the mean-field limit of training ResNets. But this paper is even further away from the current paper than is [1]. While [2] considers multi-layer NN structure, it absorbs the "$t$" dependence into the parameter configuration of $\rho$, making $\rho$ a probability measure in the entire $(\theta,t)$-space. The gradient flow thus becomes standard. By contrast,  our paper requires  $\rho$ to be a probability measure in $\theta$, for each time $t$, a more natural setup. We are therefore using the gradient flow not to find a probability measure, but a stochastic process in time.  We should mention that [2] includes some beautiful mathematics, and is a cornerstone of the area, but we do not believe that practical NNs can be defined in a way that makes it natural to view $\rho(\theta,t)$ as a probability measure jointly in $\theta$ and $t$. For this reason, we set out to study a more practical set up in the current paper.
>
> We agree with the reviewer that a clear and precise statement of our contributions is crucial, and we plan to rewrite the discussion surrounding Theorem 3.1 to identify the above-mentioned differences.
>
> $\textbf{Compare with [3]:}$
>
> We thank the authors for mentioning [3]. We believe the results of this paper are covered in another article by the same authors [4], which we cite. We agree with the reviewer that the contribution of both papers [3,4] is on the global convergence in the training time, but the technical issues are very different from ours. In particular, In [4], $z^{k+1}=\theta^k\phi(z^k)$, where $z^k$ is the neuron, $\theta^k$ is the weight, and $\phi$ is a nonlinear activation function. This structure cannot be formulated as a ResNet, and thus the continuous-limit aspect is entirely missed. This particular form of transition from one layer to the next helps greatly in reducing the technical difficulties: The "activation function" $\theta^k\phi(\cdot)$  is linear in  $\theta$, and thus $1$-homogeneous. In contrast, we can deal with both $2$-homogeneous and partially $1$-homogeneous activation functions (both nonlinear in $\theta$), so the class of activation functions that we can handle is significantly larger. Our techniques for global convergence are mostly borrowed from [5].
>
> $\textbf{References:}$
>
> [1] Zhiyan Ding, Shi Chen, Qin Li, Stephen Wright, Overparameterization of deep ResNet: zero loss and mean-field analysis, arXiv, 2021.
>
> [2] Yiping Lu, Chao Ma, Yulong Lu, Jianfeng Lu, Lexing Ying, A Mean-field Analysis of Deep ResNet and Beyond: Towards Provable Optimization Via Overparameterization From Depth. ICML 2020.
>
> [3] Huy Tuan Pham, Phan-Minh Nguyen, Global Convergence of three-layer neural networks in the mean field regime, ICLR, 2021.
>
> [4] Huy Tuan Pham, Phan-Minh Nguyen, A rigorous framework for the mean field limit of multilayer neural networks, arXiv, 2021.
>
> [5] Lenaic Chizat, Francis Bach, On the global convergence of gradient descent for over-parameterized models using optimal transport, NeurIPS, 2018.

---

### Official Review · Reviewer_jaQt · 2021-11-06

**Correctness:** 4
**Technical Novelty And Significance:** 2
**Empirical Novelty And Significance:** Not applicable
**Recommendation:** 3
**Confidence:** 4

**Main Review:**


The proof of this paper seems reasonable and the assumptions and theorems are well-presented. It contributes to an interesting and important research direction, which is to study the convergence of neural ODE/PDEs. However, I have several concerns about this paper.

1. The contributions of this paper are kind of incremental given existing works [1,2]. The main theorems in these papers and this submission are quite similar. In fact, the contributions given in Section 3 are almost identical to the contributions listed in [2], which is concerning. Below Theorem 3.1, it is only discussed how it is nontrivial to extend existing results in the mean-field regime to multilayer networks, but a detailed explanation of how this paper (in terms of the convergence results) differs from [1,2] is missing.

[1] Lu et. al., A mean field analysis of deep ResNet and beyond: Towards provably optimization via overparameterization from depth. ICML 2020.

[2] Ding et. al., Overparameterization of deep ResNet: zero loss and mean-field analysis. arXiv preprint arXiv:2105.14417.

2. It is also not very clear what is the advantage of this paper compared with prior works on mean-field or NTK. For mean-field, while the mean-field works on two-layer networks are not applicable to multi-layer networks, this paper does not seem to be applicable to any practical multi-layer network architectures either, as this paper requires the network depth to go to infinity. So the authors may need to comment on why studying this infinite limit is interesting. Besides, in (1), is it standard to consider a scaling parameter $1/L$ in each layer? In practice, the residual block is directly added without using this scaling.

3. My next concern is regarding the comparison to the NTK results. In particular, [3] shows that for two-layer NN, when considering updating the “distribution” of model parameters, the obtained solution still exhibits a “kernel-like” behavior. So I would like the authors to discuss more regarding the advantage of the developed results compared to the kernel models, rather than simply mention “NTK views DNN as a kernel model, a rather limited description, so the estimates obtained through NTK can be far from sharp”. One way to show this is to provide some examples to demonstrate the gap between these two regimes.

[3] Chen et. al.,. A generalized neural tangent kernel analysis for two-layer neural networks. NeurIPS 2020.

4. Following my previous comment, the discussion on the advantages of ResNet should also be included, since I believe the main goal of studying ResNet is because of its superior performance compared to fully-connected NN. In the NTK setting, [4] has shown that adding residual structure can improve the dependency on NN depth compared to FC-NN. Then one question is what can we gain by studying the convergence of ResNet in the mean-field regime?

[4] Frei et. al., Algorithm-dependent generalization bounds for overparameterized deep residual networks. NeurIPS 2019.

5. My last question is whether the mean-field analysis can be adapted to classification problems? It has been widely known this can be done in the NTK regime [5, 6] since the convergence analysis in the NTK regime does not require the loss function to be MSE loss but can be applied to any convex loss function. The authors may also need to discuss this point in the mean-field regime.

[5] Li and Liang, Learning Overparameterized Neural Networks via Stochastic Gradient Descent on Structured Data. NeurIPS 2018

[6] Zou et. al., Gradient descent optimizes over-parameterized deep ReLU networks. Machine Learning 2019.


Minor comments

The notations in Section 2, Assumption 6.2, and Remark 6.2 are not very consistent.




**Summary Of The Paper:**

This paper studies the training of a multi-layer ResNet using mean-field tools, where the training procedure of the ResNet is given in its continuous limit as a partial differential equation. A rigorous proof is given, and the convergence to a global minimum of the cost functional is given.

**Summary Of The Review:**

While this paper provides some interesting insights into the training of infinitely deep and wide ResNets, it is not well-demonstrated whether the results are novel and significant enough compared to various existing works. Some detailed discussion and comparison may be very helpful to improve the quality of the paper.

---

> ### Author Response · Authors · 2021-11-16
> **Response to Reviewer jaQt: Part 1**
>
> We thank the reviewer for mentioning several interesting articles which will be helpful in guiding future research.
> Many of the reviewer's comments concern the relationship between NTK and mean-field. Our view is that these are different regimes of NN requiring different perspectives that complement each other, rather than compete. While there is an ocean of literature on the behavior of NN in the NTK regime, there literature on the mean-field limit has been quite thin. We chose to focus on the mean-field regime because some early study reveals many promising results [8]. One important point is that, instead of performing linear approximation, the mean-field regime fully respects nonlinearity in the cost function, thus potentially leading to resolution of issues that cannot be addressed by NTK. Before conducting any comparison between the two regimes, much more work is needed to understand the mean-field regime more fully. We believe that it is too early to declare one regime to be more attractive, relevant, or useful than the other.
>
> The following is our answer to the reviewer's main comments.
>
> $\textbf{Regarding the reviewer's main comment No.1:}$
>
> We thank the reviewer for mentioning the two papers [1,2], both of which are already cited in our original paper.
> In particular, we are very well aware of the technicalities in [2] and have full knowledge of the similarities and differences between the current paper and [2].
> We tried to explain the differences in the text above and below Theorem 3.1, but our description is not adequate, and we will rewrite it.
> In particular, we need to stress that with the elimination of the ``full-support" condition, the technical challenges that must be dealt with in this paper are significantly greater than in [2], we as explain next.
>
> 1. In [2], the authors need the probability density function $\rho$ to have full support; see [2, Theorem 7]. As discussed in [2], this assumption is strict and impractical --- but it removes a lot of technical difficulties.  Indeed,  the relevant proof in [2] is just one page (see Section D.2, page 32). We are able to dispense with this assumption in the current paper, at the expense of considerable technical complications. Without the full support condition, there is little structure in the gradient flow that can be used to prove convergence, so the 2 -homogeneity assumption on the activation function (adopted from [8]) becomes important. The new proof (considerably longer than one page!) is found in Appendix H.2 and H.3 of the current paper.
>
> 2. Mean-field limit of training ResNets is also studied in [1]. But [1] is more distant than [2] from our current paper. While multi-layer NN structure is considered in [1], it absorbs the "$t$" dependence into the parameter configuration of $\rho$, making $\rho$ a probability measure in the entire $(\theta,t)$-space, and thus allowing a standard gradient-flow argument to be used. By contrast, the $\rho$ in our paper is a probability measure in $\theta$,  at each time "$t$". Thus, we are using the gradient flow not to find a probability measure but rather a stochastic process in time. Although [1] brings some beautiful mathematics, and is a cornerstone of the area, we are not aware of any NN that can be designed in such a way that $t$ and $\theta$ are drawn from a single probability measure. This fact motivated us to look for a more practical setup.
>
> We agree with the reviewer that these explanations must appear clearly in the paper, and we will add them to the text surrounding Theorem 3.1.
>
> $\textbf{Regarding the reviewer's main comment No.2:}$
>
> Our $L$ does not necessarily go to $\infty$ because we give a quantitative bound on $L$ in terms of the desired accuracy $\epsilon$. Specifically, to achieve $\epsilon$ accuracy, we need $L\sim\frac{1}{\epsilon}$. This bound allows a DNN of finite size to be chosen in practice. Further, our study reveals that in the setting of our paper, $M$ and $L$ are two independent variables (see Theorem 3.1 or 5.1), in contrast to results in most NTK papers, which require $M$ to be adjusted according to $L$. The scaling $\frac{1}{L}$ is a standard practice in the study of the deep ResNet model. This scaling allows one to pass to the ODE limit; see [3] and the references therein.
> Further, for a given $\epsilon$, $L$ can be determined ahead of time, and this nonzero constant can be absorbed into the activation function $f$.
>
> $\textbf{See references in part 2.}$

---

> ### Author Response · Authors · 2021-11-16
> **Response to Reviewer jaQt: Part 2**
>
> $\textbf{Regarding the reviewer's question No.3:}$
>
> We thank the reviewer for bringing this paper to our attention. We had already encountered this paper before receiving this report, and have been planning to discuss it in the revision of our paper.
>
> As we mentioned above, our view is that NTK and mean-field are complementary rather than competing perspectives. They observe the dynamics of NN in different regimes. While NTK has been studied thoroughly studied, the mean-field perspective has received much less attention. Empirical observation suggests that kernel models are not as general as NN [5-6] and certain (nonlinear) features of NN are not captured in the NTK regime. Because the mean-field perspective respects the nonlinearity in NN fully, this viewpoint could potentially promote a better understanding of NN, particularly of those aspects for which the NTK viewpoint is inadequate.
>
> As the reviewer comments,  [4] describes a  translation of NTK to the mean-field formulation by introducing an extra scaling parameter $\alpha$. When $\alpha$ is large (roughly, $O(\sqrt{M})$), the mean-field limit essentially performs similarly to NTK, showing ``kernel-like'' behavior. This is an interesting observation that bridges the two regimes. However, the setting in this paper is greatly different from ours, since we explicitly require $\alpha=1$, and the mean-field requires $\frac{1}{M}$ in the summation. Thus, this relationship is largely irrelevant to our setup.
>
> We agree with the reviewer that for a fair comparison between NTK and mean-field, some comments in the paper need to be adjusted, and we plan to modify the revision accordingly. But we reiterate our belief that a fuller understanding of NN requires both NTK and mean-field regimes to be studied more thoroughly.
>
> $\textbf{Regarding the reviewer's question No. 4:}$
>
> We thank the reviewer for bringing this article to our attention. It describes the advantages of ResNet over FC-NN in the NTK regime. We will discuss this beautiful observation in our revision.  We mention again that the study of the convergence in the mean-field regime is much less mature than that in the NTK regime. The convergence of training FC-NN is not available yet, and the comparison of the two in the mean-field regime is thus not possible. Our paper sets a cornerstone for the research direction proposed by the reviewer. While NN essentially approximates nonlinear NN structure by a linear model in the infinite-dimensional space, the mean-field treats the nonlinearity directly. This fact may allow us to examine the detailed cost function structure away from the linear approximation region. We aim to make this intuition more concrete in future research, and we believe it is still too early to give further details on the comparison between these two lines of study.
>
> $\textbf{Regarding the reviewer's question No. 5:}$
>
> We thank the reviewer for two more references on NTK. NTK performs very well in many aspects, including the classification problems mentioned by the reviewer. (We will add discussion in our revision to reflect this fact.) Note that convexity of the cost function is not required by the mean-field regime. Even if the cost function on the mismatch is convex, it will nevertheless be nonconvex in the distribution parameters so the role of convexity (in mismatch) is quite limited (see [7]).
>
> $\textbf{References:}$
>
> [1] Yiping Lu, Chao Ma, Yulong Lu, Jianfeng Lu, Lexing Ying, A Mean-field Analysis of Deep ResNet and Beyond: Towards Provable Optimization Via Overparameterization From Depth. ICML 2020.
>
> [2] Zhiyan Ding, Shi Chen, Qin Li, Stephen Wright, Overparameterization of deep ResNet: zero loss and mean-field analysis. arXiv, 2021.
>
> [3] Weinan E, Chao Ma, Lei Wu and Stephan Wojtowytsch, Towards a mathematical understanding of neural network-based machine learning: What we know and what we don't, CSIAM Trans. Appl. Math., 1 (2020), pp. 561-615.
>
> [4] Zixiang Chen, Yuan Cao, Quanquan Gu, Tong Zhang, A generalized neural tangent kernel analysis for two-layer neural networks. NeurIPS, 2020.
>
> [5]. Allen-Zhu and Yuanzhi Li. What can ResNet learn efficiently, going beyond kernels? NeurIPS, 2019.
>
> [6] Sanjeev Arora, Simon S. Du, Wei Hu, Zhiyuan Li, Ruslan Salakhutdinov, Ruosong Wang, On exact computation with an infinitely wide neural net, NeurIPS, 2019.
>
> [7] Huy Tuan Pham, Phan-Minh Nguyen, Global convergence of three-layer neural networks in the mean field regime, ICLR, 2021.
>
> [8] Lenaic Chizat, Francis Bach, On the global convergence of gradient descent for over-parameterized models using optimal transport, NeurIPS, 2018.

---

### Decision · Program_Chairs · 2022-01-20

**Decision:**

Reject

**Comment:**

This paper proposes an improved mean-field analysis for multi-player residual networks. Compared with prior works, the proposed analysis removes a full support assumption needed in prior works. The authors have addressed some of the reviewers’ concerns by adding comparisons with the existing analysis of ResNet in the NTK regime, and a more detailed comparison with Ding et al. 2021. While this paper gathers some support from a reviewer, there is still concern that the novelty of this paper is not significant, especially given that the analysis is heavily built upon prior works. I think this paper can benefit from providing a proof sketch to highlight the key difference between the new analysis and existing analyses,  or explicitly demonstrating the key proof technique/technical lemmas that enable the removal of the full support assumption. This paper might be a strong work after careful revision.